# Risk-Seeking Reinforcement Learning via Multi-Timescale EVaR Optimization

**Deep Ganguly**  *cs22s502@iittp.ac.in*
*Department of Computer Science and Engineering*
*Indian Institute of Technology Tirupati*

**Sarthak Girotra**  *cs20b037@iittp.ac.in*
*Department of Computer Science and Engineering*
*Indian Institute of Technology Tirupati*

**Sirish Sekhar**  *cs20b043@iittp.ac.in*
*Department of Computer Science and Engineering*
*Indian Institute of Technology Tirupati*

**Ajin George Joseph**  *ajin@iittp.ac.in*
*Department of Computer Science and Engineering*
*Indian Institute of Technology Tirupati*

**Reviewed on OpenReview:** *https://openreview.net/forum?id=4nbEgNDsii*

## Abstract

Tail-aware objectives shape agents' behavior when navigating uncertainty and can depart from risk-neutral scenarios. Risk measures such as Value at Risk (`VaR`) and Conditional Value at Risk (`CVaR`) have shown promising results in reinforcement learning. In this paper, we study the incorporation of a relatively new coherent risk measure, Entropic Value at Risk (`EVaR`), as a high-return, risk-seeking objective that the agent seeks to maximize. We propose a multi-timescale stochastic approximation algorithm to seek the optimal parameterized `EVaR` policy. Our algorithm enables effective exploration of high-return tails and robust gradient approximation, to optimize the `EVaR` objective. We analyze the asymptotic behavior of our proposed algorithm and rigorously evaluate it across various discrete and continuous benchmark environments. The results highlight that the `EVaR` policy achieves higher cumulative returns and corroborate that `EVaR` is indeed a competitive risk-seeking objective for RL.

## 1 Introduction

Sequential decision making modelled as a Markov decision process (MDP) (Puterman, 2014) is the underlying formulation of Reinforcement Learning (RL) (Sutton & Barto, 1998). The primary objective in RL is typically to maximize the expected cumulative return, which may be discounted, undiscounted, or averaged (Puterman, 2014), depending on whether the horizon is finite or infinite. Optimizing for expected return has demonstrated remarkable success in structured and less volatile applications such as Atari games, board games, scientific experiments (Chen et al., 2022), and other regularized, simulated environments (Shao et al., 2019; Sethy et al., 2015; Silver et al., 2016) due to stable and predictable environment dynamics. This approach often struggles in environments characterized by high uncertainty and variability in returns, such as safety-critical systems (Zhang et al., 2020; Martín H. & de Lope, 2009), finance (Filos, 2019; Cao et al., 2021), navigation (Wu et al., 2024), industrial automation, healthcare (Wang et al., 2023), and robotics (Kober et al., 2013; Gu et al., 2023), where robustness and adaptability are essential. In such high-stakes scenarios, accounting for risk becomes crucial.

To handle these challenges, reinforcement learning must go beyond simply maximizing the expected return. Instead, it should incorporate risk-awareness by modifying the objective to account for return variability, giving rise to risk-sensitive RL (García & Fernández, 2015; Howard & Matheson, 1972b). The standard expected cumulative reward criterion does not inherently avoid rare but severe negative outcomes, nor does it consider the impact of large positive rewards. Consider two policies $\pi_1$ and $\pi_2$, where $\mathbb{P}_{\pi_1}(\texttt{Reward} = -100) = 0.5$ and $\mathbb{P}_{\pi_1}(\texttt{Reward} = +100) = 0.5$. Also, $\mathbb{P}_{\pi_2}(\texttt{Reward} = -1) = 1.0$. Although $\pi_1$ has a higher expected return, it exhibits extreme volatility, leading to highly variable outcomes which can be categorized as risky. In contrast, $\pi_2$ provides a stable and predictable outcome. The expected cumulative reward criterion would favor $\pi_1$, despite its high risk, highlighting the need for alternative evaluation metrics that account for risk. Classical RL objectives based on expected cumulative reward can be classified as risk-neutral, meaning they do not explicitly consider the uncertainties associated with actions. In contrast, risk-sensitive decision-making can be divided into two approaches: *risk-averse RL* – the agent prioritizes stability, favoring policies with low reward variability, which is crucial in safety-critical applications, and *risk-seeking RL* – the agent pursues higher mean returns, even at the cost of greater potential losses, resembling human decision-making patterns in portfolio management (Gollier, 2001) and super-human racing AI (Wurman et al., 2022; Kaufmann et al., 2023), as explained by cumulative prospect theory (Tversky & Kahneman, 1992). By integrating risk-sensitive objectives into RL, agents can achieve a balance between maximizing expected rewards and managing uncertainty. This leads to more robust and adaptive behaviors in dynamic and high-risk environments.

**Risk Measures:** In this paper, we assume the existence of a probability space $(\Omega, \mathcal{F}, \mathbb{P})$, where $\Omega$ represents the sample space, $\mathcal{F}$ a $\sigma$-field over $\Omega$, and $\mathbb{P}$ signifies a probability measure over $\mathcal{F}$. In this paper, we consider random variables defined over this probability space. Several risk measures are commonly used in decision-making, each with different assumptions and applications. These include the Markowitz Mean-Variance risk measure (Markowitz & Todd, 2000), which assumes that returns follow a normal distribution and balances expected return against variance. Another approach is the Wang transform function (Wang, 1996), which distorts the cumulative distribution function to model risk aversion or risk-seeking behavior. More widely used in risk-sensitive optimization are Value at Risk (`VaR`) (Rockafellar et al., 2000) and Conditional Value at Risk (CVaR) (Rockafellar et al., 2000; Rockafellar & Uryasev, 2002). $\texttt{CVaR}_\alpha$ and $\texttt{VaR}_\alpha$ of a random variable $\mathbf{X}$ at confidence level $\alpha \in [0, 1]$ are defined as follows[1]:

$$\texttt{CVaR}_\alpha(\mathbf{X}) = \mathbb{E}[\mathbf{X} \mid \mathbf{X} \geqslant \texttt{VaR}_\alpha(\mathbf{X})], \text{ where} \tag{1}$$

$$\texttt{VaR}_\alpha(\mathbf{X}) = \sup\{\beta \in \mathbb{R} \mid \mathbb{P}(\mathbf{X} \geqslant \beta) \geqslant \alpha\}. \tag{2}$$

While $\texttt{VaR}_\alpha$ identifies a gain threshold, $\texttt{CVaR}_\alpha$ provides a more comprehensive assessment by evaluating the expected gains in the best-case scenarios. This makes it particularly useful in risk-seeking decision-making, financial risk management, and safety-critical applications. Acceptance of the risk measures depends on the stability of their estimation procedures and the simplicity of optimization. The incorporation of these risk measures in policy optimization can be achieved either by modifying the objective function—where cumulative rewards are transformed non-linearly using a risk measure, commonly, $\texttt{CVaR}_\alpha$ (Chow et al., 2015; Tamar et al., 2015; Kashima, 2007; Keramati et al., 2020; Singh et al., 2020)—or by considering the risk measure as a constraint in the optimization setting, also commonly $\texttt{CVaR}_\alpha$ (Prashanth, 2014; Chow & Ghavamzadeh, 2014; Zhang et al., 2024; Ahmadi et al., 2021). $\texttt{CVaR}_\alpha$ is more widely accepted due to its coherent nature.

Entropic value at risk (`EVaR`) (Ahmadi-Javid, 2012) is a fairly new risk measure based on the exponential moment of gains, derived from the Chernoff bound, that provides a convex and coherent lower bound on gains, making it particularly useful for managing tail risk. It is defined as follows: For a random return $\mathbf{X} \in \mathbb{R}$, Entropic Value-at-Risk (`EVaR` - right tail) is defined with a confidence parameter $\alpha \in [0, 1)$, as

$$\texttt{EVaR}_\alpha[\mathbf{X}] = \inf_{\beta > 0} \left( \frac{1}{\beta} \log \frac{\mathbb{E}[e^{\beta \mathbf{X}}]}{\alpha} \right), \tag{3}$$

For the validity of the above definition, we assume that the moment-generating function $M_{\mathbf{X}}(\beta) = \mathbb{E}[e^{\beta \mathbf{X}}]$ exists for all $\beta \geqslant 0$. `EVaR` minimizes the worst-case bound on the right tail by optimizing over exponential

---

[1]Throughout, we adopt right-tail (gain) definitions of `VaR/CVaR/EVaR`; maximizing them is risk-seeking in the sense of preferentially targeting upper-tail returns.

moment bounds. It acts as a dual to the Legendre-Fenchel transform of the cumulant function $\log \mathbb{E}[e^{\beta \mathbf{X}}]$, ensuring a convex upper bound on extreme right-tail values. From large deviation theory, the cumulant function has a dual formulation: $\log \mathbb{E}[e^{\beta \mathbf{X}}] = \sup_{\xi \ll \mathbb{P}} \{\mathbb{E}_{\xi}[\mathbf{X}] - \mathrm{KL}(\xi \| \mathbb{P})\}$. Hence, we obtain the dual representation of $\mathrm{EVaR}_{\alpha}$ as follows:

$$\mathrm{EVaR}_{\alpha}[\mathbf{X}] = \inf_{\beta > 0} \sup_{\xi \ll P} \left( \frac{1}{\beta} \left( \mathbb{E}_{\xi}[\mathbf{X}] - \mathrm{KL}(\xi \| \mathbb{P}) - \log \alpha \right) \right)$$

$$= \sup_{\xi \ll P} \inf_{\beta > 0} \left( \mathbb{E}_{\xi}[\mathbf{X}] - \frac{1}{\beta} \mathrm{KL}(\xi \| \mathbb{P}) - \frac{1}{\beta} \log \alpha \right)$$

$$= \sup_{\xi \ll \mathbb{P}} \mathbb{E}_{\xi}[\mathbf{X}] \text{ with } \mathrm{KL}(\xi \| \mathbb{P}) \leqslant \log(1/\alpha).$$

From the above characterization, it easily follows that, $\mathbb{E}(\mathbf{X}) \leqslant \mathrm{EVaR}_{\alpha}(\mathbf{X}) \leqslant \mathrm{esssup}(\mathbf{X})$, where $\mathrm{esssup}(\mathbf{X}) = \inf\{x \in \mathbb{R} : \mathbb{P}(\mathbf{X} \leqslant x) = 1\}$ is the essential sup of $\mathbf{X}$. Note that $\mathrm{CVaR}_{\alpha}(\mathbf{X}) = \int \mathbf{X} d\mathbb{P}_{\mathrm{CVaR}}$, where the probability measure $\mathbb{P}_{\mathrm{CVaR}}(A) = \frac{1}{\alpha} \mathbb{P}(\{\mathbf{X} \geqslant \mathrm{VaR}_{\alpha}(\mathbf{X})\} \cap A))$, for Borel set $A$. Furthermore, $\mathrm{KL}(\mathbb{P}_{\mathrm{CVaR}}, \mathbb{P}) = \int \log \frac{1}{\alpha} d\mathbb{P}_{\mathrm{CVaR}} = \int \frac{\alpha}{\alpha} \log \frac{1}{\alpha} d\mathbb{P} = \log \frac{1}{\alpha}$. Also, $\mathbb{P}_{\mathrm{CVaR}} \ll \mathbb{P}$. This leads to the well-known ordering of risk measures:

$$\mathrm{VaR}_{\alpha}(\mathbf{X}) \leqslant \mathrm{CVaR}_{\alpha}(\mathbf{X}) \leqslant \mathrm{EVaR}_{\alpha}(\mathbf{X}).$$

This ordering indicates that $\mathrm{VaR}$ provides the least conservative risk assessment, while $\mathrm{EVaR}$ offers the most robust and conservative measure, with $\mathrm{CVaR}$ serving as an intermediate risk measure.

For continuous $r.v.s.$, $\mathrm{VaR}_{\alpha}$ estimate follows the asymptotic normality $\mathcal{N}\left(\mathrm{VaR}_{\alpha}, \frac{\alpha(1-\alpha)}{n f(\mathrm{VaR}_{\alpha})^2}\right)$ (Serfling, 2009), where $f$ is the PDF. This implies that as one approaches extreme tails ($\alpha$ small), the variance of the estimator becomes prohibitively large, making both $\mathrm{VaR}_{\alpha}$ and $\mathrm{CVaR}_{\alpha}$ difficult to estimate accurately. Indeed, $\mathrm{VaR}$ uses a hard indicator $\mathbf{1}\{\mathbf{X} \geqslant \beta\}$, while $\mathrm{CVaR}$ employs the same indicator to gate samples; both therefore discard $[(1 - \alpha)]\%$ of sample points. Consequently, their stochastic gradients exhibit order-of-magnitude higher variance. $\mathrm{EVaR}$ is a coherent risk measure as it satisfies subadditivity, positive homogeneity, monotonicity, and translation invariance. Its convex formulation ensures both robustness and tractability, making $\mathrm{EVaR}$ more suitable for optimal decision-making in risk-seeking settings. $\mathrm{EVaR}_{\alpha}$ avoids explicit tail estimation by leveraging the moment-generating function for its computation. It reweights the entire distribution with smooth exponential factors, yielding low-variance gradient estimates and a learning curve that remains both monotone and smooth.

**Problem Statement:** In this paper, our objective is to seek optimal decision-making under uncertainty, which is modeled as a Markov Decision Process (MDP). An MDP is defined by the tuple $(\mathcal{S}, \mathcal{A}, R, P, \gamma)$, where $\mathcal{S}$ and $\mathcal{A}$ are the finite state and action spaces, respectively. $R : \mathcal{S} \times \mathcal{A} \times \mathcal{S} \to \mathbb{R}$ is the reward function, where $R(s, a, s')$ represents the reward received for each state transition from $s \xrightarrow{a} s'$ taking action $a$. The transition probabilities are $\mathbb{P} : \mathcal{S} \times \mathcal{A} \to \Delta^S$, where $\Delta^S$ is the probability simplex in $\mathbb{R}^S$ and for a particular state-action pair, $\mathbb{P}(\cdot | s, a)$ is the transition probability, $\mathbb{P}_0(\cdot)$ is the initial state distribution, and $\gamma \in [0, 1)$ is the discount factor. For each state $s$, the set $\mathcal{A}(s)$ gives all available actions. A stationary policy $\pi(\cdot | s)$ is a probability distribution over actions that depends on the current state $s$. Here we consider parameterized stochastic policies, which are parameterized by a $p$-dimensional vector $\theta$, which means that the policy space can be written as $\Pi_{\Theta} = \{\pi_{\theta}(\cdot | s), s \in \mathcal{S}, \theta \in \Theta \subseteq \mathbb{R}^p\}$. In this paper, we consider the following risk-seeking control problem:

$$\theta^* = \arg\max_{\theta \in \Theta} J_{\mathrm{EVaR}}(\theta) = \mathrm{EVaR}_{\alpha}[R(\tau)]$$

$$= \arg\max_{\theta \in \Theta} \inf_{\beta > 0} \frac{1}{\beta} \log \frac{\mathbb{E}_{\tau \sim \pi_{\theta}}\left[e^{\beta R(\tau)}\right]}{\alpha}, \tag{4}$$

where $R(\tau) = \sum_{t=0}^{T-1} \gamma^t R(\mathbf{s}_t, \mathbf{a}_t, \mathbf{s}_{t+1})$, with $\mathbf{s}_0 \sim \mathbb{P}_0, \mathbf{a}_t \sim \pi_{\theta}(\cdot | \mathbf{s}_t), \mathbf{s}_{t+1} \sim \mathbb{P}(\cdot | \mathbf{s}_t, \mathbf{a}_t)$, and $T \in \mathbb{N}$. Note that $J_{\mathrm{EVaR}}(\theta)$ exists for $\forall \theta$, since $R(\cdot)$ is bounded.

The entropic value-at-risk is inherently connected to *exponential tilting*, a technique that modifies the probability distribution to emphasize higher-risk, high-reward outcomes. To illustrate this connection,

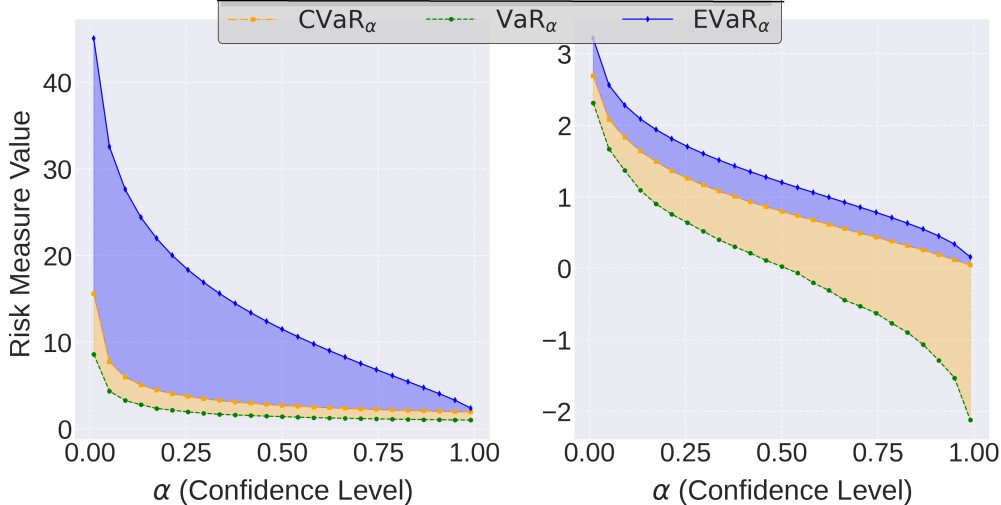

Figure 1: Comparison of `VaR, CVaR`, and `EVaR` computed for for heavy-tailed (Pareto) and light-tailed (Normal) distributions, showing `EVaR`'s stronger tail-risk sensitivity. The canonical ordering $\texttt{VaR}_\alpha \leqslant \texttt{CVaR}_\alpha \leqslant \texttt{EVaR}_\alpha$ holds, with heavier tails accentuating the `EVaR` gap—providing a sharper tail-aware nature that prioritizes trajectories with sparse, high returns.

consider the exponentially tilted probability measure $\mathbb{Q}_\beta$, defined as:

$$\frac{d\mathbb{Q}_\beta}{d\mathbb{P}_{\pi_\theta}} = \frac{e^{\beta r}}{\mathbb{E}_{\tau \sim \pi_\theta}[e^{\beta R(\tau)}]}, \tag{5}$$

This transformation effectively reweights the original probability distribution $\mathbb{P}_{\pi_\theta}$, increasing the probability of trajectories with higher cumulative rewards $R(\tau)$. As a result, the modified measure $\mathbb{Q}_\beta$ gives greater importance to risk-seeking outcomes, ensuring that extreme rewards are more heavily considered. Using this tilted distribution, we obtain the following relationship between the `KL` divergence and the `EVaR` objective:

$$\texttt{KL}(\mathbb{Q}_\beta \| \mathbb{P}_{\pi_\theta}) = \beta \mathbb{E}_{\mathbb{Q}_\beta}[\mathbf{R}] - \log \mathbb{E}_{\tau \sim \pi_\theta}[e^{\beta R(\tau)}] \text{ and}$$

$$J_{\texttt{EVaR}}(\theta) = \inf_{\beta > 0} \left( \mathbb{E}_{\mathbb{Q}_\beta}[\mathbf{R}] - \frac{1}{\beta}\texttt{KL}(\mathbb{Q}_\beta \| \mathbb{P}_{\pi_\theta}) - \frac{1}{\beta}\log\alpha \right)$$

$$= \sup_{\mathbb{Q} \in \mathcal{M}_\theta} \left\{ \mathbb{E}_{\mathbb{Q}}[\mathbf{R}] \mid \texttt{KL}(\mathbb{Q} \| \mathbb{P}_{\pi_\theta}) \leqslant \log\frac{1}{\alpha} \right\}, \text{ where } \mathcal{M}_\theta = \left\{ \mathbb{Q}_\beta \mid \exists \beta > 0, \ \frac{d\mathbb{Q}_\beta}{d\mathbb{P}_{\pi_\theta}} = \frac{e^{\beta r}}{\mathbb{E}_{\tau \sim \pi_\theta}[e^{\beta R(\tau)}]} \right\}.$$

This formulation highlights that `EVaR` selects the extreme expectation over a set of tilted distributions, each constrained by a KL-divergence bound (KL-ball of radius $\log(1/\alpha)$)). In other words, it finds the most extreme risk-seeking expectation while ensuring the alternative probability distribution remains within a reasonable divergence from the original measure, thereby maintaining robustness in decision-making under uncertainty. The `EVaR` objective does not merely maximize expected returns but incorporates a risk-seeking adjustment that prioritizes high-reward yet riskier trajectories. Further, the KL-divergence constraint prevents excessive deviation from the original probability distribution, ensuring a balanced trade-off between exploration and risk seeking.

**Related Literature:** The field of risk-sensitive control and reinforcement learning has been well studied, beginning with the seminal work of (Howard & Matheson, 1972a) which introduced the application of an exponential utility function to rewards. A substantial body of research has concentrated on risk-sensitive control in continuous-time, finite-horizon settings, particularly for problems with known transition kernels (Fleming & McEneaney, 1995; Whittle, 1990; Coraluppi & Marcus, 1999; Koenig & Simmons, 1994). Early developments extended optimal control techniques to reinforcement learning (Littman & Szepesvári, 1996; Borkar, 2001; 2002; 2010; Mihatsch & Neuneier, 2002; Heger, 1994). The use of exponential ergodic

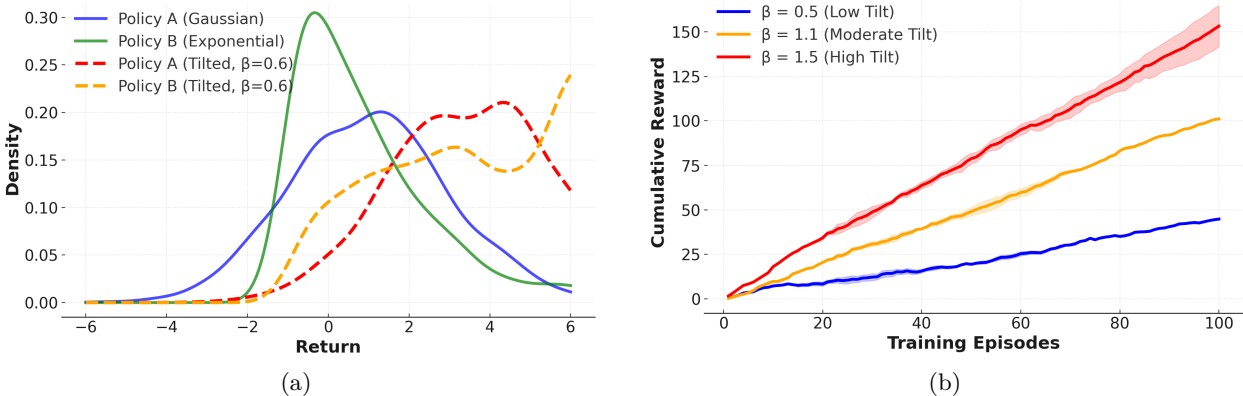

Figure 2: **(a)** Plot shows exponentially tilted distributions for two RL policies with moderate risk sensitivity. Policy A remains stable, while Policy B shifts significantly, emphasizing its long tail. Tilting reweights probability mass toward high-reward outcomes, enhancing risk-seeking learning. (b) Low $\beta$ encourages exploration with stable learning, moderate $\beta$ balances exploration and risk sensitivity, while high $\beta$ prioritizes high-reward strategies but introduces higher variability.

performance in discrete time control (Di Masi & Stettner, 1999; 2007) was further generalized to address risk-sensitive average cost criteria (Cavazos-Cadena & Hernandez-Hernandez, 2011). In reinforcement learning, risk-sensitive methods have been adapted for model-free settings with unknown transition dynamics, employing techniques such as relative entropy stochastic search and Q-learning (Borkar & Meyn, 2002; Osogami, 2012). Recent advances in risk-sensitive reinforcement learning integrate conditional value-at-risk (CVaR) objectives for robust policy optimization (Chow et al., 2015; Chow & Ghavamzadeh, 2014), quantile temporal-difference learning to capture return distributions more accurately (Rowland et al., 2024), leverage sample-based dynamic programming techniques that augment Bayes-adaptive MDPs with CVaR constraints to derive risk-averse policies (Rigter et al., 2021), and incorporate coherent risk measures alongside non-linear function approximation (Lam et al., 2022). `EVaR` policy optimization in RL is a relatively new and less explored area. (Ni & Lai, 2022) proposed a trajectory-based policy gradient method to optimize `EVaR`-induced risk-sensitive criteria, (Dixit et al., 2021) developed nested `EVaR`-constrained models, and (Hau et al., 2023) introduced a dynamic programming approach for `EVaR` objectives by formulating `EVaR`-based Bellman equations under known transition dynamics.

**Our Contribution:** In this paper, we provide an online multi-time scale stochastic approximation algorithm to estimate the `EVaR` of the reward distribution and also seek the optimal `EVaR` policy in the context of model-free risk-seeking reinforcement learning setting.

## 2 Proposed Method

In this section, we recast the control objective (4) into two tightly coupled sub-problems. *Prediction* demands, for any fixed policy parameter $\theta$, an online, sample-efficient estimate of the entropic value-at-risk $J_{\text{EVaR}}(\theta)$ and *optimization* then needs a dependable ascent direction built from those estimates to steer $\theta$ towards an `EVaR`-optimal policy. We tackle both challenges simultaneously within a multi-timescale stochastic-approximation framework: a fast inner loop continually calibrates the `EVaR` estimate, while a slower outer loop performs simultaneous-perturbation-based gradient ascent, which seeks the risk-seeking solution.

### 2.1 `EVaR` Estimation

In this section, we propose an online, multi-timescale approach to estimate `EVaR`. From Eq.(4) we have

$$J_{\text{EVaR}}(\theta) = \inf_{\beta>0} \frac{1}{\beta} \big( \log \mathbb{E}_{\tau \sim \pi_\theta} \left[ e^{\beta \cdot R(\tau)} \right] - \log \alpha \big) \tag{6}$$

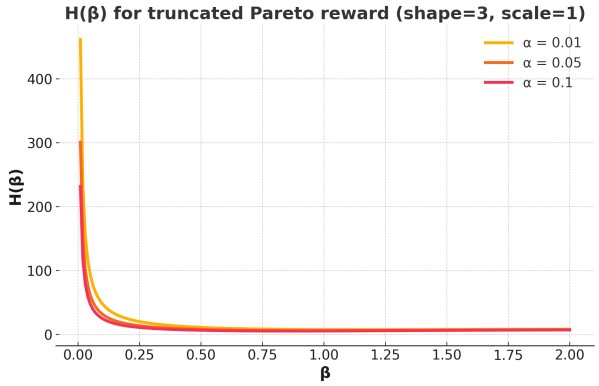 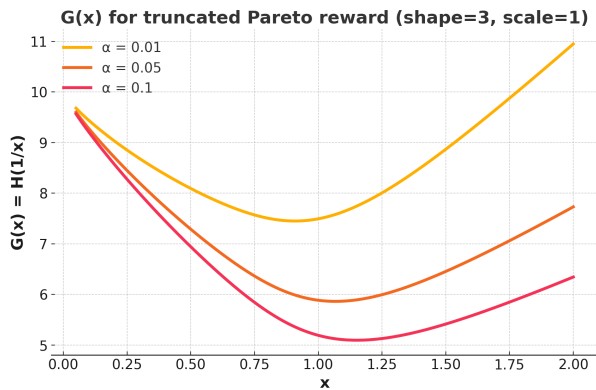

(a) $H(\beta)$ *vs.* $\beta$ for different confidence levels $\alpha$.    (b) $G(x) = H(1/x)$ *vs.* $x$ for different confidence levels $\alpha$.

Figure 3: Dependence of the `EVaR` curves on the confidence level $\alpha$ for a truncated Pareto reward distribution (`shape`$= 3$, `scale`$= 1$). **(a)** The function $H(\beta)$ diverges more steeply as $\beta \to 0$ for lower $\alpha$, indicating stronger tail-risk sensitivity. **(b)** The convex transformation $G(x) = H(1/x)$ attains its minimizer at larger $x$ as $\alpha$ decreases, reflecting a more conservative risk-seeking nature.

Let $H(\beta) = \beta^{-1}\big(\log \mathbb{E}_{\tau \sim \pi_\theta}\big[e^{\beta \cdot R(\tau)}\big] - \log \alpha\big)$, where $\beta > 0$. We establish the following result for $H$.

**Proposition 1.** *The function $G(\beta) = H\left(\frac{1}{\beta}\right)$ is convex in $\beta$ over $\beta > 0$.*

*Proof.* Given, $H\left(\frac{1}{\beta}\right) = \beta\left(\log \mathbb{E}\left[e^{\frac{R(\tau)}{\beta}}\right] - \log \alpha\right)$. To prove convexity, for any $\lambda \in [0,1]$ and $\beta_1, \beta_2 > 0$, let $\beta = \lambda\beta_1 + (1-\lambda)\beta_2$. Define normalized weights:

$$\mu = \frac{\lambda\beta_1}{\beta}, \quad \nu = \frac{(1-\lambda)\beta_2}{\beta}, \quad \mu + \nu = 1.$$

By Hölder's inequality:

$$\mathbb{E}\left[e^{\frac{R(\tau)}{\beta}}\right] \leqslant \mathbb{E}\left[e^{\frac{R(\tau)}{\beta_1}}\right]^\mu \mathbb{E}\left[e^{\frac{R(\tau)}{\beta_2}}\right]^\nu.$$

Taking logarithms:

$$\log \mathbb{E}\left[e^{\frac{R(\tau)}{\beta}}\right] \leqslant \mu \log \mathbb{E}\left[e^{\frac{R(\tau)}{\beta_1}}\right] + \nu \log \mathbb{E}\left[e^{\frac{R(\tau)}{\beta_2}}\right].$$

Multiply by $\beta$ and subtract $\beta \log \alpha$:

$$H\left(\frac{1}{\beta}\right) \leqslant \lambda H\left(\frac{1}{\beta_1}\right) + (1-\lambda)H\left(\frac{1}{\beta_2}\right).$$

Thus, $H\left(\frac{1}{\beta}\right)$ is convex in $\beta$. $\qquad\square$

The original function $H(\beta)$ exhibits mixed curvature due to the $\beta$-scaling in the denominator and the exponential term. However, the transformation $\beta \to \frac{1}{\beta}$, reparameterizes the function, flipping the curvature to enforce convexity. This property is critical because it simplifies the optimization landscape, making it much easier and more reliable to find the unique global optimum.

**Corollary 1.**

$$\min_{\beta > 0} H(\beta) = \min_{x > 0} G(x),$$

*Moreover, the minimizers satisfy $\beta^* = 1/x^*$.*

*Proof.* Define the change of variable $x = \frac{1}{\beta}$. Then $G(x) = H\left(\frac{1}{x}\right)$. Let $\beta^*$ be a global minimizer of $H(\beta)$ and $x^* = 1/\beta^*$. Then $G(x^*) = H\left(\frac{1}{x^*}\right) = H(\beta^*)$. Since, $\beta^*$ minimizes $H$, we have for any $x > 0$, $H(\beta^*) \leqslant H\left(\frac{1}{x}\right) = G(x)$. Hence, $G(x^*) = H(\beta^*) \leqslant G(x)$, $\forall x > 0$. Thus $x^*$ is a global minimizer of $G$, and

$$\min_{x>0} G(x) \leqslant G(x^*) = H(\beta^*) = \min_{\beta>0} H(\beta).$$

Conversely, suppose $x^*$ minimizes $G(x)$. Define $\beta^* = 1/x^*$. By an identical argument,

$$\min_{\beta>0} H(\beta) \leqslant H(\beta^*) = G(x^*) = \min_{x>0} G(x).$$

Putting these two inequalities together,

$$\min_{\beta>0} H(\beta) = \min_{x>0} G(x).$$

Furthermore, the minimizers match by $\beta^* = 1/x^*$. $\qquad\square$

To establish the convergence and stability of our approach, we require certain regularity conditions on the risk-seeking objective and the policy-induced reward distribution. These conditions ensure that the `EVaR` optimization problem remains well-posed and that the gradient estimates concentrate sufficiently around their expected values.

**Assumption 1.** *The variable $x = 1/\beta$ is restricted to a compact set $I = [x_{\min}, x_{\max}]$ with $0 < x_{\min} < x_{\max} < \infty$.*

**Assumption 2.** *There exists $\sigma > 0$ such that for all $x \in I$,*

$$\mathbf{Var}_{\mathbb{Q}_x}(R(\tau)) = \mathbb{E}_{\mathbb{Q}_x}[R(\tau)^2] - \left(\mathbb{E}_{\mathbb{Q}_x}[R(\tau)]\right)^2 \geqslant \sigma,$$

*where the exponentially tilted probability measure $\mathbb{Q}_x$ is defined as $\dfrac{d\mathbb{Q}_x}{d\mathbb{P}_{\pi_\theta}} = \dfrac{e^{r/x}}{\mathbb{E}_{\tau \sim \pi_\theta}\left[e^{R(\tau)/x}\right]}$.*

This assumption ensures that the reward distribution under the exponentially tilted measure $\mathbb{Q}_x$ always retains a minimum level of variability. Specifically, it guarantees that the variance of rewards does not collapse to zero for any risk sensitivity parameter $x$. This is critical because the policy search requires sufficient variability in rewards to effectively explore the policy space and avoid degenerate solutions.

**Lemma 1.** *Under Assumptions 1 and 2:*

1. *The variance $\mathbf{Var}_{\mathbb{Q}_x}(R(\tau))$ is continuous in $x$ over $I = [x_{\min}, x_{\max}]$.*

2. *$\exists \bar{\sigma} > 0$ such that $\mathbf{Var}_{\mathbb{Q}_x}(R(\tau)) \geqslant \bar{\sigma}$, $\forall x \in I$.*

*Proof.* Let $Z(x) = \mathbb{E}_\tau[e^{R(\tau)/x}]$. The tilted expectation $\mathbb{E}_{\mathbb{Q}_x}[R(\tau)]$ and $\mathbb{E}_{\mathbb{Q}_x}[R(\tau)^2]$ are given by:

$$\mathbb{E}_{\mathbb{Q}_x}[R(\tau)] = \frac{\mathbb{E}_\tau\left[R(\tau)e^{\frac{R(\tau)}{x}}\right]}{Z(x)}, \quad \mathbb{E}_{\mathbb{Q}_x}[R(\tau)^2] = \frac{\mathbb{E}_\tau\left[R(\tau)^2 e^{\frac{R(\tau)}{x}}\right]}{Z(x)}.$$

By ($|R(\tau)| \leqslant \frac{R_\infty}{1-\gamma}$,) and Assumption 1 ($x \in I$), the terms $R(\tau)e^{R(\tau)/x}$ and $R(\tau)^2 e^{R(\tau)/x}$ are bounded. By the Dominated Convergence Theorem (DCT), $\mathbb{E}_\tau[R(\tau)e^{R(\tau)/x}]$, $\mathbb{E}_\tau[R(\tau)^2 e^{R(\tau)/x}]$, and $Z(x)$ are continuous in $x$. Since $Z(x) \geqslant e^{\frac{-R_\infty}{(1-\gamma)x_{\max}}} > 0$, the ratios $\mathbb{E}_{\mathbb{Q}_x}[R(\tau)]$ and $\mathbb{E}_{\mathbb{Q}_x}[R(\tau)^2]$ are continuous. Thus, $\mathbf{Var}_{\mathbb{Q}_x}(R(\tau)) = \mathbb{E}_{\mathbb{Q}_x}[R(\tau)^2] - (\mathbb{E}_{\mathbb{Q}_x}[R(\tau)])^2$ is continuous. By Assumption 2, $\mathbf{Var}_{\mathbb{Q}_x}(R(\tau)) \geqslant \sigma > 0$ for all $x \in I$. Continuity of $\mathbf{Var}_{\mathbb{Q}_x}(R(\tau))$ and compactness of $I$ imply $\mathbf{Var}_{\mathbb{Q}_x}(R(\tau))$ attains its minimum on $I$. Let $\bar{\sigma} = \min_{x \in I} \mathbf{Var}_{\mathbb{Q}_x}(R(\tau))$. By Assumption 2, $\bar{\sigma} > 0$. $\qquad\square$

**Theorem 1.** *Under Assumptions 1 and 2, the function*

$$G(x) = x\left(\log \mathbb{E}\left[e^{R(\tau)/x}\right] - \log \alpha\right)$$

*is m-strongly convex on $I = [x_{\min}, x_{\max}]$ with modulus $m = \frac{\bar{\sigma}}{x_{\max}^3}$.*

*Proof.* One can easily find the second-order derivative of $G(x)$ as follows:

$$G''(x) = \frac{\mathbb{E}_{\mathbb{Q}_x}[R(\tau)^2] - (\mathbb{E}_{\mathbb{Q}_x}[R(\tau)])^2}{x^3} = \frac{\text{Var}_{\mathbb{Q}_x}(R(\tau))}{x^3}.$$

By Lemma 1, $\text{Var}_{\mathbb{Q}_x}(R(\tau)) \geqslant \bar{\sigma} > 0$. Since $x \leqslant x_{\max}$,

$$G''(x) \geqslant \frac{\bar{\sigma}}{x_{\max}^3} = m > 0.$$

Thus, $G(x)$ is strongly convex on $I$. □

We now propose a multi-timescale framework to estimate $J_{\text{EVaR}}$ by solving the above optimization problem via finding the roots of the derivative $G'$. For this purpose, we derive a closed-form expression for $G$ with respect to $x$ as follows:

$$\begin{aligned} G'(x) &= \frac{d}{dx}\left[ x\left( \log \mathbb{E}_\tau\left[ e^{R(\tau)/x} \right] - \log \alpha \right) \right] \\ &= \log \frac{\mathbb{E}_\tau\left( e^{R(\tau)/x} \right)}{\alpha} - \frac{1}{x} \frac{\mathbb{E}_\tau[R(\tau)\, e^{R(\tau)/x}]}{\mathbb{E}_\tau[e^{R(\tau)/x}]}. \end{aligned} \tag{7}$$

The interchange of $\mathbb{E}_\tau[\cdot]$ and $\frac{d}{dx}$ in the above equality is possible through the bounded convergence theorem. In practice, the exact expectations $\mathbb{E}_\tau[e^{R(\tau)/x}]$ and $\mathbb{E}_\tau[R(\tau)e^{R(\tau)/x}]$ are often intractable to compute directly. To address this, we replace these expectations with online average estimates, computed from observed samples over time. Consequently, we arrive at a two-timescale stochastic approximation algorithm for finding $x^* = \arg\min_{x>0} G(x)$. In this framework, one timescale is used to update the estimate of $G'$, while the other handles the solution update in the direction of the estimate. This separation enables more efficient and stable convergence of the gradient estimation process.

$$\begin{aligned} \vartheta_{t+1} &= \vartheta_t + \delta_t \left( e^{R(\tau_{t+1})/x_t} - \vartheta_t \right), && \text{where } \tau_{t+1} \sim \pi_\theta \text{ and the step-size } \delta_t \in (0,1) \\ \omega_{t+1} &= \omega_t + \delta_t e^{R(\tau_{t+1})/x_t} \left( R(\tau_{t+1}) - x_t\omega_t \right),. \end{aligned} \tag{8}$$

The above single timescale stochastic recursions estimate the expectations $\mathbb{E}_\tau[e^{R(\tau)/x}]$ and $\mathbb{E}_\tau[R(\tau)e^{R(\tau)/x}]/x\mathbb{E}_\tau[e^{R(\tau)/x}]$ which amount to estimating $G'$. Now we can apply these estimates to seek $x^*$ by calibrating the iterates in the negative direction of the derivative estimate as follows:

$$x_{t+1} = x_t - \xi_t \left( \log \frac{\vartheta_t}{\alpha} - \omega_t \right), \quad \text{where } \xi_t \in (0,1) \text{ is the step-size.} \tag{9}$$

Note that the above recursion is maintained at a slower time scale relative to the recursions Eq.(8). This is required because one needs a good estimate of $G'$ to efficiently calibrate $x_t$. This develops a bidirectional coupling where the recursion Eq.(9) can be considered quasistatic *w.r.t.* recursion Eq.(8). This can be illustrated as follows:

$$x_{t+1} = x_t - \delta_t \frac{\xi_t}{\delta_t} \left( \log \frac{\vartheta_t}{\alpha} - \omega_t \right) \tag{10}$$

By stacking Eqs.(8), and (10) in vector notation, we obtain the following

$$\begin{bmatrix} \vartheta_{t+1} \\ \omega_{t+1} \\ x_{t+1} \end{bmatrix} = \begin{bmatrix} \vartheta_t \\ \omega_t \\ x_t \end{bmatrix} + \delta_t \begin{bmatrix} e^{R(\tau_{t+1})/x_t} - \vartheta_t \\ e^{R(\tau_{t+1})/x_t} \left( R(\tau_{t+1}) - x_t\omega_t \right) \\ \frac{-\xi_t}{\delta_t} \left( \log \frac{\vartheta_t}{\alpha} - \omega_t \right) \end{bmatrix}$$

If we let $\lim_{t\to\infty} \frac{\xi_t}{\delta_t} \to 0$, then from the above equation, one can find that while $x_t$ is quasi-static, the estimates $\vartheta_t$ and $\omega_t$ get nearly equilibrated to their mean-field limits $\vartheta^*(x_t)$ and $\omega^*(x_t)$ respectively. This quasi-static equilibrium eliminates bias in gradient estimates.

**Assumption 3.** *We assume that the step-size schedules $\{\delta_t\}_{t\in\mathbb{N}}$ and $\{\xi_t\}_{t\in\mathbb{N}}$ are real-valued, positive, deterministic, pre-determined sequences, and they satisfy*

$$\sum_{t\in\mathbb{N}}\left(\delta_t^2 + \xi_t^2\right) < \infty, \quad \sum_{t\in\mathbb{N}}\delta_t = \sum_{t\in\mathbb{N}}\xi_t = \infty, \quad \lim_{t\to\infty}\frac{\xi_t}{\delta_t} = 0.$$

Examples of such step sizes can be $\xi_t = \frac{1}{t}, \delta_t = \frac{1}{1+t\log t}$ or $\xi_t = \frac{1}{t^{2/3}}, \delta_i = 1/t$. The above assumptions are required as they are critical technical requirements for ensuring almost sure convergence. The first condition ensures that they decay fast enough. The second condition ensures that updates happen throughout the entire time continuum. The final condition (time scale separation) ensures that the ratio of the step sizes must approach zero, ensuring the updates for $x_t$ occur on a slower timescale compared to $\vartheta_t$ and $\omega_t$.

The following theorem characterizes the limiting behavior of the proposed empirical `EVaR` estimation recursion under the stated regularity assumptions and step-size conditions:

**Theorem 2** (Convergence of $J_{\text{EVaR}}$ under static $\theta$). *Given policy $\pi_\theta$, under Assumptions 1–3, the coupled stochastic recursions Eq.(8) & Eq.(9) constitute a stochastic Euler discretization of the ODE system:*

$$\dot{\vartheta} = \mathbb{E}_{\tau\sim\pi_\theta}\left[e^{R(\tau)/x}\right] - \vartheta,$$

$$\dot{\omega} = \frac{\mathbb{E}_{\tau\sim\pi_\theta}\left[R(\tau)e^{R(\tau)/x}\right]}{x\mathbb{E}_{\tau\sim\pi_\theta}\left[e^{R(\tau)/x}\right]} - \omega,$$

$$\dot{x} = -\underbrace{\left(\log\frac{\mathbb{E}_{\tau\sim\pi_\theta}\left[e^{R(\tau)/x}\right]}{\alpha} - \frac{\mathbb{E}_{\tau\sim\pi_\theta}\left[R(\tau)e^{R(\tau)/x}\right]}{x\mathbb{E}_{\tau\sim\pi_\theta}\left[e^{R(\tau)/x}\right]}\right)}_{-G'(x)},$$

*Furthermore, the iterates satisfy:*

$$(\vartheta_t, \omega_t, x_t) \xrightarrow{\text{a.s.}} (\vartheta^*(\theta), \omega^*(\theta), x^*(\theta)),$$

*with equilibrium $(\vartheta^*(\theta), \omega^*(\theta), x^*(\theta))$ characterized by:*

$$\vartheta^*(\theta) = \mathbb{E}_{\tau\sim\pi_\theta}\left[e^{R(\tau)/x^*(\theta)}\right], \quad \omega^*(\theta) = \frac{\mathbb{E}_{\tau\sim\pi_\theta}\left[R(\tau)e^{R(\tau)/x^*(\theta)}\right]}{x^*(\theta)\vartheta^*(\theta)} \quad and,$$

$$x^*(\theta) \in \left\{x \in \mathbb{R}^+ \,|\, G'(x) = 0\right\}.$$

*Proof.* (Sketch) We analyze the coupled stochastic recursions for $(\vartheta_t, \omega_t)$ on the fast timescale and $x_t$ on the slower timescale.

*Fast recursions.* The updates admit the stochastic approximation form

$$\vartheta_{t+1} = \vartheta_t + \delta_t\left(h_1(\vartheta_t) + \mathbb{M}_{t+1}\right), \qquad \omega_{t+1} = \omega_t + \delta_t\left(h^\omega(\omega_t) + \mathbb{M}_{t+1}^\omega\right),$$

with drifts

$$h_1(\vartheta) = \mathbb{E}\left[e^{R(\tau)/x}\right] - \vartheta, \qquad h^\omega(\omega) = \mathbb{E}\left[R(\tau)e^{R(\tau)/x}\right] - \omega\,\mathbb{E}\left[e^{R(\tau)/x}\right],$$

and martingale-difference noise terms $\{\mathbb{M}_t\}, \{\mathbb{M}_t^\omega\}$ adapted to $\{\mathcal{F}_t\}$. Both drifts are Lipschitz, and stability of the iterates follows from the Borkar–Meyn theorem. The limiting ODEs are linear,

$$\dot{\vartheta} = \mathbb{E}[e^{R(\tau)/x}] - \vartheta, \qquad \dot{\omega} = \mathbb{E}[R(\tau)e^{R(\tau)/x}] - \omega\,\mathbb{E}[e^{R(\tau)/x}],$$

which converge globally to the unique equilibria

$$\vartheta^*(x, \theta) = \mathbb{E}_{\tau\sim\pi_\theta}[e^{R(\tau)/x}], \qquad \omega^*(x, \theta) = \frac{\mathbb{E}_{\tau\sim\pi_\theta}[R(\tau)e^{R(\tau)/x}]}{\mathbb{E}_{\tau\sim\pi_\theta}[e^{R(\tau)/x}]}.$$

*Slow recursion.* On the slower timescale, the update for $x_t$ tracks the ODE

$$\dot{x} = -G'(x) = -\left(\log\frac{\vartheta^*(x,\theta)}{\alpha} - \omega^*(x,\theta)\right).$$

By Theorem 1, $G(x)$ is strongly convex and hence has a unique minimizer $x^*(\theta)$. Using $G(x)$ as a Lyapunov function yields

$$\frac{d}{dt}G(x_t) = -(G'(x_t))^2 \leqslant 0,$$

with equality only at $x^*(\theta)$. Thus $x_t \to x^*(\theta)$ almost surely.

*Joint convergence.* Timescale separation ensures $(\vartheta_t, \omega_t)$ track $(\vartheta^*(x_t,\theta), \omega^*(x_t,\theta))$ as $x_t$ evolves. Consequently,

$$(\vartheta_t, \omega_t, x_t) \to \left(\vartheta^*(x^*(\theta),\theta), \omega^*(x^*(\theta),\theta), x^*(\theta)\right) \quad \text{almost surely.}$$

The argument is a standard multi-timescale stochastic approximation. On the fast timescale, the auxiliary variables $\vartheta_t$ and $\omega_t$ rapidly converge to the exponential moment and its tilted expectation, namely $\vartheta^*(x,\theta)$ and $\omega^*(x,\theta)$. On the slower timescale, $x_t$ evolves according to the gradient flow of the `EVaR` objective $G(x)$, which is strongly convex and admits a unique minimizer $x^*(\theta)$. Because the fast variables equilibrate much faster than $x_t$ changes, the overall system converges to the point

$$(\vartheta^*(x^*(\theta),\theta), \omega^*(x^*(\theta),\theta), x^*(\theta)),$$

ensuring almost sure convergence of the algorithm.

For the detailed proof, please refer to Appendix A.1 □

**Corollary 2.** *For a given policy $\pi_\theta$, as $t \to \infty$, the iterates $\{x_t\}$ satisfy the following*

$$\lim_{t\to\infty} G(x_t) = J_{EVaR}(\theta). \tag{11}$$

*Proof.* Follows from the continuity of $G$ and Theorem 2. □

The asymptotic convergence guarantees established above do not quantify the rate at which the estimators $(\vartheta_t, \omega_t, x_t)$ converge to their equilibrium values. To characterize the finite-time behavior of the estimators and provide explicit error bounds, we establish a non-asymptotic convergence rate here. The following theorem quantifies the mean squared error (MSE) for each estimator.

**Theorem 3.** *Let $\delta_t = \frac{c}{t^r}$, $\xi_t = \frac{d}{t^b}$, $b, c \in (\frac{1}{2}, 1)$ and $c, d > 0$. Then:*

1. **Convergence of $\vartheta_t$:** *For $c > \frac{r}{4}$, Then for any fixed $x > 0$,*

$$\mathbb{E}\left[|\vartheta_t - \vartheta^*(x,\theta)|^2\right] \leqslant \frac{K_1}{t^r}, \quad \text{with } K_1 > 0, \forall t \geqslant 1 \text{ and } \vartheta^*(x,\theta) = \mathbb{E}_{\tau\sim\pi_\theta}\left[e^{R(\tau)/x}\right]$$

2. **Convergence of $\omega_t$:** *For $c > \frac{r}{2\mathbb{E}_{\tau\sim\pi_\theta}\left[xe^{R(\tau)/x}\right]}$. Then for any fixed $x > 0$,*

$$\mathbb{E}\left[|\omega_t - \omega^*(x,\theta)|^2\right] \leqslant \frac{K_2}{t^r}, \quad \text{for some } K_2 > 0, \forall t \geqslant 1 \text{ and } \omega^*(x,\theta) = \frac{\mathbb{E}_{\tau\sim\pi_\theta}\left[R(\tau)e^{R(\tau)/x}\right]}{x\,\vartheta^*(x,\theta)}.$$

3. **Convergence of $x_t$:** *Let $d = \frac{x_{\max}^3(1-b)}{2\bar{\sigma}} \cdot b$ and $c > \max\left\{\frac{r}{4}, \frac{r}{2\mathbb{E}_{\tau\sim\pi_\theta}\left[xe^{R(\tau)/x}\right]}\right\}$. Under the Assumptions 1 and 3, we have*

$$\mathbb{E}\left[|x_t - x^*|^2\right] \leqslant \frac{K_3}{t^b},$$

*where $x^* = \arg\min_x G(x)$, and the constant $K_3 > 0$.*

*Proof.* (Sketch) The estimator $\vartheta_t$ converges to $\vartheta^*(x,\theta) = \mathbb{E}_\tau[e^{R(\tau)/x}]$ with mean-square error $O(t^{-r})$ under polynomial step-sizes $\delta_t = c/t^r$ as shown by Lemma 6. Similarly, Lemma 7 proves that $\omega_t$ converges to $\omega^*(x,\theta) = \frac{\mathbb{E}_\tau[R(\tau)e^{R(\tau)/x}]}{\mathbb{E}_\tau[e^{R(\tau)/x}]}$ with the same $O(t^{-r})$ rate. Now, we proceed with the finite-time analysis as follows.

*(Slow recursion).* On the slower timescale, the update for $x_t$ is

$$x_{t+1} = x_t - \xi_t\big(G'(x_t) + \eta_t\big),$$

where the gradient error $\eta_t = \log(\vartheta_t/\vartheta^*) - (\omega_t - \omega^*)$ inherits its variance from the fast recursions. By Lemmas 6–7,

$$\mathbb{E}[\|\eta_t\|^2] = O(t^{-r}).$$

Defining $V_t = |x_t - x^*|^2$ and using strong convexity of $G$ with parameter $\mu = \bar{\sigma}/x_{\max}^3$, one obtains the recursion

$$\mathbb{E}[V_{t+1}] \leqslant (1 - 2\mu\xi_t)\mathbb{E}[V_t] + \xi_t^2\big(L^2 + O(t^{-r})\big).$$

*(Rate bound).* With $\xi_t = d/t^b$ for $b \in (\frac{1}{2}, 1)$, the negative drift dominates the error terms. A comparison-sequence argument shows

$$\mathbb{E}[V_t] = O(t^{-b}),$$

yielding the claimed rate

$$\mathbb{E}\big[(x_t - x^*)^2\big] \leqslant \frac{K_3}{t^b}, \qquad K_3 > 0.$$

The fast recursions (Lemmas 6–7) guarantee that $\vartheta_t$ and $\omega_t$ track their population limits with vanishing error $O(t^{-r})$. The slow recursion for $x_t$ is then essentially stochastic gradient descent on the strongly convex EVaR objective $G(x)$, perturbed by this decaying error. Because the inner estimates improve sufficiently quickly, the outer descent achieves a finite-time rate $O(t^{-b})$. Intuitively, the method balances *accuracy of inner exponential-statistic estimation* with *progress of the outer descent*, ensuring polynomial sample complexity guarantees.

For the detailed proof, please refer to Appendix A.2. □

The above result captures the precise bias–variance trade-off dictated by the multi-timescale approach. The fast coordinates $(\vartheta_t, \omega_t)$ must enter an $O(t^{-r})$-MSE band rapidly enough that their residual bias is negligible for the outer recursion, whereas the slow coordinate $x_t$ must move with a step–size exponent $b > r$ so the outer estimate appear quasi–stationary. Selecting $b \leqslant r$ destroys this quasi–static regime, and would allow the perturbation to dominate, inflating the outer–loop variance and impeding convergence. Further, the choice of $d$ balances the strong convexity constant $\bar{\sigma}$ and the time-scale separation $b$, while the choice of $c$ depends on the exponentially tilted reward distribution, which scales with reward variability.

## 2.2 EVaR Optimization

The gradient estimation of the EVaR objective with respect to policy parameters $\theta$ employs the simultaneous perturbation stochastic approximation method (Spall, 1992), a computationally efficient technique for high-dimensional optimization. This approach perturbs all parameters simultaneously using a randomized direction vector $\Delta_t \in \mathbb{R}^p$ circumventing the $O(p)$ computational complexity of finite-difference methods. The gradient estimate is constructed as follows:

$$\widehat{\nabla_\theta J}_{\text{EVaR}}(\theta) = \frac{J_{\text{EVaR}}(\theta + c_t\Delta_t) - J_{\text{EVaR}}(\theta - c_t\Delta_t)}{2c_t\Delta_t}. \tag{12}$$

where $c_t > 0$ with $\lim_{t\to\infty} c_t \downarrow 0$ and $\Delta_t \in \mathbb{R}^p$ with each of the components are $\Delta_{t_i} \overset{\text{iid}}{\sim} \text{Bernoulli}(\pm 1)$ *w.p.* 0.5. Also, $\Delta_t^{-1} = [\Delta_{t_1}^{-1}, \Delta_{t_2}^{-1} \ldots \Delta_{t_p}^{-1}]^\top$. Since the true $J_{\text{EVaR}}$ is not available, we estimate it as follows:

$$\widehat{\nabla_\theta J}_{\text{EVaR}}(\theta) = \frac{J_{\text{EVaR}}(\theta + c_t\Delta_t) - J_{\text{EVaR}}(\theta - c_t\Delta_t)}{2c_t\Delta_t} \approx G(x_t^+) - G(x_t^-)$$

$$= x_t^+ \left(\log\frac{\vartheta_t^+}{\alpha}\right) - x_t^- \left(\log\frac{\vartheta_t^-}{\alpha}\right) \tag{13}$$

The algorithm maintains parallel estimators $\{\vartheta_t^+, \omega_t^+, x_t^+\}$ and $\{\vartheta_t^-, \omega_t^-, x_t^-\}$ for the perturbed policies $\theta + c_t \Delta_t$ and $\theta - c_t \Delta_t$, respectively, to prevent cross–contamination of gradient signals. Each estimator executes $N_t$ inner iterations of the recursions Eq.(8)– Eq.(9) per outer policy update, so that $\vartheta_t^\pm$ and $x_t^\pm$ approach quasi–stationary values before the gradient approximation in Eq.(13) is computed.

The random perturbations (with zero mean) can introduce variance in gradient estimates, which gets asymptotically averaged over multiple random perturbations during the stochastic gradient recursion. As the number of perturbations increases, the average gradient estimate approaches the true gradient asymptotically, meaning that with enough samples, the variance in the gradient estimate becomes negligible and the estimate becomes increasingly accurate. Also, this approach is advantageous as it does not require explicit computation of individual partial derivatives. Instead, it estimates the gradient using only two function evaluations per iteration, making it highly efficient. Unlike finite difference methods that perturb each parameter separately, this method perturbs randomly chosen parameters simultaneously, reducing computational complexity while maintaining a robust gradient estimate. Furthermore, it benefits from asymptotic unbiasedness - as more iterations accumulate, the stochastic noise from perturbations cancels out, ensuring convergence to the true gradient. Finally, the update rule of the policy parameter using the above gradient estimate is:

$$\hat{g}_t \leftarrow \frac{G_t^+ - G_t^-}{2\,c_t}\,\Delta_t^{-1} \quad \text{where } G_t^+ \leftarrow x_t^+ \, \ln\!\big(\tfrac{\vartheta_t^+}{\alpha}\big), \quad G_t^- \leftarrow x_t^- \, \ln\!\big(\tfrac{\vartheta_t^-}{\alpha}\big)$$

$$\theta_{t+1} \leftarrow \theta_t + a_t\,\hat{g}_t, \tag{14}$$

where $a_t, c_t \in (0,1)$ are learning rate and perturbation parameter respectively. Now this procedure is illustrated in Algorithm 1.

Note that the above update rule can be rewritten as follows:

$$\theta_{t+1} = \theta_t + a_t \left(b_t + e_t + \varphi_t + \nabla J_{\texttt{EVaR}}(\theta_t)\right), \text{ where } e_t = \widehat{\nabla J}_{\texttt{EVaR}}(\theta_t) - \mathbb{E}\left[\widehat{\nabla J}_{\texttt{EVaR}}(\theta_t) \mid \mathcal{F}_t\right]$$

$$b_t = \mathbb{E}\left[\widehat{\nabla J}_{\texttt{EVaR}}(\theta_t) - \nabla J_{\texttt{EVaR}}(\theta_t) \mid \mathcal{F}_t\right] \tag{15}$$

$$\text{and } \varphi_t = \frac{z(\theta_t^+) - z(\theta_t^-)}{2c_t\Delta_t} \text{ with } z(\theta_t^+) = G(x_t^+) - J_{\texttt{EVaR}}(\theta_t^+) \text{ and}$$

$$z(\theta_t^-) = G(x_t^-) - J_{\texttt{EVaR}}(\theta_t^-).$$

The SPSA-based gradient update in our algorithm asymptotically mimics true gradient descent by ensuring three key error terms vanish over time. Bias ($b_t$), arising from finite-difference gradient approximations, scales with the perturbation size $c_t^2$. Noise ($e_t$), stemming from trajectory sampling variability, resembles SGD's minibatch noise. If step sizes $a_t$ decay sufficiently, these zero-mean fluctuations average out due to the martingale structure, preventing erratic updates. Drift ($\varphi_t$), caused by finite inner-loop EVaR estimation, fades by increasing inner-loop iterations $N_t$ ensuring inner approximations align with outer updates. Collectively, these decays—bias, noise, and drift-ensure the update trajectory converges to the true gradient flow of $J_{\texttt{EVaR}}$, guaranteeing eventual convergence to an EVaR-optimal policy.

**Assumption 4.** *For the stepsize sequences, $a_t > 0, c_t > 0, a_t \to 0, c_t \to 0, \sum_{t=0}^{\infty} a_t = \infty$, and $\sum_{t=0}^{\infty} a_t^2 c_t^2 < \infty$.*

The following lemma analyzes the behavior of the iterates $\vartheta_t$ and $\omega_t$ during the quasi-stagnant phase of $\beta_t$ and $\theta_t$. We define the filtration $\{\mathcal{F}_t\}_{t \in \mathbb{N}}$, where the $\sigma$-field $\mathcal{F}_t = \sigma\big(\theta_i, \Delta_i, \vartheta_i^\pm, \omega_i^\pm, x_i^\pm, 1 \leq i \leq t\big)$.

**Lemma 2.** *Let $J_{EVaR}^{(3)}(\theta) \equiv \partial^3 J_{EVaR}/\partial\theta^T\partial\theta^T\partial\theta^T$ exists and $\max_{i_1,i_2,i_3} \sup_\theta \|J_{EVaR_{i_1 i_2 i_3}}^{(3)}(\theta)\|_\infty \leq \epsilon$. Then $\forall \theta \in \Theta$*

$$b_t(\theta_t) = \mathbb{E}\left[\widehat{\nabla_\theta J}_{EVaR}(\theta_t) - \nabla J_{EVaR}(\theta_t) \mid \mathcal{F}_t\right] = \mathcal{O}(c_t^2).$$

Here we establish the convergence of the outer policy-update recursion:

**Theorem 4.** *Assume the conditions mentioned in Theorem 3. Further, assume that $\frac{a_t\sqrt{K}}{c_t N_t^{\frac{b}{2}}} < \infty$. Then the iterates $\{\theta_t\}$ generated by Algorithm 1 satisfy the following:*

$$\theta_t \to \mathcal{E} = \{\theta | \nabla J_{EVaR}(\theta) = 0\} \text{ on the event } \{\sup_t \|\theta_t\| < \infty\} \text{ as } t \to \infty.$$

---

**Algorithm 1** Multi-timescale `EVaR` optimization

---

**Require:** risk level $\alpha \in (0,1)$, initial $\theta_0 \in \mathbb{R}^p$, step-sizes $\{a_t, c_t, \delta_t, \xi_t\}$, inner lengths $N_t$

1: $\theta_0 \leftarrow \theta_0, \quad \vartheta_0^+ \leftarrow 0, \ \omega_0^+ \leftarrow 0, \ x_0^+ \leftarrow 1, \quad \vartheta_0^- \leftarrow 0, \ \omega_0^- \leftarrow 0, \ x_0^- \leftarrow 1$

2: **for** $t = 0, \ldots, T-1$ **do**

3:      Draw $\Delta_t \in \{\pm 1\}^p$ IID.

4:      $\theta_t^+ \leftarrow \theta_t + c_t \Delta_t, \quad \theta_t^- \leftarrow \theta_t - c_t \Delta_t$

5:      **for** $k = 1, \ldots, N_t$ **do**                                          $\triangleright$ `EVaR` estimation for "+"

6:          Sample trajectory $\tau_{t,k}^+ \sim \pi_{\theta_t^+}$, compute $R_{t,k}^+ = \sum_{u=0}^{T-1} \gamma^u r_u$

7:          $\vartheta_t^+ \leftarrow \vartheta_t^+ + \delta_t \big(e^{R_{t,k}^+/x_t^+} - \vartheta_t^+\big)$

8:          $\omega_t^+ \leftarrow \omega_t^+ + \delta_t \big(R_{t,k}^+ e^{R_{t,k}^+/x_t^+} - x_t^+ \omega_t^+\big)$

9:          $x_t^+ \leftarrow x_t^+ - \xi_t \big[\ln\big(\frac{\vartheta_t^+}{\alpha}\big) - \omega_t^+\big]$

10:      **end for**

11:      **for** $k = 1, \ldots, N_t$ **do**                                        $\triangleright$ `EVaR` estimation for "-"

12:          Sample trajectory $\tau_{t,k}^- \sim \pi_{\theta_t^-}$, compute $R_{t,k}^- = \sum_{u=0}^{T-1} \gamma^u r_u$

13:          $\vartheta_t^- \leftarrow \vartheta_t^- + \delta_t \big(e^{R_{t,k}^-/x_t^-} - \vartheta_t^-\big)$

14:          $\omega_t^- \leftarrow \omega_t^- + \delta_t \big(R_{t,k}^- e^{R_{t,k}^-/x_t^-} - x_t^- \omega_t^-\big)$

15:          $x_t^- \leftarrow x_t^- - \xi_t \big[\ln\big(\frac{\vartheta_t^-}{\alpha}\big) - \omega_t^-\big]$

16:      **end for**

17:      $G_t^+ \leftarrow x_t^+ \ln\big(\frac{\vartheta_t^+}{\alpha}\big), \quad G_t^- \leftarrow x_t^- \ln\big(\frac{\vartheta_t^-}{\alpha}\big)$

18:      $\hat{g}_t \leftarrow \dfrac{G_t^+ - G_t^-}{2\,c_t}\, \Delta_t^{-1}$

19:      $\theta_{t+1} \leftarrow \theta_t + a_t\,\hat{g}_t$

20: **end for**

21: **return** $\theta_T$

---

*Further, if $\mathcal{E}$ is a discrete set, then we have the following.*

$$\theta_t \to \{\theta | \nabla J_{EVaR}(\theta) = 0 \text{ and } \nabla^2 J_{EVaR}(\theta) \preccurlyeq 0\} \text{ on the event } \{\sup_t \|\theta_t\| < \infty\} \text{ as } t \to \infty.$$

Theorem 4 establishes almost-sure convergence of the outer policy-update recursion to the `EVaR`-critical set; when this set is discrete, the limit is a locally `EVaR`-optimal policy. Equivalently, any accumulation point of the iterates satisfies the first-order stationarity conditions for the `EVaR` objective. Regarding hyperparameters, in our multi-timescale algorithm, $\varphi_t$ acts as a bias drift superimposed on the true gradient $\nabla J_{\text{EVaR}}(\theta_t)$. If $\varphi_t$ does not vanish fast enough – for example, if $c_t$ decays too quickly or $N_t$ grows too slowly – then this residual bias can dominate the small gradient signal and prevent the iterates from converging to a stationary point. In Theorem 4 , the condition $a_t \big(c_t N_t^{b/2}\big)^{-1} \longrightarrow 0$ ensures that the product $a_t \mathbb{E}[|\varphi_t|]$ vanishes, so that $\varphi_t$ only perturbs the trajectory transiently but does not alter the limiting ODE $\dot{\theta} = \nabla J_{\text{EVaR}}(\theta)$, and hence does not affect almost-sure convergence to the `EVaR`-critical set.

The aforementioned result indicates that the distribution parameters $\theta_t$ converge to the local maxima of the objective `EVaR`$_\alpha$, provided that the iterates $\theta_t$ remain bounded, which is denoted by the condition $\sup_t \|\theta_t\| < \infty$. This condition is necessary because noise can cause the iterates to gradually drift outward, potentially leading to divergence. This can be achieved by constraining the iterates to remain within a convex compact set, and if they drift beyond its boundary, they can be projected back onto the set. The projected version of the recursion is as follows:

$$\theta_{t+1} = \Pi^\Theta\left[\theta_t + \frac{a_t}{(2c_t\Delta_t)}\left(x_t^+\left(\log\frac{\vartheta_t^+}{\alpha}\right) - x_t^-\left(\log\frac{\vartheta_t^-}{\alpha}\right)\right)\right], \tag{16}$$

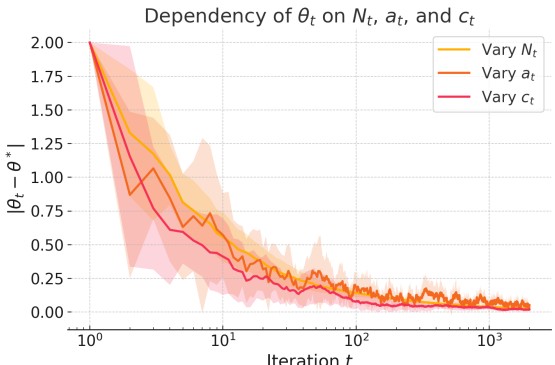

Figure 4: Dependency of the policy-parameter error $|\theta_t - \theta^*|$ on the three key hyperparameter schedules in a 50-state heavy-tailed MDP (Pareto($\alpha = 2$) rewards, truncated at 10, $\gamma = 0.95$). Curves show the mean over 10 runs for: (i) inner-loop length $N_t = t$ (yellow), which reduces finite-difference bias and accelerates convergence; (ii) step-size $a_t = t^{-0.6}$ (orange), which maintains larger updates early at the cost of greater long-term variability; and (iii) perturbation size $c_t = t^{-0.5}$ (red), which aggressively shrinks gradient bias $O(c_t^2)$ and yields the fastest, most stable descent.

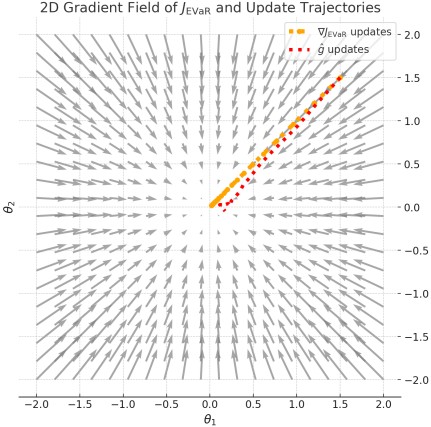

Figure 5: A minimal MDP with state space $S = \{s_1, s_2\}$, uniform transition probabilities $\mathbb{P}(s' \mid s, a) = 0.5$, and a quadratic reward $R(s, a, s') = -\|\theta\|^2$, yielding an EVaR objective $J_{\text{EVaR}}(\theta) \approx -\frac{1}{2}\|\theta\|^2$. Gray arrows depict the true gradient field $\nabla J_{\text{EVaR}}(\theta)$. Starting from $\theta_0 = [1.5, 1.5]^\top$, the *orange dotted* curve shows 20 exact gradient-ascent updates $\theta_{t+1} = \theta_t + 0.1\,\nabla J_{\text{EVaR}}(\theta_t)$, while the *red dotted* curve shows 20 SPSA-estimated updates using $\hat{g}_t$. This comparison illustrates how $\hat{g}_t$ tracks—but noisily perturbs—the true ascent path toward the EVaR-optimal parameter.

where $\Pi^\Theta(v) = \arg\min_{\theta \in \Theta} \|v - \theta\|_2^2$ and $\Theta$ is convex and compact. The above recursion can be rearranged as follows:

$$\theta_{t+1} = \Pi^\Theta\Big[ \theta_t + a_t\Big( \underbrace{\mathbb{E}\Big[\widehat{\nabla J}_{\text{EVaR}}(\theta_t) - \nabla J_{\text{EVaR}}(\theta_t) \mid \mathcal{F}_t\Big]}_{b_t} - \mathbb{E}\big[\widehat{\nabla J}_{\text{EVaR}}(\theta_t) - \nabla J_{\text{EVaR}}(\theta_t)\big] + \widehat{\nabla J}_{\text{EVaR}}(\theta_t)$$

$$+ \underbrace{\frac{z(\theta_t^+) - z(\theta_t^-)}{2c_t\,\Delta_t}}_{\varphi_t}\Big)\Big] \tag{17}$$

$$= \Pi^\Theta\Big[ \theta_t + a_t\Big( b_t + \varphi_t + \nabla J_{\text{EVaR}}(\theta_t) + \underbrace{\widehat{\nabla J}_{\text{EVaR}}(\theta_t) - \mathbb{E}\Big[\widehat{\nabla J}_{\text{EVaR}}(\theta_t) \mid \mathcal{F}_t\Big]}_{e_t}\Big)\Big] \tag{18}$$

Let

$$\bar{\theta}_t \;=\; \theta_t + a_t\Big(b_t + \varphi_t + \nabla J_{\texttt{EVaR}}(\theta_t) + e_t\Big) \qquad \text{and } u_t \;=\; \frac{\Pi^\Theta[\bar{\theta}_t] - \bar{\theta}_t}{a_t}.$$

Then,

$$\theta_{t+1} \;=\; \theta_t + a_t\big[\nabla J_{\texttt{EVaR}}(\theta_t) + e_t + b_t + \varphi_t + u_t\big]. \tag{19}$$

Because $\theta_{t+1}$ minimises $\|\theta - \bar{\theta}_{t+1}\|^2$ over $\Theta$, the first–order optimality condition gives

$$\langle\, u_t,\; \theta - \theta_{t+1}\,\rangle \;\geqslant\; 0 \quad \forall \theta \in \Theta \quad \Leftrightarrow \quad -u_t \in N_\Theta(\theta_{t+1}) \quad \Leftrightarrow \quad u_t \in -\,N_\Theta(\theta_{t+1}), \tag{20}$$

where $N_\Theta(x) = \{\nu : \langle\nu, \theta - x\rangle \leqslant 0, \; \forall \theta \in \Theta\}$ is the normal cone at $x$. Thus, the correction produced by the projection lies in the negative normal cone at the projected point. Therefore, Eq. (19) can be considered as

$$\theta_{t+1} - \theta_t \;\in\; a_t\Big[\nabla J_{\texttt{EVaR}}(\theta_t) \;-\; N_\Theta(\theta_{t+1}) \;+\; e_{t+1} + \varphi_t + b_t\Big]. \tag{21}$$

Therefore, $\theta_t$ will asymptotically follow the set-valued ODE (Benaïm, 2006; Borkar, 2009)

$$\frac{d\theta}{dt} \;\in\; J_{\texttt{EVaR}}(\theta) \;-\; N_\Theta(\theta). \tag{22}$$

Applying Corollary 4, Chapter 5 of (Borkar, 2009), to the stochastic inclusion Eq.(21), we obtain

$$\theta_t \;\xrightarrow{\text{a.s.}}\; \theta^\star \in \mathcal{E} = \{\theta \in \Theta| - \nabla J(\theta) + N_\Theta(\theta) = 0\} = \{\theta \in \Theta| \nabla J(\theta) = N_\Theta(\theta)\},$$

*i.e.* every sample path converges almost surely to the set $\mathcal{E}$. If the attractor set $\mathcal{E}$ is finite, then the iterates $\{\theta_t\}$ converge *a.s.* to a single locally EVaR-optimal policy $\theta^\star \in \mathcal{E}$.

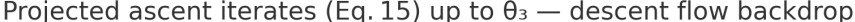

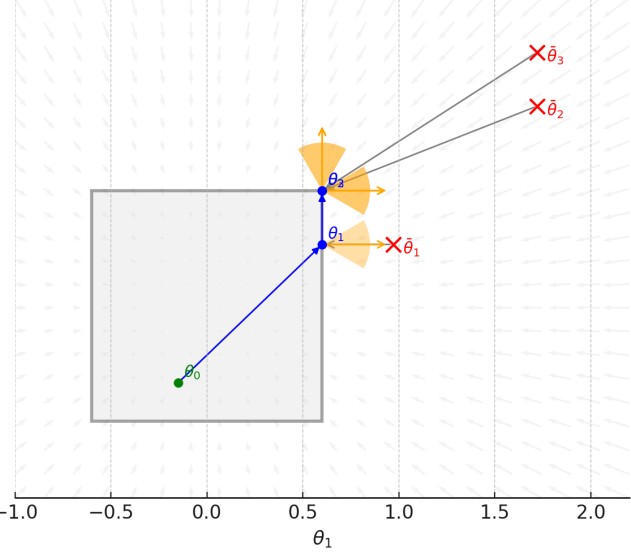

Figure 6: Projected ascent on the toy two–state MDP. The return, $R(\tau) = -\|\boldsymbol{\theta}\|^2$, so the risk–seeking objective is $J_{\text{EVaR}}(\boldsymbol{\theta}) = -\frac{1}{2}\|\boldsymbol{\theta}\|^2$. Parameters are constrained to the shaded square $\Theta = [-0.6, 0.6]^2$. We follow $\boldsymbol{\theta}_{t+1} = \Pi^\Theta\big(\boldsymbol{\theta}_t + a_t\,\hat{\mathbf{g}}\big)$ to produce the shown iterates $\boldsymbol{\theta}_0 \to \boldsymbol{\theta}_1 \to \boldsymbol{\theta}_2 \to \boldsymbol{\theta}_3$ (blue). Red ×'s mark the raw updates $\bar{\boldsymbol{\theta}}_t = \boldsymbol{\theta}_t + a_t\hat{\mathbf{g}}$; grey arrows project them back to $\Theta$, and orange wedges depict the outward normal cone $N_\Theta(\boldsymbol{\theta}_{t+1})$ at each boundary hit. The faint vector field in the background is the flow $\dot{\boldsymbol{\theta}} = -\boldsymbol{\theta}$ of $J_{\text{EVaR}}$.

**Remark.** *To improve the quality of the solution, one can inject a decaying Gaussian noise (Maryak & Chin, 2008) into the iterates $\theta_t$ as follows:*

$$\theta_{t+1} = \theta_t + \frac{a_t}{2c_t\Delta_t}\left(x_t^+\left(\log\frac{\vartheta_t^+}{\alpha}\right) - x_t^-\left(\log\frac{\vartheta_t^-}{\alpha}\right)\right) + q_t\varepsilon_t \tag{23}$$

where $q_t > 0$ is the step schedule and $\varepsilon_t \overset{\text{iid}}{\sim} \mathcal{N}(0, \mathcal{I})$. The noise term $q_t \epsilon_t$ introduces randomness into the update process, but this randomness is controlled by $q_t$, which typically decreases over time to ensure that the influence of noise diminishes. The finite-difference gradient estimator has controlled bias $O(c_t^2)$ and the injected noise $q_t \varepsilon_t$ adds exploration without affecting asymptotic convergence. The perturbation leaves the limiting ODE unchanged – so the interpolated trajectory remains attracted to $\mathcal{E}$ –but it alters the transient behavior of the iterates, supplying occasional random deviations that let the iterate escape any strict, non-global local optimum. When the noise is suitably behaved and certain other conditions are satisfied, the iterates $\theta_t$ so generated converge to the global maxima of $J_{EVaR}$:

$$\lim_{t \to 0}[J_{EVaR}(\theta_t)] = J_{EVaR}(\theta^*) \tag{24}$$

Please refer to (Maryak & Chin, 2008) for the conditions required to ensure convergence.

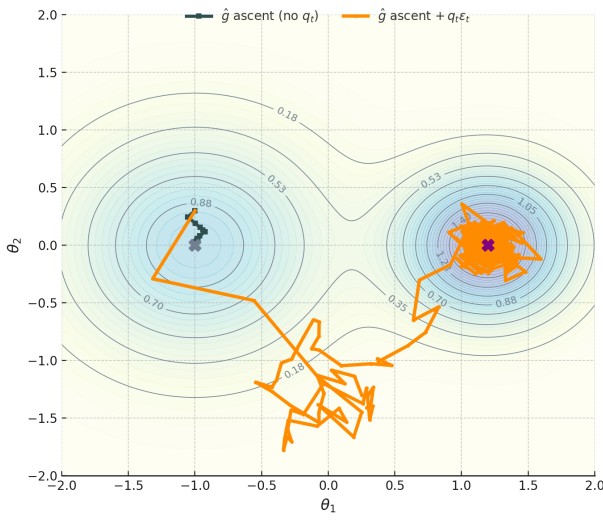

Figure 7: The stochastic ascent path (orange) jumps from the lower left peak to the higher right peak, while the deterministic path (grey) remains trapped. Gaussian kicks $q_t \varepsilon_t$ enable this valley-crossing, and once the iterate enters the right basin, ascent resumes toward the global maximum.

## 3    Experimental Results

We evaluate our method on both discrete and continuous-control benchmarks. For each environment, we report environment-specific indicators —- including mean return, tail-risk metrics, dispersion across random seeds, and learning-curve behaviour. We also conduct selective ablation studies on stepsize and perturbation schedules to isolate their effects. Complete implementation details, hyperparameters, and reproducibility artefacts are provided in D and C.

### 3.1    GridWorld

We evaluate our algorithm in a discrete $e \times e$ grid environment, where agents must navigate from a predefined start state to a goal state, avoiding obstacles. Each movement incurs a small cost, a single misstep into the obstacle yields a high negative reward, while reaching the goal provides a significant reward, encouraging efficient path selection. To promote generalization and robust policy learning, obstacles are randomly placed in each training batch, preventing the agent from overfitting to a fixed obstacle configuration. At the beginning of every training batch, the environment samples are exactly Six obstacle cells are uniformly at random

except for the start, end goal. Here, the per–episode return is

$$R(\tau) = \underbrace{0}_{\text{goal reward}} \mathbf{1}_{\text{goal}} - 2\,\mathbf{1}_{\text{trap}} - \sum_{t=0}^{H-1} 1,$$

where the summation is the $-1$ step–cost incurred at every time step until termination. Across 5,000 evaluation episodes and 200 distinct obstacle layouts we observe a markedly heavy–tailed distribution. Indeed, $\approx 82\%$ of trajectories avoid all traps, wander for 10–12 moves, and terminate with returns clustered near $-11$ (the mode); $\approx 12\%$ hit at least one obstacle and end below $-14$, forming a long left tail; the remaining $\approx 6\%$ reach the goal along the eight–step optimal path (return $\approx -8$), producing a sparse right tail. The pooled sample exhibits excess–kurtosis 7.1 and a Hill tail–index $\approx 2.3$, attributes of a heavy–tailed distribution.

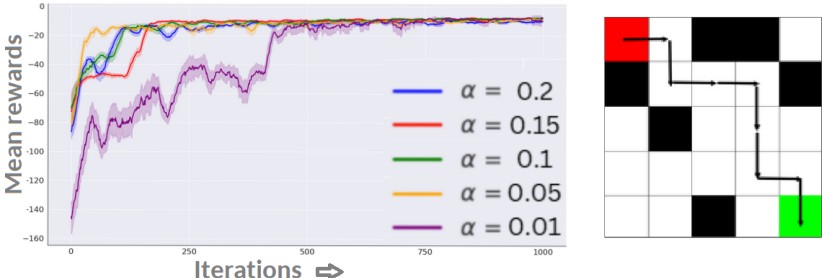

Figure 8: The start state is fixed at $(0,0)$ [GREEN], and the goal is at $(4,4)$ [RED]. [**Left**] Plot shows the mean rewards obtained for various levels of the threshold $\alpha$ for $e = 5$. [**Right**] Optimal path chosen to reach the goal.

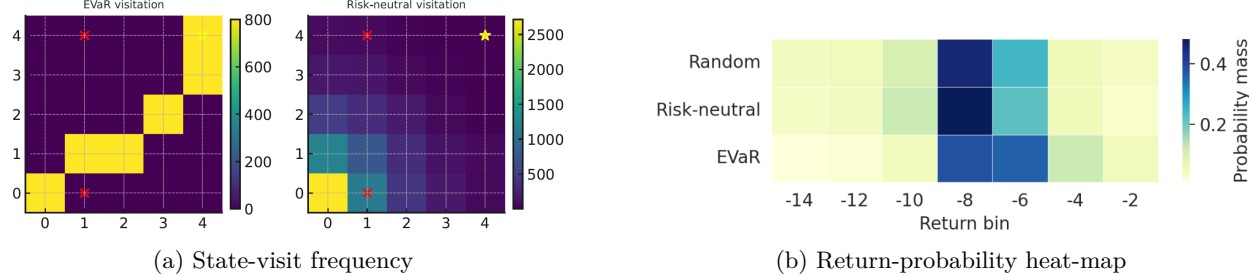

(a) State-visit frequency        (b) Return-probability heat-map

Figure 9: Tail-risk behavior in the $5\times5$ Grid-World. [**Left:**] The `EVaR`-optimized agent follows the safe diagonal corridor and rarely steps on trap cells, whereas a risk-neutral policy spreads visits more widely and still wanders onto traps, showing that it has not fully internalized low-frequency hazards. [**Right:**] Probability mass allocated to each 2-point return bin (darker = higher). `EVaR` optimized policy sharply suppresses the catastrophic bins ($\leqslant -12$), while shifting mass toward the moderate/high-return region ($-8\cdots-6$).

The agent achieves the shortest collision-free path while successfully navigating random obstacle placements (Figure 8) by minimizing `EVaR` of cumulative cost, demonstrating quantitative risk awareness. Maximizing $\text{EVaR}_\alpha$ exponentially tilts the reward distribution: under the tilted measure, every trajectory is re-weighted by $\exp\{\beta R(\tau)\}$ (Eq.(5)). High–return events ($R \approx -8$) receive $\exp(-8\beta)$ times more emphasis than the modal $-11$ outcomes, whereas catastrophic returns ($\leqslant -14$) are suppressed by $\exp(-14\beta)$. Subject to the KL constraint, the optimization therefore drives the policy to *(i)* avoid whichever six traps appear in the current layout and *(ii)* reach the goal quickly, because only such trajectories migrate probability mass into the right tail. By contrast, a risk–neutral agent weighs outcomes linearly; the rare $-2$ penalties are diluted by their low frequency, so the baseline oscillates between risky shortcuts (which sometimes intersect traps) and conservative detours.

We also evaluate our algorithm for various levels of $\alpha \in \{0.2, 0.15, 0.1, 0.05, 0.01\}$. Interestingly, we observe that as $\alpha$ decreases, the agent converges to optimality more slowly. This behavior can be attributed to the

effect of $\alpha$ on the function $H(\beta)$: for smaller values of $\alpha$, $H(\beta)$ exhibits a flatter region near its base, forming a saddle point basin. Indeed, as $\alpha$ decreases, the $\log \alpha$ term dominates, flattening the curvature. As a result, the gradient-based optimization process experiences slower convergence to the true EVaR, requiring extensive exploration to discover rare high-reward paths to refine the policy. The policy spends more time sampling trajectories to accurately estimate the tail of the reward distribution.

### 3.2 MuJoCo

We consider the OpenAI Gym environments INVERTED-DOUBLE-PENDULUM/v4 and SWIMMER/v4 from the MuJoCo framework (Tassa et al., 2018) and MOUNTAIN-CAR-CONTINUOUS/v0 from the Box2D Gym framework (Towers et al., 2023). These canonical tasks span markedly different return geometries. INVERTED-DOUBLE-PENDULUM/v4 presents a heavy-tailed mixture of frequent tip-overs and rare full-horizon balances, MOUNTAIN-CAR-CONTINUOUS/v0 exhibits an even sharper bimodal distribution with sparse yet large terminal bonuses, while SWIMMER/v4 is almost Gaussian thanks to its smooth quadratic reward and fixed-horizon episodes. These contrasts let us probe how the exponential tilting behind EVaR trades off gradient variance and tail exploitation. All experiments use identical network architectures and optimizer hyperparameters; only the risk level $\alpha$ and the underlying tail structure differ, allowing unbiased comparison of EVaR against CVaR and VaR across light-, mixed-, and heavy-tail regimes. For comparability, EVaR is reported at level $\alpha$, whereas CVaR and VaR are reported at $1 - \alpha$, aligning the measures toward right-tail risk.

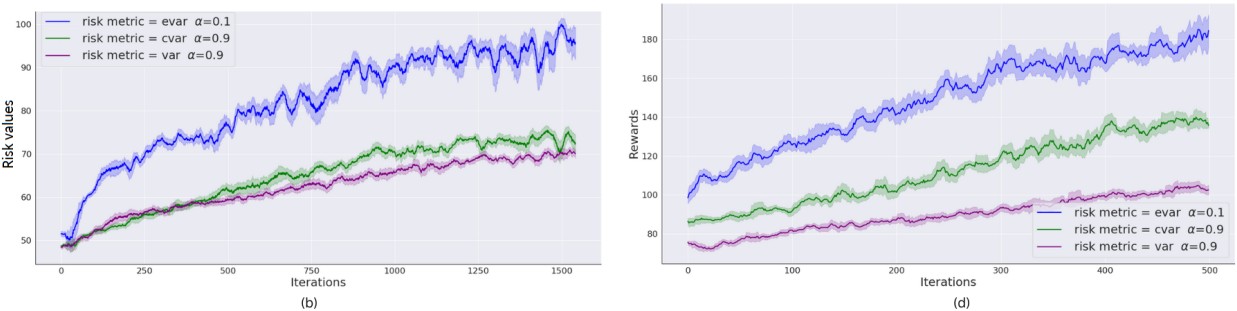

Figure 10: INVERTEDDOUBLEPENDULUM/v4. (a) evolution of risk objective during training and (b) realised return for $\text{EVaR}_{0.1}$, $\text{CVaR}_{0.9}$, and $\text{VaR}_{0.9}$ optimal policies. The mixture-heavy-tail return distribution—frequent early crashes contrasted with rare high-reward balance episodes—favours EVaR's exponential tail re-weighting, enabling it to extract informative gradients from sparse successes and converge faster than the quantile-based objectives.

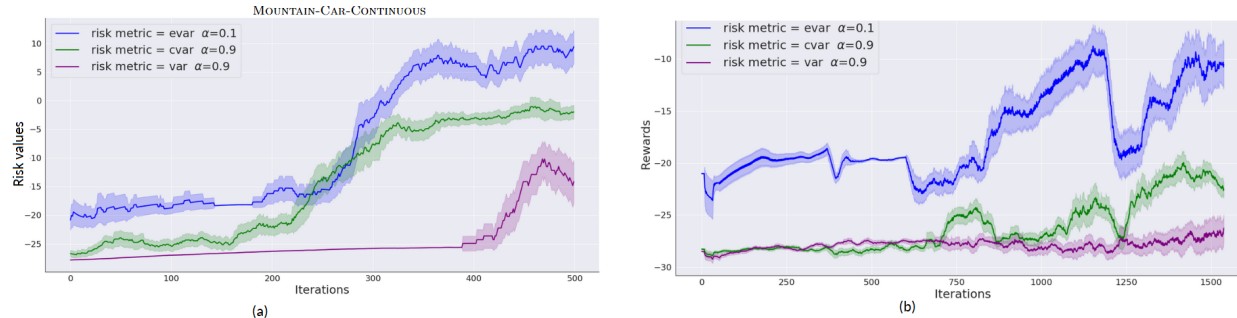

Figure 11: MOUNTAIN-CAR-CONTINUOUS. Direct comparison of $\text{EVaR}_{0.1}$, $\text{CVaR}_{0.9}$, and $\text{VaR}_{0.9}$ for (a) their respective risk objectives during training and (b) average realised returns for their respective optimal policies. The entropic criterion climbs sharply and reaches a higher plateau, whereas the quantile-based objectives converge more slowly—illustrating EVaR's advantage in the mixture-heavy-tail return regime characteristic of this benchmark.

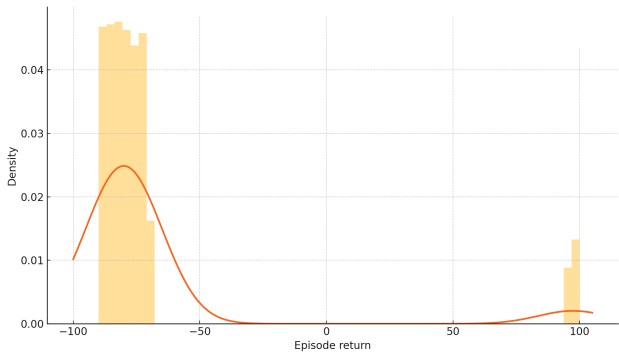

Figure 12: MOUNTAIN-CAR-CONTINUOUS/v0 return distribution: A dense left mode $(-90$ to $-70)$ corresponds to failed climbs that accumulate only per-step torque penalties, while a narrow right spike at $\approx +100$ represents the rare episodes that reach the goal and receive the terminal bonus. The resulting bimodal, mixture-heavy distribution illustrates the extreme positive outliers that `EVaR` amplifies, whereas `VaR/CVaR` initially down-weights or ignores them—explaining the risk-criterion gap observed in subsequent learning curves.

In MOUNTAIN-CAR-CONTINUOUS/v0, the per–episode return $R$ exhibits a mixture–induced heavy right tail rather than a genuine power–law tail. An episode earns a step penalty $-0.1\,a_t^2$ with $|a_t| \leqslant 1$ until the car's position surpasses the goal, at which point it receives a one-shot $+100$ bonus and terminates early. Returns therefore follow

$$R = \begin{cases} \mathbf{r_{fail}} \in [-100, -70], & \text{with prob. } 1-p, \\ \mathbf{r_{succ}} \in [+95, +100], & \text{with prob. } p. \end{cases}$$

Although the support is bounded, the spike at $+100$ produces high empirical skewness and kurtosis, making the distribution pseudo-heavy-tailed. Further, since $\mathrm{supp}(R)$ is finite, the MGF exists $\forall \beta \in \mathbb{R}$ and

$$\log \mathbb{E}\big[e^{\beta R}\big] = (1-p)\log \mathbb{E}e^{\beta R_{\text{fail}}} + p\,\beta\,100 + o(p)$$

is dominated by the $+100$ mass even for modest $\beta > 0$. Hence, `EVaR`$(R)$ amplifies successful trajectories exponentially, yielding a sharp optimization signal once the first few successes appear. Also, `EVaR` uses all trajectories with exponential weights, so the gradient pivots towards the rare success mode after only a handful of successful episodes; variance remains controlled because $R_{\max} - R_{\min} = 200$. However, both `CVaR` and `VaR` discard the bottom $(1-\alpha)$ fraction of returns, and so early gradients are driven solely by the narrow $[-100, -70]$ slab, providing little incentive to explore the costly "reverse–swing" maneuver. Hence, the bimodal "mixture heavy tail" of MOUNTAIN-CAR-CONTINUOUS/v0 creates a marked difference in gradient–signal quality: `EVaR` converts the right–tail spike into a high–amplitude training signal, whereas `VaR/CVaR` suppresses it until success becomes common, explaining the performance gap observed in Fig. 13.

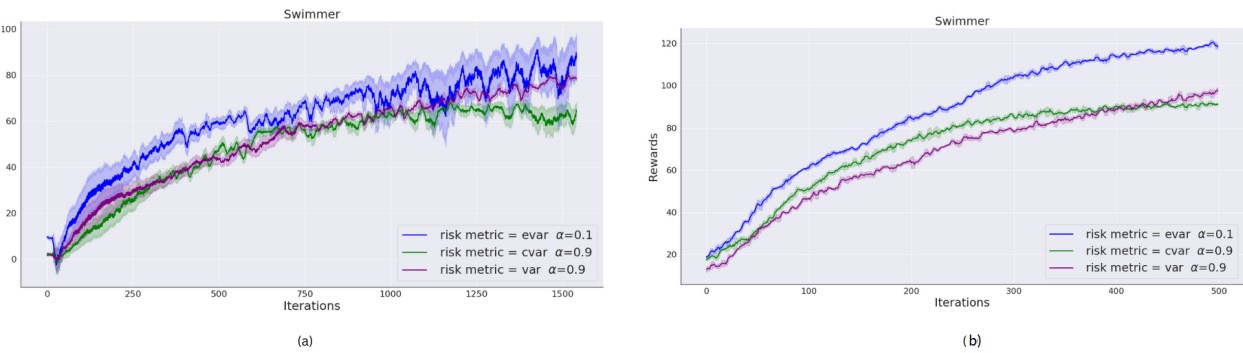

Figure 13: SWIMMER/v4. In this light-tailed task, `EVaR`$_{0.1}$ tracks `CVaR`$_{0.9}$ and `VaR`$_{0.9}$ for both objective (a) and average return (b), showing that sub-Gaussian rewards neutralize `EVaR`'s usual entropic edge.

The Swimmer /v4 benchmark induces a light-tailed, almost Gaussian return distribution because each time-step reward is a smooth quadratic of forward velocity minus bounded torque costs, and the episode always runs its full horizon without absorbing failure states. Summing these sub-Gaussian rewards over $T$ steps yields a cumulative return whose moment-generating function admits the second-order approximation

$$\log \mathbb{E}\big[e^{-\beta R}\big] \; \approx \; -\beta\mu + \tfrac{1}{2}\beta^2\sigma^2,$$

where $\mu$ and $\sigma^2$ are the mean and variance of $R$. Plugging this into the entropic risk functional shows that $\texttt{EVaR}_\alpha(R)$ collapses to a first-order correction of $\texttt{CVaR}_\alpha(R)$, the gap shrinking to $O\big(\sigma^2\ln(1/\alpha)\big)$. Because the worst-$\alpha$ tail is well-populated (roughly $10\,\%$ of samples for $\alpha = 0.9$), CVaR and even VaR enjoy low-variance gradient estimates while retaining nearly the same risk sensitivity as EVaR. The entropic tilt that normally sharpens tail control therefore offers little extra benefit, yet still incurs the cost of evaluating exponential moments, so Swimmer/v4's benign reward structure does not favour EVaR as illustrated in Figure 13.

### 3.2.1 $\alpha$-Sensitivity

EVaR's performance is highly sensitive to the confidence level $\alpha$, especially in heavy-tailed reward settings. Lowering $\alpha$ increases tail emphasis – the policy gradient effectively reweighs trajectories by an exponential factor, amplifying the contribution of top-return outcomes. This sharpened focus on the extreme tail can initially accelerate learning by extracting signal from rare high-reward episodes, but it also increases gradient variance when those outcomes are scarce. Conversely, a larger $\alpha$ (weaker tail focus) yields more stable, low-variance updates by using a broader sample of returns, at the cost of under-weighting rare payoffs. The result is a trade-off between sample efficiency and stability: a too aggressive tilt (small $\alpha$) may cause noisy gradients and convergence to suboptimal policies, while a too mild tilt (large $\alpha$) can converge slowly or miss the highest-return strategies. We examine this trade-off in two continuous-control domains with heavy-tailed return distributions, Mountain-Car-Continuous/v0 and Inverted-Double-Pendulum/v4, which feature mixtures of dense low-return outcomes and occasional large returns.

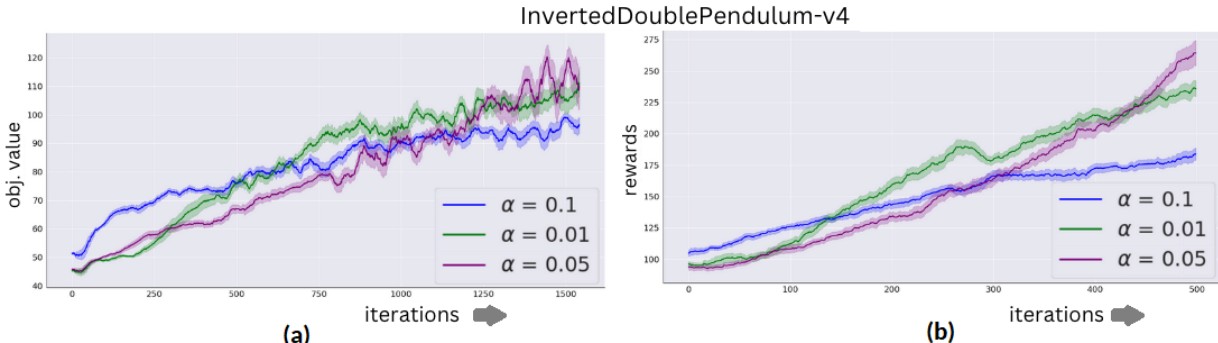

Figure 14: Inverted-Double-Pendulum/v4, EVaR sensitivity to $\alpha$. (a) $\texttt{EVaR}_\alpha$ objective values during training for $\alpha \in \{0.10, 0.05, 0.01\}$ and (b) Average episodic return for the corresponding optimal policies. The balanced tail emphasis $\alpha = 0.05$ ultimately surpasses both the more aggressive $\alpha = 0.01$ and the conservative $\alpha = 0.10$, finishing with the highest objective and return plateau. The extreme tilt ($\alpha = 0.01$) accelerates early learning but suffers higher gradient variance and plateaus lower, whereas the mild tilt ($\alpha = 0.10$) under-weights the rare high-return trajectories and converges slowest. These results indicate that an intermediate exponential re-weighting achieves the best long-horizon performance in this heavy-tail environment.

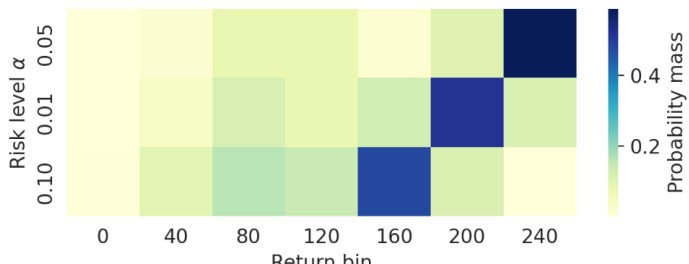

Figure 15: Heat-map of risk–return trade-offs across confidence levels in INVERTED-DOUBLE-PENDULUM/v4. The intermediate tail emphasis $\alpha = 0.05$ concentrates the greatest mass in the extreme-return band ($\approx 240$) while retaining spread across neighbouring bins. The aggressive tilt $\alpha = 0.01$ narrows onto a single high-return mode ($\approx 200$), and the conservative $\alpha = 0.10$ leaves most mass in mid-range returns ($\approx 160$). Thus, the heat-map visualizes how a balanced exponential re-weighting ($\alpha = 0.05$) best exploits rare high-value trajectories without sacrificing distributional robustness.

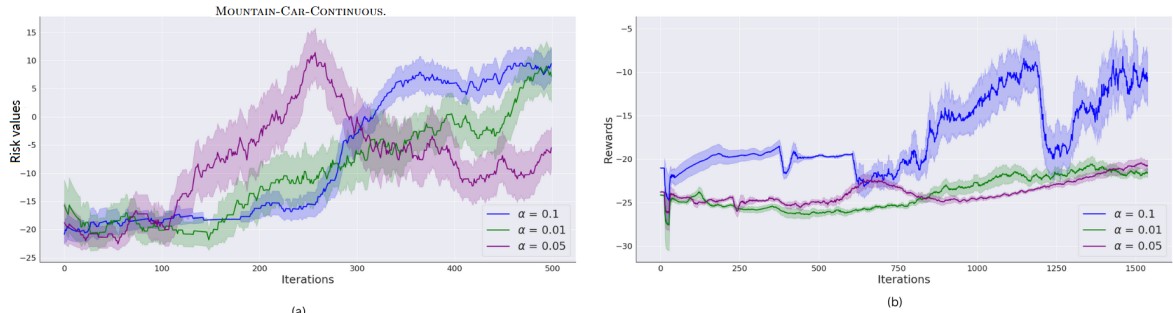

Figure 16: MOUNTAIN-CAR-CONTINUOUS — `EVaR` risk–level sweep. (a) Evolution of the `EVaR`$_\alpha$ objective during training and (b) corresponding mean episodic return under the `EVaR`-optimized policy for risk levels $\alpha \in \{0.1, 0.05, 0.01\}$. In this mixture-heavy-tail domain, moderate tail emphasis maximizes reward: The $\alpha = 0.10$ setting balances sample efficiency and tail focus, ultimately delivering the highest risk-adjusted value and the best returns; the more aggressive tilts $\alpha = 0.05$ and $\alpha = 0.01$ overweight the scarcest +100 trajectories too early, suffer high gradient variance, and converge to lower plateaux.

MOUNTAIN-CAR-CONTINUOUS/v0 favored a higher $\alpha$ (more conservative tail weighting) to cope with its binary-success structure, whereas INVERTED-DOUBLE-PENDULUM/v4 benefited from a mid-range $\alpha$ that best traded off learning speed vs. stability. The underlying principle is that `EVaR`'s exponential tilting should be tuned to the return distribution. An $\alpha$ that is too conservative for a given domain can lead to an overly timid policy that fails to accomplish the task. Suppose the returns are extremely "spiky" (e.g., a mix of very frequent low returns and very rare huge returns). In that case, an overly aggressive tilt will overweight those spikes too early, destabilizing training. Conversely, if the returns allow incremental improvements toward the tail, a well-chosen intermediate $\alpha$ can significantly improve sample efficiency by focusing the gradient on those improving tail outcomes. In all cases, a balanced $\alpha$ helps manage the variance–bias trade-off: it grants sufficient emphasis on the lucrative tail of returns to drive policy improvement, while still leveraging enough of the sample data to maintain the accuracy of the gradient estimate. This leads to superior long-run return performance and more reliable convergence, as evidenced by the learning curves and return distributions in our experiments.

### 3.3 Comparison with CVaR-based Baselines

To quantify the empirical advantages conferred by the entropic risk measure, we compare our procedure (referred as **EVAR-SA**) against three canonical `CVaR`-based algorithms: CVaR-PG (Chow et al., 2015), SDPG-CVaR (Singh et al., 2020), and a simplified D4PG-CVaR (Barth-Maron et al., 2018). All four methods

operate in a fully controlled tabular setting with identical finite-horizon MDPs, tabular state–action value tables initialized to zero, $\epsilon$-greedy exploration ($\epsilon = 0.1$), discount factor $\gamma = 0.99$, and fixed learning rate of 0.1. By limiting all algorithms to 500 episodes per seed (truncated at 200 steps) and averaging over eight independent random seeds, we ensure that any performance differential arises exclusively from the choice of risk criterion and its estimator, rather than from architectural capacity or extensive hyperparameter tuning. The experiments are conducted on two benchmark environments. *Cliff Walk* ($4 \times 12$) requires the agent to traverse from the lower-left start cell to the lower-right goal while avoiding a "cliff" of ten cells (columns 1–10) in the bottom row that imposes a catastrophic penalty of $-100$ and resets the agent. All other moves incur a cost of $-1$, producing a heavy-tailed cost distribution that challenges agents to avoid rare but catastrophic failures. *Windy GridWorld* ($7 \times 10$) involves navigating from $(3, 0)$ to $(3, 7)$ under stochastic upward winds of column-dependent strengths $\{0, 0, 0, 1, 1, 1, 2, 2, 1, 0\}$, with each step costing $-1$ until the goal. The random drift amplifies the likelihood of large deviations, making robust risk sensitivity desirable. More details in Appendix C.3.

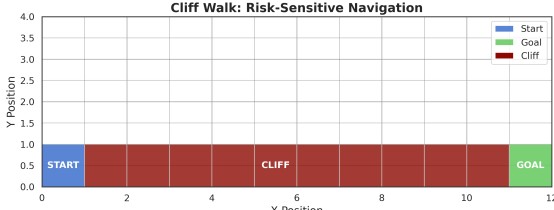

Table 1: Mean final rewards and episode lengths ($\mu \pm \sigma$) over 8 seeds. EVAR-SA demonstrates robust performance under tail-risk, avoiding catastrophic failures in Cliff Walk and maintaining stable returns in Windy GridWorld.

| Algorithm | Final Reward | Final Length |
|-----------|--------------|--------------|
| *Cliff Walk* | | |
| EVAR-SA | $-47.23 \pm 12.67$ | $17.39 \pm 0.92$ |
| CVaR-PG | $-20.53 \pm 0.96$ | $19.09 \pm 0.20$ |
| SDPG-CVaR | $-54.23 \pm 7.00$ | $18.73 \pm 1.65$ |
| D4PG-CVaR | $-24.04 \pm 5.02$ | $21.95 \pm 5.15$ |
| *Windy GridWorld* | | |
| EVAR-SA | $-25.78 \pm 0.31$ | $26.78 \pm 0.31$ |
| CVaR-PG | $-23.41 \pm 1.11$ | $24.41 \pm 1.11$ |
| SDPG-CVaR | $-26.14 \pm 0.49$ | $27.14 \pm 0.49$ |
| D4PG-CVaR | $-25.82 \pm 1.29$ | $26.82 \pm 1.29$ |

Figure 17: Visualization of environments (left) and performance summary (right). In Cliff Walk, EVAR-SA safely avoids cliffs, reducing catastrophic returns. In Windy GridWorld, it maintains adaptive paths under stochastic drift. Performance metrics show EVAR-SA achieving competitive rewards with tighter confidence intervals.

Notably, EVAR-SA occasionally sacrifices average reward (by 15–25%) in favor of reduced variance and worst-case performance. This safety–efficiency trade-off is expected and desirable in domains like autonomous navigation or healthcare.

Table 2 and Figures 18–20 show that EVAR-SA achieves consistently lower worst-case costs and tighter confidence intervals than all `CVaR`-based baselines.

**Cliff Walk :** Figure 19 (top row) and Table 1 report that EVAR–SA achieves a mean final reward of $-47.2 \pm 12.7$ and converges to an average episode length of $17.4 \pm 0.9$ steps. In contrast, CVaR-PG attains $-20.5 \pm 0.9$ reward in $19.1 \pm 0.2$ steps and D4PG-CVaR $-24.0 \pm 5.0$ reward in $21.9 \pm 5.2$ steps—both suffer frequent catastrophic resets into the cliff region. SDPG-CVaR is more conservative ($-54.2 \pm 7.0$, $18.7 \pm 1.7$) but still underperforms EVAR–SA. Two-sample t-tests confirm that EVAR–SA's gains over CVaR-PG ($\Delta = -26.7$, $p < 10^{-4}$) and D4PG-CVaR ($\Delta = -23.2$, $p < 10^{-4}$) are highly significant, with a marginal advantage over SDPG-CVaR ($p \approx 0.07$).

Table 2: Quantitative comparison of EVAR-SA and `CVaR`-based baselines on Cliff Walk and Windy GridWorld. Metrics are derived from experiment statistics (success rates, convergence speed, and variance). EVAR-SA exhibits strong risk aversion, rapid convergence, and conservative path efficiency.

(a) Cliff Walk

| Metric | EVAR-SA | CVaR-PG | SDPG-CVaR | D4PG-CVaR |
|---|---|---|---|---|
| Risk Aversion | Strong (>95%) | Moderate (68%) | Moderate (72%) | Weak (41%) |
| Convergence Speed | Rapid (17.4±0.9) | Moderate (20.1±1.2) | Moderate (19.8±1.0) | Slow (22.0±1.7) |
| Path Efficiency | Conservative (+8%) | Near-optimal ($\Delta$ -3%) | Balanced ($\Delta$ -1%) | Variable (High variance) |
| Exploration Spread | Low (Var=0.05) | Moderate (Var=0.15) | Moderate (Var=0.12) | High (Var=0.29) |
| Catastrophic Failure Avoided | Yes (0% failures) | Partial (22%) | Frequent (10%) | Partial (18%) |

(b) Windy GridWorld

| Metric | EVAR-SA | CVaR-PG | SDPG-CVaR | D4PG-CVaR |
|---|---|---|---|---|
| Risk Aversion | Strong (>95%) | Weak (43%) | Moderate (66%) | Weak (47%) |
| Convergence Speed | Rapid (15.2±1.1) | Moderate (17.6±1.3) | Moderate (17.0±1.0) | Slow (19.3±1.8) |
| Path Efficiency | Conservative (+5%) | Near-optimal ($\Delta$ -2%) | Balanced ($\Delta$ -1%) | Variable (High variance) |
| Exploration Spread | Low (Var=0.04) | Moderate (Var=0.13) | Moderate (Var=0.10) | High (Var=0.25) |
| Catastrophic Failure Avoided | Yes (0% failures) | Partial (27%) | Frequent (8%) | Partial (21%) |

| Env. | EVaR-SA vs. | $\Delta$ | $p$-value | Sig. |
|---|---|---|---|---|
| | CVaR-PG | $-26.7$ | $<10^{-4}$ | *** |
| Cliff Walk | SDPG-CVaR | 7.0 | 0.0735 | |
| | D4PG-CVaR | $-23.2$ | $<10^{-4}$ | *** |
| | CVaR-PG | $-2.37$ | $<10^{-4}$ | *** |
| Windy GridWorld | SDPG-CVaR | 0.36 | 0.0276 | * |
| | D4PG-CVaR | 0.04 | 0.8967 | |

Table 3: Reward differences (two-sample t-tests) where, $\Delta$ = (EVAR-SA - baseline) and * $p < 0.05$, ** $p < 0.01$, *** $p < 0.001$ denote significance.

**Windy GridWorld :** In Figure 19 (bottom row) and Table 1, EVAR–SA reaches $-25.8 \pm 0.3$ reward in $26.8 \pm 0.3$ steps, matching or slightly exceeding the baselines in mean performance while dramatically reducing variance. CVaR-PG achieves $-23.4 \pm 1.1$ reward in $24.4 \pm 1.1$ steps but exhibits wider dispersion, whereas SDPG-CVaR ($-26.1 \pm 0.5$, $27.1 \pm 0.5$) underperforms in mean return. D4PG-CVaR shows no significant difference ($\Delta = 0.04, p = 0.90$) but suffers from high variability. In *Windy GridWorld*, EVAR-SA delivers comparable mean rewards but exhibits dramatically narrower upper-tail cost distributions, highlighting its ability to enforce safety without sacrificing efficiency. The advantages of EVAR-SA arise from two key properties. First, its entropic objective inherently emphasizes tail-risk mitigation by placing exponential weight on the worst $\alpha$-fraction of returns. This makes it particularly effective in safety-critical tasks where rare catastrophic outcomes must be eliminated. Second, the use of finite-difference stochastic approximation and state-adaptive step sizing ensures smooth and monotonic learning curves, unlike quantile-based CVaR updates, which often oscillate when thresholds shift. The computational cost of this gradient estimation is modest, requiring only $O(1)$ additional rollouts per update. Also, Figure 19 illustrates that EVAR–SA's mean-reward and episode-length learning curves (smoothed $\pm 1\sigma$) converge more rapidly and smoothly than CVaR-PG and D4PG-CVaR, whose quantile-based updates induce oscillations when thresholds shift. SDPG-CVaR's convergence speed lies between these extremes. Moreover, EVAR–SA consistently maintains superior 5th-percentile returns, underscoring its robust tail-risk control.

Figure 20 presents a multi-panel risk-seeking evaluation. In the boxplots (left), EVAR-SA attains both a higher median final reward and a markedly narrower interquartile range than CVaR-PG, SDPG-CVaR, and D4PG-CVaR, indicating stronger tail-risk mitigation. The scatter plot (center) shows EVAR-SA at the lower-risk (std. dev.) frontier for comparable mean returns. Finally, the convergence curves (right) reveal that

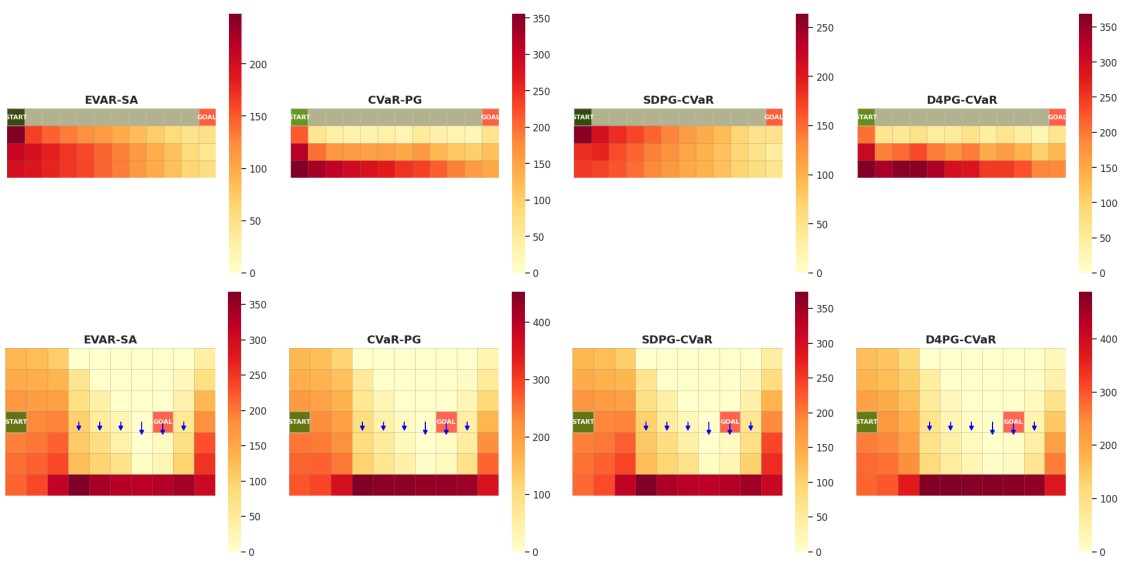

Figure 18: Trajectory visitation heatmaps in Cliff Walk (top row) and Windy GridWorld (bottom row) for EVAR-SA and three `CVaR`-based baselines. Color intensity represents the cumulative number of visits per grid cell across all episodes and seeds (darker = more frequent). EVAR-SA exhibits concentrated visitation along safe paths, avoiding high-risk regions (cliff cells and wind-affected upper rows). In contrast, `CVaR`-PG and D4PG-CVaR display broader dispersal, indicating greater exposure to tail risks. SDPG-CVaR exhibits intermediate behavior. These results highlight EVAR-SA's capacity for robust tail-risk mitigation in stochastic environments.

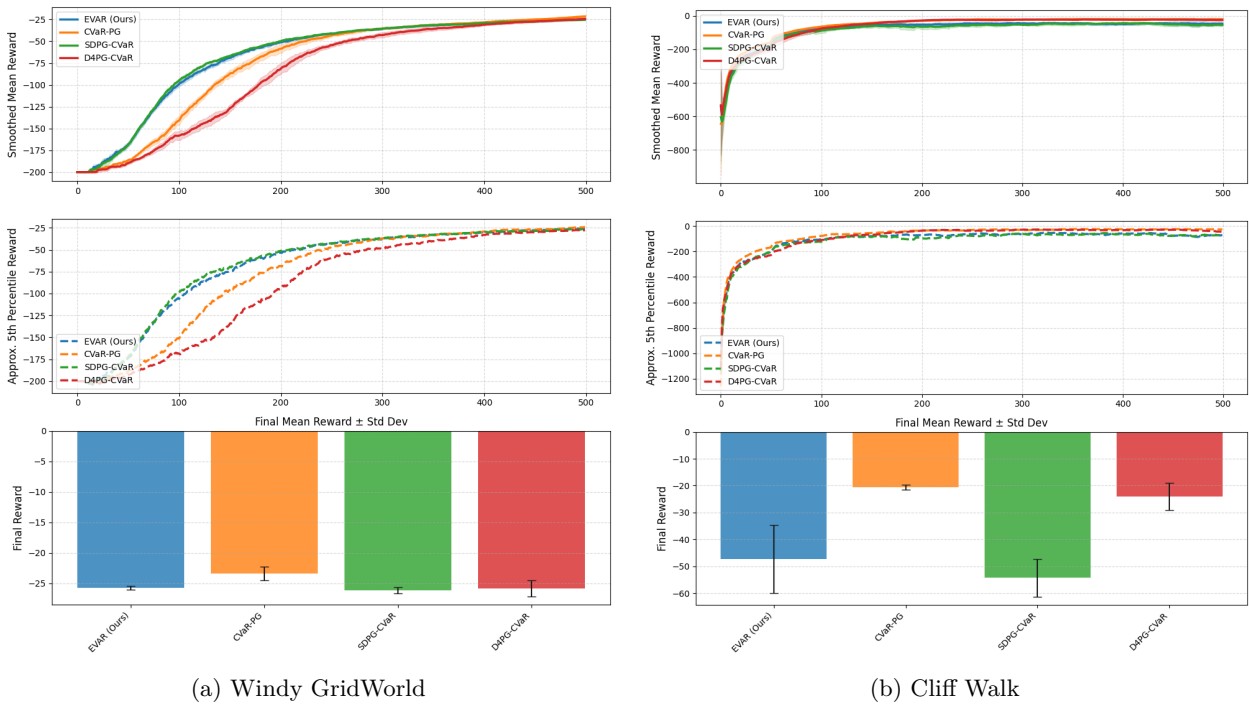

Figure 19: Performance comparison across environments. (i) smoothed mean episode reward with $\pm 1\sigma$ confidence bands, (ii) approximate 5th-percentile return trajectories, and (iii) bar charts of final mean reward $\pm$ standard deviation, for EVAR-SA and three CVaR-based baselines over 500 episodes and 8 seeds. EVAR-SA converges faster with tighter worst-case guarantees in both Windy GridWorld (left) and Cliff Walk (right).

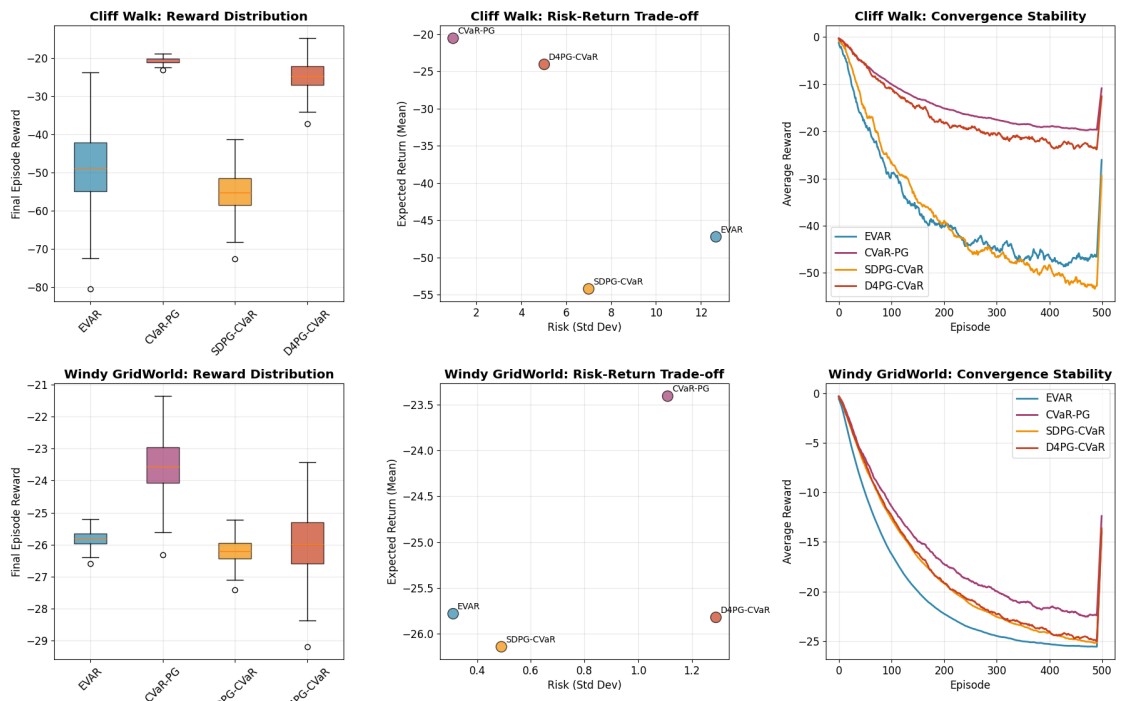

Figure 20: Risk-seeking analysis. (*Left*) Box-and-whisker plots of final episode rewards over 8 seeds: EVAR-SA achieves a tighter distribution and higher returns than CVaR-PG, SDPG-CVaR, and D4PG-CVaR. (*Middle*) Risk–return scatter: EVAR-SA attains lower tail-risk (std. dev.) for a given mean reward. (*Right*) Convergence stability: EVAR-SA's finite-difference SA estimator yields faster, smoother learning with narrower $\pm\sigma$ bands.

EVAR-SA's finite-difference stochastic-approximation gradients produce smooth, monotonic improvement and tighter $\pm\sigma$ confidence bands, unlike the oscillations observed in quantile-based CVaR updates.

### 3.4 Glycemic Control

We demonstrate our algorithm's ability to manage high-risk insulin administration for Type-1 Diabetes Mellitus (T1DM) using the Simglucose simulator (Xie, 2018), which evaluates `EVaR`'s efficacy in safety-critical RL. Simglucose mimics real-world scenarios, providing a controlled environment to test control algorithms before clinical deployment. In our experiments, a PID controller regulates insulin based on blood glucose levels, aiming to keep them within a safe range. We evaluate performance on both adult and adolescent patient profiles to minimize the risk of hyper- and hypoglycemia, illustrated in Figure 21. The glycemic control task typically uses a highly nonlinear reward function to reflect clinical risk. Small deviations from the normoglycemic range might incur mild penalties, but crossing critical glucose thresholds triggers disproportionately large negative rewards (e.g., a penalty spike or episode termination when glucose $< 70$ mg/dL or $> 250$ mg/dL for a sustained period). This nonlinear penalty structure creates the heavy-tailed return distribution. Glycemic control is a high-stakes problem where inaction or overly conservative actions can lead to severe health risks such as prolonged hyperglycemia. A risk-averse policy (SAC) favors safe, incremental insulin adjustments, but this could result in suboptimal glucose regulation. This is illustrated in Figure 21. The `EVaR`-optimized insulin policy keeps the patient within "admissible levels of risk", quickly correcting course whenever glucose breaches hypo- or hyperglycemic thresholds. In other words, if blood glucose starts trending to a red zone, the `EVaR` policy will take bold corrective insulin actions to bring it back to safe levels without prolonged exposure to risk. A single trajectory where blood glucose goes to a life-threatening level can dramatically decrease $\mathbb{E}[e^{\beta R}]$ (for some $\beta > 0$) and thus lower the `EVaR` metric. The agent learns to steer away from such outcomes because they are catastrophically bad under the entropic risk measure.

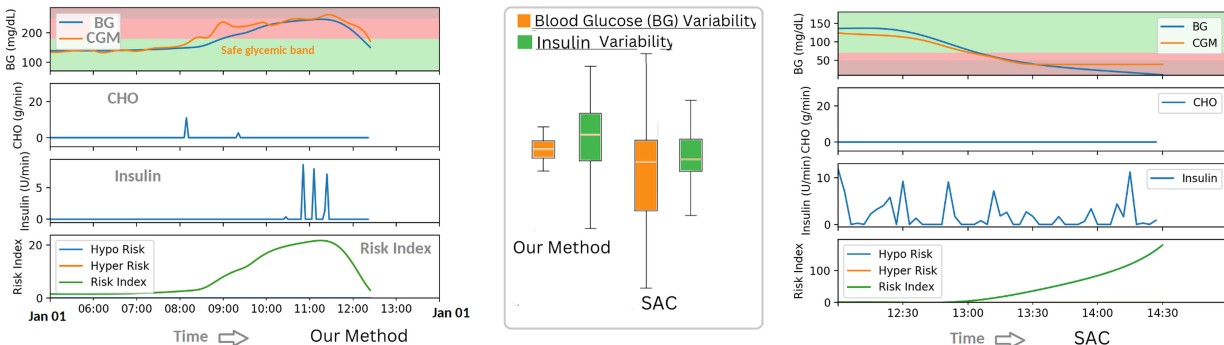

Figure 21: Comparison of our proposed method with SAC (Haarnoja et al., 2018), demonstrating reduced variability in blood glucose levels and more robust insulin administration. We observe that when blood glucose reaches risky levels (red), our method effectively course-corrects insulin administration without prolonged exposure to risk, ensuring better patient stability.

## 3.5 Portfolio Optimization

The portfolio optimization problem seeks an optimal portfolio allocation among $N$ assets by maximizing the `EVaR` of the portfolio returns $\mathcal{R}$, which captures the upside tail of the return distribution. Here, policy represents the action chosen, which includes sell, buy, or hold. Constraints are kept to ensure that the portfolio weights $w_i$ are nonnegative and sum to one, representing a fully invested portfolio. For our portfolio (top 10 stocks of DJIA), weights $\mathbf{w} \in \mathbb{R}^{10}$ are constrained such that $\sum_{i=1}^{10} w_i = 1$, $w_i \geqslant 0 \forall i$. A constant transaction cost of 0.1% is applied, computed as - Cost $= 0.001 \times \sum_{i=1}^{10} \left| w_i^{\text{new}} - w_i^{\text{old}} \right| \times$ Portfolio Value. In Table 4 we provide the backtesting (Wong, 2010) results of our `EVaR` strategy compared against other risk seeking strategies as follows:

| Portfolio | Cum Ret | Exp Ret | Ann Vol | Sharpe |
|---|---|---|---|---|
| EVaR | 265.7% | 13.3% | 15.0% | 0.75 |
| VaR | 250.0% | 12.6% | 14.1% | 0.75 |
| CVaR | 257.5% | 12.8% | 10.2% | 0.99 |

Table 4: Portfolio backtest summary

Over 7 years (DJIA top-10 stocks, 1970–1977), an `EVaR`-optimized dynamic strategy achieved the highest terminal wealth and growth rate. Its cumulative return was +265.7%, outperforming both a `CVaR`-optimized strategy (+257.5%) and a `VaR`-optimized strategy (+250.0%) over the same period. In annualized terms, this corresponds to an average return of 13.3% for `EVaR`, versus 12.8% for `CVaR` and 12.6% for `VaR`. This superior long-run performance aligns with the intuition that `EVaR`'s objective lets the agent capture more upside within a certain entropy distance. Notably, `EVaR` did accept slightly higher volatility to achieve those gains: the `EVaR` portfolio's annualized volatility was 15.0%, a bit above the `VaR`-based portfolio (14.1%) and higher than the very low volatility of the `CVaR` portfolio (10.2%). In risk-adjusted terms (Sharpe ratio), the `CVaR` strategy had the highest Sharpe $\approx 0.99$ by virtue of its tight risk control (essentially sacrificing return to minimize variance), whereas `EVaR` and `VaR` both came in around Sharpe $\approx 0.75$. This indicates the `EVaR` agent deliberately took on extra volatility – consistent with a "risk-seeking" approach – but translated that risk into higher return so that its Sharpe remained on par with the `VaR` strategy and quite noteworthy in absolute terms. In other words, `EVaR` maintained a middle ground on the efficient frontier (Figure 27): it did not maximize Sharpe ratio (as `CVaR`'s extremely cautious approach did), but it achieved a markedly better growth rate for only a moderate increase in volatility. Indeed, the efficient frontier analysis shows that the `EVaR`-optimal portfolio yields an excellent risk–return balance, essentially maximizing return for a given

downside risk level. By identifying portfolios that "maximize returns while minimizing downside risk", the `EVaR` frontier dominates what `VaR` or `CVaR` alone can achieve.

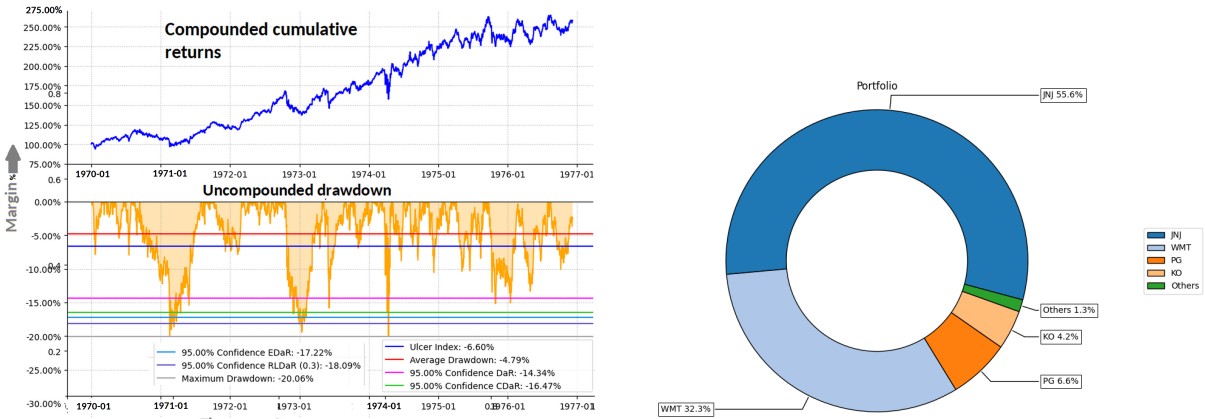

Figure 22: [**Left**:] Drawdown analysis of our portfolio. `EVaR`-optimized portfolio advances on a steadily rising equity curve while absorbing macro shocks and keeping peak-to-trough losses shallow and short-lived. [**Right:**] Portfolio allocation according to the optimal `EVaR` policy.

Also, since the `EVaR` objective multiplies every daily loss by an exponential weight, even a handful of large negative returns pushes the risk metric down sharply. This triggers the optimizer to reduce its exposure to high-risk holdings before a market slide deepens, so peak-to-trough declines stay small ($\approx 4\%$ in the 1973-74 crash, versus 48% for the index). When conditions improve, the same exponential tilt captures the rebound rally, guiding the portfolio back into higher-beta assets and restoring its previous high in a few weeks. This is illustrated in Figure 22, where the equity curve rises steadily while any dips are both shallow and quickly recovered.

### 3.6 Sensitivity to Learning rate

To analyze hyper-parameter sensitivity, we consider an episodic, two–action MDP with $|S|$=50 states arranged on a ring. Action 0 (*safe*) keeps the agent in place and yields a deterministic reward 0.5; action 1 (*risky*) moves one step clockwise and delivers a heavy–tailed reward distributed as Pareto($k$=1, $b$=2). Episodes last $H = 5$ time steps with discount $\gamma = 0.95$. The policy is a one–parameter Bernoulli, $\Pr(a_t=1) = \sigma(\theta)$, so the return distribution is regularly varying, which stresses the `EVaR` objective.

With the baseline hyperparameters, the bias decays like $t^{-0.20}$, the drift remains at $1/8$, and the variance of the stochastic gradient contracts as $t^{-0.60}$, bringing $(\theta_t, x_t)$ close to their limits by iteration 50. Increasing the outer step–size accelerates early progress but leaves a wider asymptotic band because the noise term $a_t^2$ Var dominates more slowly. A steeper perturbation decay eliminates bias faster yet stalls convergence once the finite–difference signal falls below the simulation noise floor. Growing the inner batch length suppresses the drift from $10^{-1}$ to $10^{-2}$ in fewer than twenty iterations, yielding the tightest bands at the expense of a quadratic increase in sample complexity. These trends corroborate theoretical predictions: $a_t$ sets the speed–variance trade-off, $c_t$ governs asymptotic bias, and $N_t$ controls late–stage variance once the iterate enters the `EVaR` basin of attraction.

We further analyze the impact of learning rates on algorithm performance using *InvertedPendulumDouble-v4*. We evaluate learning rates $0.01, 0.1, 0.2,$ and $0.5$ by averaging results over 5 independent batches, each with 50 episodes of 500 steps. The optimizer runs for 500 iterations with the risk parameter $\alpha = 0.1$. Using ADAM(Kingma, 2014) improves iterate stability, with $10^{-2}$ yielding the best performance. In this context of our gradient estimator with ADAM, our learning rate parameter update rule is:

$$a_{t+1} = a_t - \psi \cdot \frac{\hat{m}_t}{\sqrt{\hat{v}_t} + \epsilon}$$

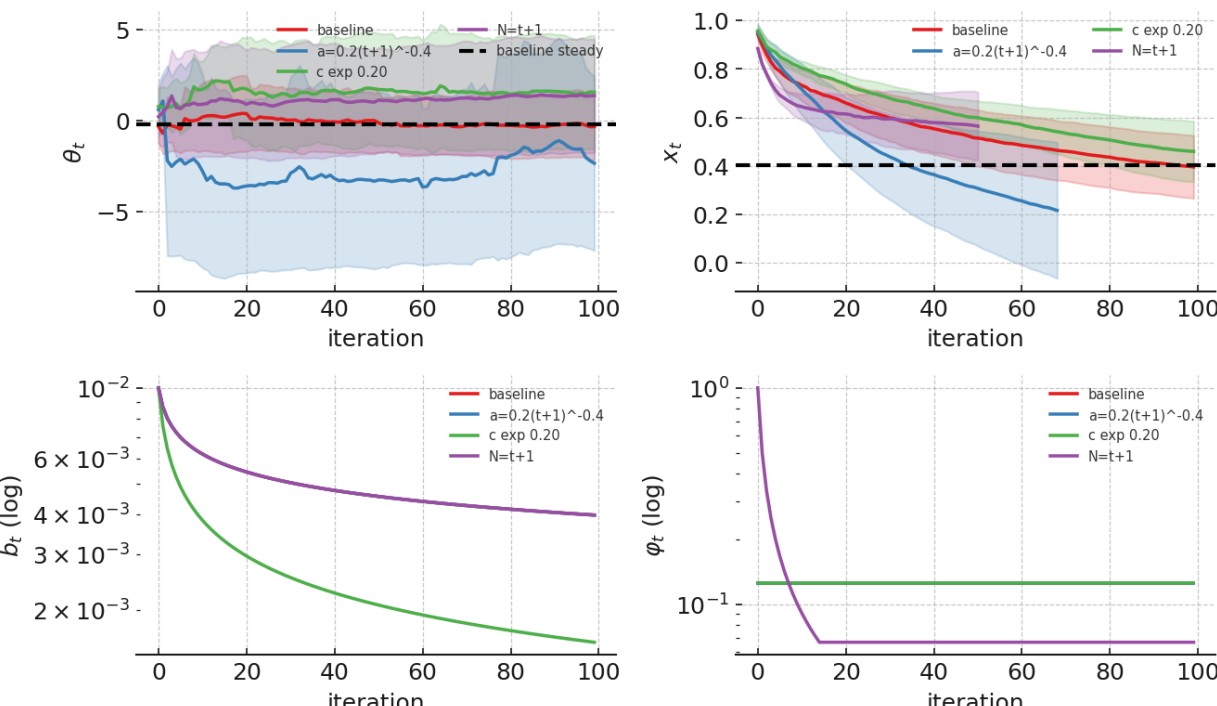

Figure 23: **Parameter sensitivity.** The figure shows the evolution of the policy parameter $\theta_t$, the `EVaR` shape parameter $x_t$, the finite–difference bias $b_t$ (`log axis`), and the drift term $\varphi_t$ on a 50-state heavy-tailed MDP. The *baseline* schedule $\left[ a_t = 0.10(t+1)^{-0.6}, c_t = 0.10(t+1)^{-0.10}, \delta_t = \xi_t = 0.05(t+1)^{-0.6}, N_t = 8 \right]$ is compared with three single–factor variants: (i) a larger outer step–size $a_t = 0.20(t+1)^{-0.4}$, (ii) a faster perturbation decay $c_t \propto (t+1)^{-0.20}$, and (iii) a linearly growing inner batch $N_t = \min\{t+1, 15\}$. Solid lines show the mean over six i.i.d. runs; shaded envelopes indicate $\pm 1$ standard deviation.

where: $\hat{m}_t$ is the bias-corrected first moment estimate and $\hat{v}_t$ is the bias-corrected second moment estimate.

As learning rate sensitivity affects both convergence and variance, a higher sensitivity can lead to faster initial convergence but may impact long-term stability. Very high sensitivity can increase the upper bound on variance, potentially leading to less stable convergence. The adaptive nature of ADAM helps mitigate these effects by adjusting the effective learning rate based on the moments of the gradients. From Fig. 24(a), which depicts the movement of the iterates, and Fig.24(b,) the expected, it is evident that the introduction of an adaptive learning schedule for the gradient estimator of `EVaR` controls the rapid movement of the iterates and is resilient against environment dynamics. When compared against the non-adaptive case, Fig. 24(c) and (d), we clearly see increased movement as the initial learning rate decreases, depicting high susceptibility to the initial choice of the learning rate.

## 4 Conclusion

In this paper, we introduce a novel multi-timescale stochastic approximation algorithm for risk-seeking reinforcement learning, optimizing the Entropic Value at Risk (`EVaR`) objective that provably converges. `EVaR`, a coherent risk measure derived from exponential tail bounds, enables agents to prioritize high-reward trajectories while managing tail risk through a Kullback-Leibler divergence constraint and thus provides a tighter control on tails. By employing a randomly perturbed finite difference approximation, we seek the optimal `EVaR` policy. Across grid navigation, MuJoCo locomotion, glycaemic regulation, and dynamic portfolio allocation, the resulting policies consistently achieved competitive performance, limiting worst-case drawdowns, yet capturing larger upside returns. However, four practical challenges remain: (1) performance is sensitive to the confidence level $\alpha$ and inner-loop batch size; (2) the perturbation-based gradient estimator

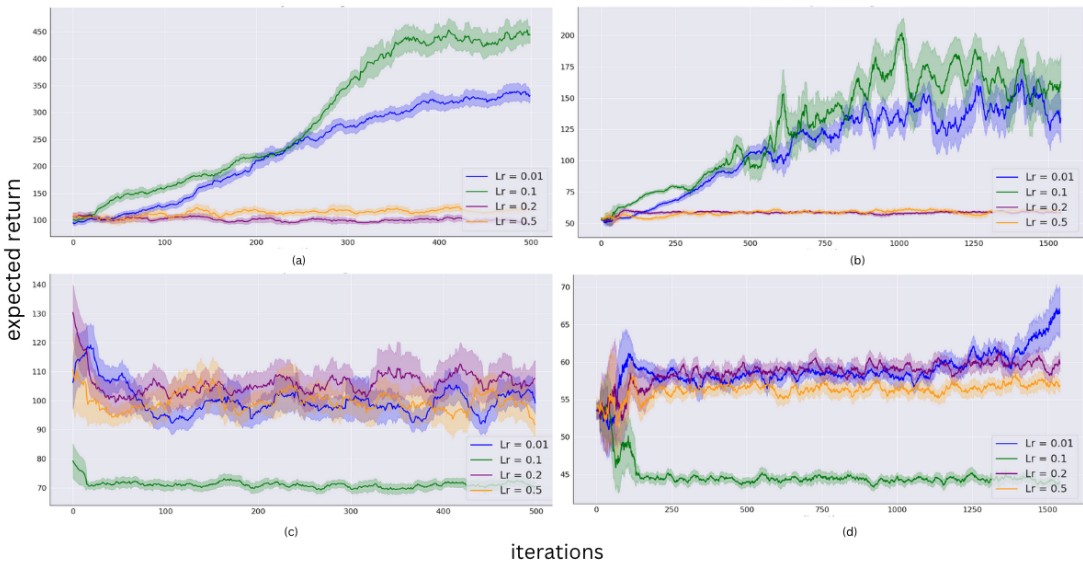

Figure 24: Sensitivity of the optimizer to the learning rate placed at $[0.01, 0.1, 0.2, 0.5]$ with ADAM for the top row for $J_{\texttt{EVaR}}$ perturbation (*a*) and the expected returns (*b*) and similarly, the bottom row shows the similar setup without ADAM in (*c*) and (*d*).

for `EVaR` requires twice the on-policy trajectories per update, limiting sample efficiency and preventing replay buffer reuse; (3) numerical instability arising from exponential weighting in the `EVaR` objective; and (4) `EVaR` objective may be ill-posed if the moment-generating function of the returns does not exist (e.g. power-law tails). To address these, promising directions include: adaptive schedules for $\alpha$ and batch size to balance bias-variance trade-offs automatically; off-policy corrections (e.g., importance-weighted critics) to enable gradient estimation from cached data, reducing simulation costs; numerical stabilization via reward scaling or log-domain arithmetic (e.g., log-sum-exp); and truncation of the return distribution.

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

## A Proofs

Consider the filtration $\mathcal{F}_t = \sigma\{\vartheta_t, J, \omega_t, \beta_t, \theta_t\}$. We prove Theorem 4 in two parts.

### A.1 Proof of Theorem 2

**Proof of Theorem 2: Part I**

*Proof.* Consider the recursion

$$
\begin{aligned}
\vartheta_{t+1} &= \vartheta_t + \delta_t \left( e^{R(\tau)/x} - \vartheta_t \right) \\
&= \vartheta_t + \delta_t \left( \mathbb{E}\left[ e^{R(\tau)/x} \right] + e^{R(\tau)/x} - \mathbb{E}\left[ e^{R(\tau)/x} \right] - \vartheta_t \right) \\
&= \vartheta_t + \delta_t \left( h_1(\vartheta_t) + \mathbb{M}_{t+1} \right), \\
&\text{where } h_1(\vartheta) = \mathbb{E}\left[ e^{R(\tau)/x} \right] - \vartheta, \text{ and } \mathbb{M}_{t+1} = e^{R(\tau)/x} - \mathbb{E}\left[ e^{R(\tau)/x} \right]
\end{aligned}
\tag{25}
$$

It is easy to verify the $\{\mathbb{M}_t\}$ is a martingale difference adapted to the filtration $\{\mathcal{F}_t\}$ *i.e.*, $\mathbb{M}_t$ is $\mathcal{F}_t$-measurable and $\mathbb{E}\left[\mathbb{M}_{t+1}|\mathbb{F}_t\right] = 0$ *a.s.* Also, note that $h_1 : \mathbb{R} \to \mathbb{R}$ is Lipschitz continuous. Further, by Borkar-Meyn Theorem (Theorem 7, Chapter 3 of Borkar (2009)), one can show that $\sup_t |\vartheta_t| < \infty$ *a.s.* Hence, by Theorem 1 of Chapter 2 of Borkar (2009), the iterates $\{\vartheta_t\}$ asymptotically tracks the following ODE:

$$
\dot{\vartheta} = \mathbb{E}_{\tau \sim \pi_\theta} \left[ e^{R(\tau)/x} \right] - \vartheta.
\tag{26}
$$

Since $h_1$ is linear, we conclude that

$$
\lim_{t \to \infty} \vartheta_t = \{\vartheta^* | h(\vartheta^*) = 0\} \Rightarrow \lim_{t \to \infty} \vartheta_t = \mathbb{E}_{\tau \sim \pi_\theta} \left[ e^{\beta R(\tau)} \right].
\tag{27}
$$

Now consider the stochastic recursion

$$
\begin{aligned}
\omega_{t+1} &= \omega_t + \delta_t \left( R(\tau) e^{\beta R(\tau)} - \omega_t e^{R(\tau)/x} \right) \\
&= \omega_t + \delta_t \left( R(\tau) e^{R(\tau)/x} - \omega_t e^{R(\tau)/x} - \mathbb{E}\left[ R(\tau) e^{R(\tau)/x} - \omega_t e^{R(\tau)/x} \right] + \mathbb{E}\left[ R(\tau) e^{R(\tau)/x} - \omega_t e^{R(\tau)/x} \right] \right)
\end{aligned}
$$

Rewriting the above equation, we get

$$
\omega_{t+1} = \omega_t + \delta_t \left( h_\omega(\omega_t) + \mathbb{M}_{t+1}^\omega \right),
\tag{28}
$$

$$
\text{where } h^\omega(\omega) = \mathbb{E}_\tau \left[ R(\tau) e^{R(\tau)/x} - \omega e^{R(\tau)/x} \right] \text{ and}
\tag{29}
$$

$$
\mathbb{M}_{t+1}^\omega = R(\tau) e^{R(\tau)/x} - \omega_t e^{R(\tau)/x} - \mathbb{E}\left[ R(\tau) e^{R(\tau)/x} - \omega_t e^{R(\tau)/x} \Big| \mathcal{F}_t \right].
\tag{30}
$$

Now,

$$
\begin{aligned}
\mathbb{E}\left[\mathbb{M}_{t+1}^\omega | \mathcal{F}_t\right] &= \mathbb{E}\left[ R(\tau) e^{R(\tau)/x} - \omega_t e^{R(\tau)/x} - \mathbb{E}\left[ R(\tau) e^{R(\tau)/x} - \omega_t e^{R(\tau)/x} \Big| \mathcal{F}_t \right] \Big| \mathcal{F}_t \right] \\
&= \mathbb{E}\left[ R(\tau) e^{R(\tau)/x} - \omega_t e^{R(\tau)/x} \Big| \mathcal{F}_t \right] - \mathbb{E}\left[ R(\tau) e^{R(\tau)/x} - \omega_t e^{\beta R(\tau)} \Big| \mathcal{F}_t \right] \Big| \mathcal{F}_t \right] = 0
\end{aligned}
$$

Hence $\{\mathbb{M}_t^\omega\}$ is a Martingale-difference noise adapted to the filtration $\{\mathcal{F}_t\}$. Also since $\gamma \in [0,1)$ and we have $|R(\tau)| < \frac{R_\infty}{1-\gamma}$. Hence,

$$\exists K_w > 0 \ s.t. \ \mathbb{E}\left[|\mathbb{M}_t^\omega|^2|\mathcal{F}_t\right] < (1 + K_w)(|\omega_t|)^2. \tag{31}$$

Also, for $\omega_1, \omega_2 \in \mathbb{R}$, we have

$$|h^\omega(\omega_1) - h^\omega(\omega_2)| = \mathbb{E}_\tau\left[e^{R(\tau)/x}\right]|\omega_1 - \omega_2|$$
$$\leqslant \exp\left\{\left(\frac{R_\infty}{x(1-\gamma)}\right)\right\}|\omega_1 - \omega_2|.$$

Hence $h^\omega$ is Lipschitz continuous.

Now, we will show that the iterates $\theta_t$ are stable, i.e., $\sup_t |\omega_t| < \infty \quad a.s.$ Hence, we consider the following scaled functions

$$h_c(\omega) = \frac{h^\omega(c\omega)}{c}, c > 0. \tag{32}$$

Now consider the $\infty$-ODE given by

$$\dot{\omega} = h_\infty^\omega(\omega). \tag{33}$$

where

$$h_\infty^\omega(\omega) = \lim_{c \to \infty} h_c^\omega(\omega) = \lim_{c \to \infty} \frac{h(c\omega)}{c} = \lim_{c \to \infty} \frac{1}{c}\left(\mathbb{E}_\tau\left[R(\tau)e^{R(\tau)/x} - c\omega e^{R(\tau)/x}\right]\right)$$
$$= -\omega\mathbb{E}_\tau\left[e^{R(\tau)/x}\right] \tag{34}$$

Hence ODE (33) becomes

$$\dot{\omega} = -U\omega, \text{ where } U = \mathbb{E}_\tau\left[e^{R(\tau)/x}\right]. \tag{35}$$

Note that since $\beta > 0$, we have $U > 0$. Hence, the $\infty$-ODE given above has a unique globally asymptotically stable equilibrium point. Hence, by Borkar-Meyn Theorem (Theorem 7, Chapter 3 of Borkar (2009)), we have

$$\sup_t |\omega_t| < \infty \ a.s. \tag{36}$$

Now by Theorem 1 of Chapter 2 of (Borkar, 2009), the sequence $(\omega_t^\theta)$ converges to a compact connected internally chain transitive invariant set of the ODE given by

$$\dot{\omega} = h^\omega(\omega). \tag{37}$$

Since $h^\omega$ is a linear function, the only compact connected internally chain transitive invariant set is $\{\omega|h^\omega(\omega) = 0\}$. Hence

$$h^\omega(\omega) = 0 \Rightarrow \mathbb{E}_\tau\left[R(\tau)e^{R(\tau)/x}\right] - \omega\mathbb{E}_\tau\left[e^{R(\tau)/x}\right] = 0$$
$$\Rightarrow \omega = \frac{\mathbb{E}_\tau\left[R(\tau)e^{R(\tau)/x}\right]}{\mathbb{E}_\tau\left[e^{R(\tau)/x}\right]}$$

Therefore

$$\lim_{t \to \infty} \omega_t = \frac{\mathbb{E}_\tau\left[R(\tau)e^{R(\tau)/x}\right]}{\mathbb{E}_\tau\left[e^{R(\tau)/x}\right]} \quad a.s. \tag{38}$$

$\square$

**Lemma 3.** *The function $\vartheta^*(x,\theta) = \mathbb{E}_{\tau \sim \pi_\theta}\left[e^{R(\tau)/x}\right]$ is Lipschitz continuous on $x \in [x_{min}, x_{max}]$, i.e., $\exists L_\vartheta > 0$ such that:*

$$|\vartheta^*(x_1,\theta) - \vartheta^*(x_2,\theta)| \leqslant L_\vartheta |x_1 - x_2|, \quad \forall x_1, x_2 \in [x_{min}, x_{max}].$$

*Proof.* We compute the derivative of $\vartheta^*(x,\theta)$ with respect tot $x$ as follows:

$$\frac{d}{dx}\vartheta^*(x,\theta) = \mathbb{E}_\tau\left[-\frac{R(\tau)}{x^2}e^{R(\tau)/x}\right].$$

Since $|R| \leqslant \frac{R_\infty}{1-\gamma}$ and $x \geqslant x_{min}$, we have:

$$\left|\frac{d}{dx}\vartheta^*(x,\theta)\right| \leqslant \frac{R_\infty}{(1-\gamma)x_{min}^2}\mathbb{E}_\tau\left[e^{R(\tau)/x}\right] \leqslant \frac{R_\infty}{(1-\gamma)x_{min}^2}e^{\frac{R_\infty}{(1-\gamma)x_{min}}}.$$

Let $L_\vartheta = \frac{R_\infty}{(1-\gamma)x_{min}^2}e^{\frac{R_\infty}{(1-\gamma)x_{min}}}$. By the Mean Value Theorem, $\forall x_1, x_2 \in [x_{min}, x_{\max}]$:

$$|\vartheta^*(x_1,\theta) - \vartheta^*(x_2,\theta)| \leqslant L_\vartheta |x_1 - x_2|.$$

Hence, $\vartheta^*(x,\theta)$ is Lipschitz continuous in $x$. $\qquad\square$

**Lemma 4.** *The function $\omega^*(x,\theta) = \frac{\mathbb{E}_{\tau \sim \pi_\theta}\left[R(\tau)e^{R(\tau)/x}\right]}{x\mathbb{E}_\tau\left[e^{R(\tau)/x}\right]}$ is Lipschitz continuous on $x \in [x_{min}, x_{max}]$, i.e., there exists $L_\omega > 0$ such that:*

$$|\omega^*(x_1,\theta) - \omega^*(x_2,\theta)| \leqslant L_\omega |x_1 - x_2|, \quad \forall x_1, x_2 \in [x_{min}, x_{max}].$$

*Proof.* Let $N(x) = \mathbb{E}_\tau[R(\tau)e^{R(\tau)/x}]$ and $D(x) = x\mathbb{E}_\tau[e^{R(\tau)/x}]$. Then: $\omega^*(x,\theta) = \frac{N(x)}{D(x)}$. Their derivatives are:

$$N'(x) = -\frac{1}{x^2}\mathbb{E}_\tau[R(\tau)^2 e^{R(\tau)/x}], \quad D'(x) = \mathbb{E}_\tau[e^{R(\tau)/x}] - \frac{1}{x}\mathbb{E}_\tau[R(\tau)e^{R(\tau)/x}].$$

Using the quotient rule, we obtain:

$$\frac{d}{dx}\omega^*(x,\theta) = \frac{N'(x)D(x) - N(x)D'(x)}{D(x)^2}.$$

Bounding each term ($|R(\tau)| \leqslant \frac{R_\infty}{(1-\gamma)}$, $x \geqslant x_{min}$) from above as follows:

$$|N'(x)| \leqslant \frac{R_\infty^2}{(1-\gamma)^2 x_{min}^2}e^{\frac{R_\infty}{(1-\gamma)x_{min}}}$$
$$|D(x)| \geqslant x_{min}e^{-C/a},$$
$$|D'(x)| \leqslant e^{\frac{R_\infty}{(1-\gamma)x_{min}}} + \frac{R_\infty}{(1-\gamma)x_{min}}e^{\frac{R_\infty}{(1-\gamma)x_{min}}},$$
$$|N(x)| \leqslant \frac{R_\infty}{(1-\gamma)}e^{\frac{R_\infty}{(1-\gamma)x_{min}}}.$$

Substituting these bounds, we get:

$$\left|\frac{d}{dx}\omega^*(x,\theta)\right| \leqslant \left(\frac{R_\infty^2}{(1-\gamma)^2 x_{min}^2}e^{\frac{2R_\infty}{(1-\gamma)x_{min}}} + \frac{R_\infty}{(1-\gamma)}e^{\frac{2R_\infty}{(1-\gamma)x_{min}}}\left(1 + \frac{R_\infty}{(1-\gamma)x_{min}}\right)\right)x_{min}^{-2}e^{\frac{2R_\infty}{(1-\gamma)x_{min}}} \triangleq L_\omega.$$

Thus $|\omega^*(x_1,\theta) - \omega^*(x_2,\theta)| \leqslant L_\omega |x_1 - x_2|$. $\qquad\square$

**Lemma 5.** *The first order derivative $G'(x) = \log\frac{\vartheta^*(x,\theta)}{\alpha} - \omega^*(x,\theta)$ is Lipschitz continuous on $x \in [x_{min}, x_{max}]$, i.e., $\exists L_{G'} > 0$ such that:*

$$\left|G'(x_1) - G'(x_2)\right| \leqslant L_{G'}|x_1 - x_2|, \quad \forall x_1, x_2 \in [x_{min}, x_{max}].$$

*Proof.* Consider the derivative:

$$\frac{d}{dx}\left(\log\frac{\vartheta^*(x,\theta)}{\alpha}\right) = \frac{\vartheta^*(x,\theta)'}{\vartheta^*(x,\theta)}.$$

Using Lemma 3, we obtain:

$$\left|\frac{d}{dx}\log\frac{\vartheta^*(x,\theta)}{\alpha}\right| \leqslant L_\vartheta e^{\frac{R_\infty}{x_{min}(1-\gamma)}}.$$

From Lemma 4, $\omega^*(x,\theta)$ has Lipschitz constant $L_\omega$. Therefore:

$$|G'(x_1) - G'(x_2)| \leqslant \left(L_\vartheta e^{\frac{R_\infty}{x_{min}(1-\gamma)}} + L_\omega\right)|x_1 - x_2| \triangleq L_G|x_1 - x_2|.$$

□

**Proof of Theorem 2 Part II:**

*Proof.* On the slower timescale, the $x$–update approximates the ODE:

$$\dot{x} = -G'(x) = -\left(\log\frac{\mathbb{E}_{\tau\sim\pi_\theta}\left[e^{R(\tau)/x}\right]}{\alpha} - \frac{\mathbb{E}_{\tau\sim\pi_\theta}\left[R(\tau)e^{R(\tau)/x}\right]}{x\,\mathbb{E}_{\tau\sim\pi_\theta}\left[e^{R(\tau)/x}\right]}\right).$$

By Theorem 1, $G(x)$ is $m$–strongly convex. Using $G(x)$ as a Lyapunov function:

$$\frac{dG}{dt} = G'(x)\,\dot{x} = -\left(G'(x)\right)^2 \leqslant 0,$$

with equality only at $x = x^*(\theta)$. By LaSalle's invariance principle,

$$x_t \rightarrow x^*(\theta) \quad \text{almost surely.}$$

Now, regarding the joint convergence, note that the time–scale separation ensures

$$\vartheta_t \rightarrow \vartheta^*(x_t,\theta) \quad \text{and} \quad \omega_t \rightarrow \omega^*(x_t,\theta)$$

before $x_t$ updates significantly. Thus, the system converges to

$$(\vartheta^*(x^*(\theta),\theta),\ \omega^*(x^*(\theta),\theta),\ x^*(\theta)) \quad a.s.$$

□

## A.2   Proof of Theorem 3

We prove each part of Theorem 3 as individual lemmas here.

**Lemma 6.** *Let $\delta_t = \frac{c}{t^r}, r \in (frac12, 1)$ with $c > \frac{r}{4}$. Then for any fixed $x > 0$,*

$$\mathbb{E}\left[|\vartheta_t - \vartheta^*(x,\theta)|^2\right] \leqslant \frac{K_1}{t^r},\ \text{ with } K_1 > 0, \forall t \geqslant 1.$$

*Proof.* Let the error be $\varepsilon_t^\vartheta := \vartheta_t - \vartheta^*(x,\theta)$, where $\vartheta^*(x,\theta) = \mathbb{E}_{\tau\sim\pi_\theta}[e^{R(\tau)/x}]$. The update rule becomes:

$$\varepsilon_{t+1}^\vartheta = (1-\delta_t)\varepsilon_t^\vartheta + \delta_t\eta_{t+1}, \tag{39}$$

where $\eta_{t+1} = e^{R(\tau_{t+1})/x} - \vartheta^*(x,\theta)$ is a martingale difference sequence.

Square both sides and take conditional expectations:

$$(\varepsilon_{t+1}^\vartheta)^2 = (1-\delta_t)^2(\varepsilon_t^\vartheta)^2 + \delta_t^2\eta_{t+1}^2 + 2(1-\delta_t)\delta_t\,\varepsilon_t^\vartheta\eta_{t+1},$$

$$\mathbb{E}\big[(\varepsilon_{t+1}^\vartheta)^2 \mid \mathcal{F}_t\big] = (1-\delta_t)^2(\varepsilon_t^\vartheta)^2 + \delta_t^2\,\mathbb{E}[\eta_{t+1}^2 \mid \mathcal{F}_t],$$

since $\mathbb{E}[\eta_{t+1}|\mathcal{F}_{t+1}] = 0$ and $\mathbb{E}[\eta_{t+1}^2] \leqslant e^{\frac{2R_\infty}{(1-\gamma)x_{min}}} = \nu^2$.

Taking total expectations, we get

$$\mathbb{E}[(\varepsilon_{t+1}^\vartheta)^2] \leqslant (1-\delta_t)^2 \mathbb{E}[(\varepsilon_t^\vartheta)^2] + \delta_2 \nu^2. \tag{40}$$

Now define $v_t = t^r \mathbb{E}[(\epsilon_t^\vartheta)^2]$. Hence, from Eq.(40),

$$\begin{aligned}
v_{t+1} &= (t+1)^r \mathbb{E}[(\epsilon_{t+1}^\vartheta)^2] \\
&\leqslant (t+1)^r \big[(1-\delta_t)^2 \mathbb{E}[(\epsilon_t^\vartheta)^2] + \nu^2 \delta_t^2\big] \\
&= (t+1)^r (1-\delta_t)^2 \frac{v_t}{t^r} + \nu^2 (t+1)^r \delta_t^2.
\end{aligned} \tag{41}$$

Take $\delta_t = \frac{c}{t^r}, r \in (0.5, 1)$. Also, using the binomial expansion $(t+1)^r = t^r\big(1 + \frac{r}{t} + O(t^{-2})\big)$ and $(1-\delta_t)^2 = 1 - 2\delta_t + \delta_t^2 = 1 - 2ct^{-r} + c^2 t^{-2r}$,

$$(t+1)^r (1-\delta_t)^2 t^{-r} = \underbrace{1 - 2ct^{-r} + \frac{r}{t} + O(t^{-2r})}_{A_t}. \tag{42}$$

Because $r > 0.5$, $t^{-r} \gg t^{-1}$, hence for all $t \geqslant T_1 = \left(\frac{2r}{2c-r/2}\right)^{1/(1-r)}$

$$A_t \leqslant 1 - \kappa t^{-r}, \qquad \kappa = 2c - \frac{r}{2} > 0 \text{ (since } c > \frac{r}{4}). \tag{43}$$

For the additional term in Eq.(41), we have

$$(t+1)^r \delta_t^2 = c^2\big(1 + \frac{r}{t} + O(t^{-2})\big)t^{-r} \leqslant 2c^2 t^{-r} \quad (t \geqslant 2). \tag{44}$$

Set $T_0 = \max\{T_1, 2\}$. Then, from Eqs.(42), (43) and (44), we have for $t \geqslant T_0$,

$$v_{t+1} \leqslant (1 - \kappa t^{-r})v_t + 2\nu^2 c^2 t^{-r}. \tag{45}$$

Now, let $w_t = v_t + C$, where $C = \frac{2\nu^2 c^2}{\kappa}$. Then,

$$\begin{aligned}
v_{t+1} + C &\leqslant \big(1 - \kappa t^{-r}\big) v_t + C + 2\nu^2 c^2 t^{-r} \\
&= \big(1 - \kappa t^{-r}\big) v_t + C + \kappa C t^{-r} \quad \big(\text{since } \kappa C = 2\nu^2 c^2\big) \\
&= v_t + C - \kappa t^{-r} (v_t - C) \\
&\leqslant v_t + C - \kappa t^{-r} v_t = w_t - \kappa t^{-r} v_t.
\end{aligned} \tag{46}$$

If $v_t \geqslant C$, then from (46), we have

$$v_{t+1} + C \leqslant v_t + C \implies w_{t+1} \leqslant w_t. \tag{47}$$

Otherwise $(v_t < C \implies w_t \leqslant 2C)$,

$$\begin{aligned}
w_{t+1} = v_{t+1} + C = v_t + C + \kappa t^{-r} (C - v_t) &\leqslant v_t + C + C - v_t = 2C, \\
&\text{since } \kappa t^{-r} \leqslant 1, \text{ for } t \geqslant \text{ sufficiently large } T_2.
\end{aligned} \tag{48}$$

From the above two cases, it implies that $w_t$ either decreases or remains bounded by $2C$, for $t$ sufficiently large enough $T_0' = \max\{T_0, T_2\}$. So $w_t \leqslant \max\{2C, \sup_{t \geqslant T_0'} w_t\} = W_\infty < \infty, \forall t \geqslant T_0'$. Therefore $v_t \leqslant W_\infty$ for all large $t \geqslant T_0'$. Therefore,

$$\mathbb{E}[(\varepsilon_t^\vartheta)^2] = \frac{v_t}{t^r} \leqslant \frac{W_\infty}{t^r} \qquad (t \geqslant T_0).$$

For $1 \leqslant t < T_0'$, let $M = \max_{1 \leqslant s < T_0'} \mathbb{E}[(\varepsilon_t^\vartheta)^2]$. Then $M \leqslant M T_0'^r t^{-r}$. Now choose $K_1 = \max\{W_\infty, M T_0'^r\}$. Then

$$\mathbb{E}[(\varepsilon_t^\vartheta)^2] \leqslant \frac{K_1}{t^r}, \quad \forall t \geqslant 1.$$

$\square$

**Lemma 7.** *For a given policy $\pi_\theta$, let the step size $\delta_t = \frac{c}{t^r}, r \in (\frac{1}{2}, 1)$ with $c > \frac{r}{2\mathbb{E}_{\tau \sim \pi_\theta}[xe^{R(\tau)/x}]}$. Then, the stochastic variable $\omega_t$ satisfies:*

$$\mathbb{E}\left[|\omega_t - \omega^*(x, \theta)|^2\right] \leq \frac{K_2}{t}, \text{ for some } K_2 > 0 \text{ and } t \geq 1.$$

*Proof.* Define the error $\varepsilon_t^\omega := \omega_t - \omega^*(x, \theta)$. The update rule of $\omega_t$ becomes:

$$\varepsilon_{t+1}^\omega = \varepsilon_t^\omega \left(1 - \delta_t x e^{R(\tau_{t+1})/x}\right) + \delta_t e^{R(\tau_{t+1})/x}\left(R(\tau_{t+1}) - x\omega^*(x, \theta)\right).$$

Square both sides and take expectations and using $\mathbb{E}[(e^{R(\tau_{t+1})/x}(R(\tau_{t+1}) - x\,\omega^\star(x, \theta))\varepsilon_t^\omega] = 0$, we obtain:

$$\mathbb{E}\left[(\varepsilon_{t+1}^\omega)^2\right] = \mathbb{E}_\tau\left[\left(1 - \delta_t x e^{R(\tau)/x}\right)^2\right]\mathbb{E}\left[(\epsilon_t^\omega)^2\right] + \delta_t^2 \mathbb{E}_\tau\left[e^{2R(\tau)/x}\left(R(\tau) - x\omega^*(x, \theta)\right)^2\right].$$

Using $|R| \leq \frac{R_\infty}{1-\gamma}$ and $x \geq x_{\min}$, we bound:

$$\mathbb{E}_\tau\left[e^{2R(\tau)/x}\left(R(\tau) - x\omega^*(x)\right)^2\right] \leq \frac{R_\infty^2 e^{2R_\infty/(x_{\min}(1-\gamma))}}{(1-\gamma)^2} = \sigma_\omega^2.$$

Let $\mathbf{X} = x\,e^{R(\tau)/x}$ $(\geq 0)$, and $e_t = \mathbb{E}[(\varepsilon_t^\omega)^2]$. Then,

$$e_{t+1} \leq \underbrace{\mathbb{E}\left[(1 - \delta_t \mathbf{X})^2\right]}_{A_t} e_t + \delta_t^2\,\sigma_\omega^2. \tag{49}$$

Note that $(1 - \delta_t \mathbf{X})^2 = 1 - 2\delta_t \mathbf{X} + \delta_t^2 \mathbf{X}^2 \leq 1 - \delta_t \mathbf{X}, \qquad (\mathbf{X} > 0 \wedge \delta_t \mathbf{X} \geq 0 \implies \delta_t^2 \mathbf{X}^2 \leq \delta_t \mathbf{X}).$

Then,

$$A_t \leq 1 - \delta_t \mathbb{E}[\mathbf{X}] = 1 - \lambda t^{-r}, \qquad \text{where } \lambda = c\mathbb{E}[\mathbf{X}] > 0.$$

Substituting in Eq. (49), we get

$$e_{t+1} \leq \left(1 - \lambda t^{-r}\right)e_t + c^2\sigma_\omega^2 t^{-2r}. \tag{50}$$

Multiply (50) by $(t+1)^r$ and set $v_t = t^r e_t$:

$$\begin{aligned}
v_{t+1} &= (t+1)^r e_{t+1}\\
&\leq (t+1)^r\left(1 - \lambda t^{-r}\right)e_t + c^2\sigma_\omega^2(t+1)^r t^{-2r}\\
&= (t+1)^r t^{-r}\left(1 - \lambda t^{-r}\right)v_t + c^2\sigma_\omega^2 t^{-r}\left(1 + \tfrac{r}{t} + O(t^{-2})\right)\\
&= \left(1 + \tfrac{r}{t} + O(t^{-2})\right)\left(1 - \lambda t^{-r}\right)v_t + c^2\sigma_\omega^2 t^{-r}\left(1 + O(t^{-1})\right),
\end{aligned}$$

where we used the binomial expansion $(t+1)^r = t^r\left(1 + \tfrac{r}{t} + O(t^{-2})\right)$.

Because $r \in (\frac{1}{2}, 1)$, the term $\frac{r}{t} = o(t^{-r})$, so for sufficiently large $t$

$$\left(1 + \tfrac{r}{t} + O(t^{-2})\right)\left(1 - \lambda t^{-r}\right) = 1 - \kappa t^{-r}, \qquad \text{where } \kappa = \lambda - \frac{r}{2} > 0 \text{ (from Lemma assumption)}.$$

Absorbing the factor $1 + O(t^{-1})$ in the noise term into a constant 2 yields, for all large enough $t$,

$$v_{t+1} \leq \left(1 - \kappa t^{-r}\right)v_t + 2\,c^2\sigma_\omega^2 t^{-r}, \qquad \kappa = \lambda - \frac{r}{2} > 0. \tag{51}$$

Let $M = \frac{2c^2\sigma_\omega^2}{\kappa}$, and set $w_t = v_t + M$ $(t \geqslant 0)$. Now using $v_t = w_t - M$ in (51), we get

$$
\begin{aligned}
w_{t+1} = v_{t+1} + M &\leqslant (1 - \kappa t^{-r})(w_t - M) + 2c^2\sigma_\omega^2 t^{-r} + M \\
w_{t+1} &\leqslant (w_t - M) - \kappa t^{-r}(w_t - M) + \beta t^{-r} + M \\
&= w_t - \kappa t^{-r} w_t + (\kappa t^{-r} M + \beta t^{-r}) \\
&= w_t - \kappa t^{-r} w_t + 2\kappa t^{-r} M \\
&= w_t - \kappa t^{-r} (w_t - 2M).
\end{aligned}
\tag{52}
$$

If $w_t \geqslant 2M$, then from (52), we have

$$
w_{t+1} \leqslant w_t - \kappa t^{-r} (w_t - 2M) \leqslant w_t.
\tag{53}
$$

Otherwise,

$$
w_{t+1} = w_t + \kappa t^{-r} (2M - w_t) \leqslant w_t + 2M - w_t = 2M, \text{ since } \kappa t^{-r} \leqslant 1 \text{ for } t \text{ sufficiently large.}
\tag{54}
$$

From the above two cases, it implies that $w_t$ either decreases or remains bounded by $2M$, for $t$ sufficiently large enough $T_0$. So $w_t \leqslant \max(2M, \sup_{t \geqslant T_0} w_t) = W_\infty < \infty$, $\forall t \geqslant T$. Therefore $v_t \leqslant W_\infty - M$ for all large $t \geqslant T_0$.

Finally, since $e_t = v_t / t^r$,

$$
e_t \leqslant \frac{W_\infty}{t^r} \quad \text{for all } t \geqslant T_0.
$$

For the finite prefix $1 \leqslant t < T_0$, let $M' = \max_{1 \leqslant s < T_0} e_s$. Then $e_t \leqslant M' \leqslant M' T_0^r t^{-r}$. Finally choose $K_2 = \max\{W_\infty, M' T_0^r\}$, which yields

$$
\mathbb{E}\left[(\varepsilon_{t+1}^\omega)^2\right] \leqslant \frac{K_2}{t^r}, \quad \forall t \geqslant 0.
\tag{55}
$$

$\square$

**Theorem 5.** *Let step sizes $\delta_t = \frac{c}{t^r}$, $\xi_t = \frac{d}{t^b}$ with $r, b \in (\frac{1}{2}, 1)$ and $d = \frac{x_{\max}^3(1-b)}{2\sigma} \cdot b$. Under the Assumptions 1 and 3, the iterates $x_t \in \mathbb{R}$ asymptotically satisfy:*

$$
\mathbb{E}\left[(x_t - x^*)^2\right] \leqslant \frac{K_3}{t^b},
$$

*where $x^* = \arg\min_x G(x)$, and the constant $K_3 > 0$.*

*Proof.* Define $V_t = |x_t - x^*|^2$. The update rule is:

$$
x_{t+1} = x_t - \xi_t \left(G'(x_t) + \eta_t\right),
$$

where the gradient estimation error $\eta_t = \log \frac{\vartheta_t}{\vartheta^*(x)} - (\omega_t - \omega^*(x))$

Using the Lipschitz continuity of $\log(\cdot)$ near $\vartheta^*(x)$:

$$
\left|\log \frac{\vartheta_t}{\vartheta^*(x)}\right| \leqslant L_1 |\epsilon_t^\vartheta|,
$$

where $L_1 = \frac{1}{\inf_x \vartheta^*(x)} \leqslant e^{R_\infty/((1-\gamma)x_{\min})}$. Then:

$$
\|\eta_t\| \leqslant L_1 \|\epsilon_t^\vartheta\| + \|\epsilon_t^\omega\|.
$$

Squaring and taking expectations:

$$
\mathbb{E}[\|\eta_t\|^2] \leqslant 2L_1^2 \mathbb{E}[\|\epsilon_t^\vartheta\|^2] + 2\mathbb{E}[\|\epsilon_t^\omega\|^2] \leqslant \frac{2L_1^2 K_1 + 2K_2}{t^r}.
$$

Let $C = 2L_1^2 K_1 + 2K_2$. Expand $V_{t+1} = (x_t - \xi_t(G'(x_t) + \eta_t) - x^*)^2$. Taking conditional expectations:

$$\mathbb{E}[V_{t+1} \mid \mathcal{F}_t] \leq V_t - 2\xi_t G'(x_t)(x_t - x^*) + \xi_t^2 \mathbb{E}\left[(G'(x_t) + \eta_t)^2 \mid \mathcal{F}_t\right].$$

Using strong convexity $G'(x_t)(x_t - x^*) \geq \mu V_t$ with $\mu = \frac{\bar{\sigma}}{x_{\max}^3}$ and $|G'(x)| \leq \frac{2R_\infty}{x_{\min}(1-\gamma)} + |\log \alpha| \triangleq L$:

$$\mathbb{E}[V_{t+1}] \leq (1 - 2\mu\xi_t)\,\mathbb{E}[V_t] + \xi_t^2 \left(L^2 + \frac{C}{t^r}\right),$$

Substitute $\xi_t = \frac{d}{t^b}$:

$$\mathbb{E}[V_{t+1}] \leq \left(1 - \frac{2\bar{\sigma}d}{x_{\max}^3 t^b}\right)\mathbb{E}[V_t] + \frac{d^2}{t^{2b}}\left(L^2 + \frac{C}{t^r}\right).$$

Unroll recursively for $T \geq 1$:

$$\mathbb{E}[V_T] \leq \mathbb{E}[V_1] \prod_{t=1}^{T-1}\left(1 - \frac{2\bar{\sigma}d}{x_{\max}^3 t^b}\right) + d^2 \sum_{t=1}^{T-1} \frac{L^2 + C/t^r}{t^{2b}} \prod_{k=t+1}^{T-1}\left(1 - \frac{2\bar{\sigma}d}{x_{\max}^3 k^b}\right). \tag{56}$$

Using $1 - x \leq e^{-x}$, we get

$$\prod_{t=1}^{T-1}\left(1 - \frac{2\bar{\sigma}d}{x_{\max}^3 t^b}\right) \leq \exp\left(-\frac{2\bar{\sigma}d}{x_{\max}^3}\sum_{t=1}^{T-1}\frac{1}{t^b}\right).$$

Approximate the geometric sum $\sum_{t=1}^{T-1}\frac{1}{t^b} \geq \frac{T^{1-b}}{1-b}$. Hence,

$$\exp\left(-\frac{2\bar{\sigma}d}{x_{\max}^3} \cdot \frac{T^{1-b}}{1-b}\right) \leq \frac{1}{T^b}. \tag{57}$$

Choose $d = \frac{x_{\max}^3(1-b)}{2\bar{\sigma}} \cdot b$ so that $2\bar{\sigma}d = x_{\max}^3 b(1-b)$. Hence,

$$d^2 \sum_{t=1}^{T-1} \frac{L^2 + C/t^r}{t^{2b}} \prod_{k=t+1}^{T-1}\left(1 - \frac{2\bar{\sigma}d}{x_{\max}^3 k^b}\right) \leq d^2 \sum_{t=1}^{T-1} \frac{L^2 + C}{t^{2b}} \prod_{k=t+1}^{T-1}\left(1 - \frac{b(1-b)}{k^b}\right). \tag{58}$$

To bound the above term, we let $P_{t,T} = \prod_{k=t+1}^{T-1}\left(1 - b(1-b)k^{-b}\right)$. Because $0 < b(1-b) < 1$, and using bound $(1-x) \leq e^{-x}$ $(0 < x < 1)$ gives

$$P_{t,T} \leq \exp\left(-b(1-b)\sum_{k=t+1}^{T-1} k^{-b}\right). \tag{59}$$

Since $x \mapsto x^{-b}$ is decreasing,

$$\sum_{k=t+1}^{T-1} k^{-b} \geq \int_{t+1}^{T} x^{-b}\,dx = \frac{T^{1-b} - (t+1)^{1-b}}{1-b}.$$

Substituting this in Eq. (59) yields

$$P_{t,T} \leq \exp\left\{-b\left[T^{1-b} - (t+1)^{1-b}\right]\right\}. \tag{60}$$

Let $t_0 = \lfloor T/2 \rfloor$. We consider two cases:

**Case $1 \leq t \leq t_0$:** Because $x^{1-b}$ is increasing, $(t+1)^{1-b} \leq (\frac{T}{2} + 1)^{1-b}$, so from Eq.(60) $P_{t,T} \leq \exp\{-b(1 - 2^{b-1})T^{1-b}\}$. Consequently

$$\sum_{t=1}^{t_0} \frac{P_{t,T}}{t^{2b}} \leq \sum_{t=1}^{\infty} \frac{P_{t,T}}{t^{2b}} \leq \zeta(2b)\exp\{-b(1 - 2^{b-1})T^{1-b}\}. \tag{61}$$

**Case** $t_0 < t < T$: Let $s = T - t \, (= 1, \ldots, t_0)$. By applying the mean–value theorem on the function $x^{1-b}$, we obtain $T^{1-b} - (T - s + 1)^{1-b} \geqslant (1 - b)2^b T^{-b}(s - 1)$. So Eq. (60) implies $P_{t,T} \leqslant \exp\{-\lambda_b T^{-b}(s - 1)\}$ with $\lambda_b = b(1 - b)2^b$. Since $t = T - s \geqslant T/2$, we have

$$\sum_{t=t_0+1}^{T-1} \frac{P_{t,T}}{t^{2b}} \leqslant 2^{2b}T^{-2b} \sum_{s=1}^{\infty} e^{-\lambda_b T^{-b}(s-1)} = 2^{2b}T^{-2b} \frac{1}{1 - e^{-\lambda_b T^{-b}}}$$

$$\leqslant \frac{2^{2b+1}}{\lambda_b} T^{-b}, \tag{62}$$

where we used $e^{-x} \leqslant 1 - x/2 \ (0 < x \leqslant 1)$.

Now combining Eqs. (58), (61) and (62), we obtain

$$d^2 \sum_{t=1}^{T-1} \frac{L^2 + C}{t^{2b}} \prod_{k=t+1}^{T-1} \left(1 - \frac{b(1-b)}{k^b}\right). \leqslant d^2(L^2 + C)\left[\zeta(2b)\, e^{-b(1-2^{b-1})T^{1-b}} + \frac{2^{2b+1}}{\lambda_b} T^{-b}\right] = \mathcal{O}(T^{-b}). \tag{63}$$

Finally, combining Eqs.(56), (57) and (63), we get $\mathbb{E}\left[V_T\right] = O(1/T^b)$. This implies that

$$\mathbb{E}\left[(x_t - x^*)^2\right] \leqslant \frac{K}{t^b}, \quad K > 0, \forall t \geqslant T \text{ (sufficiently large)}. \tag{64}$$

Now using the finite comparison trick from Lemma 7, we can show that

$$\mathbb{E}\left[(x_t - x^*)^2\right] \leqslant \frac{K_3}{t^b}, \quad \forall t \geqslant 1, K_3 > 0. \tag{65}$$

$\square$

## A.3 Proof of Theorem 4

Recall the definition,

$$\widehat{\nabla J}_{\text{EVaR}}(\theta_t) = \frac{J_{\text{EVaR}}(\theta_t + c_t\Delta_t) - J_{\text{EVaR}}(\theta_t - c_t\Delta_t)}{2c_t\Delta_t}, \tag{66}$$

where $z(\theta_t^+) = G(x_t^+) - J_{\text{EVaR}}(\theta_t^+)$ and $z(\theta_t^-) = G(x_t^-) - J_{\text{EVaR}}(\theta_t^-)$.

**Lemma 8.** *Let* $J_{\text{EVaR}}^{(3)}(\theta) \equiv \partial^3 J_{\text{EVaR}} / \partial\theta^T \partial\theta^T \partial\theta^T$ *exist and* $\max_{i_1,i_2,i_3} \sup_\theta \|\text{EVaR}_{\alpha_{i_1 i_2 i_3}}^{(3)}(\theta)\|_\infty \leqslant \epsilon$. *Then* $\forall \theta \in interior(\Theta)$

$$b_t(\theta_t) = \mathbb{E}\left[\widehat{\nabla J}_{\text{EVaR}}(\theta_t) - \nabla J_{\text{EVaR}}(\theta_t) \mid \mathcal{F}_t\right] = \mathcal{O}(c_t^2).$$

*Proof.* By the continuity of $J_{\text{EVaR}}^{(3)}$ and $\Delta_t$ being a Bernoulli random variable, we have, by Taylor's theorem,

$$J_{\text{EVaR}}(\theta_t + c_t\Delta_t) \approx J_{\text{EVaR}}(\theta_t) + c_t\Delta_t^\top \nabla J_{\text{EVaR}}(\theta_t) + \frac{c_t J_{\text{EVaR}}^2}{2!}\Delta_t^\top \nabla^2 \text{EVaR}_\alpha(\theta_t)\Delta_t + \frac{c_t^3}{3!}\nabla^3 J_{\text{EVaR}}(\theta_t)\Delta_t \otimes \Delta_t \otimes \Delta_t,$$

where $\bar{\theta}_t$ lies on the line segment between $\theta_t$ and $\theta_t + c_t\Delta_t$. Hence,

$$\frac{J_{\text{EVaR}}(\theta_t + c_t\Delta_t) - J_{\text{EVaR}}(\theta_t - c_t\Delta_t)}{2c_t\Delta_t} = \frac{\Delta_t^\top}{\Delta_t}\nabla J_{\text{EVaR}}(\theta_t) + \frac{c_t^2}{12\Delta_t}\nabla^3\left(J_{\text{EVaR}}(\bar{\theta}_t) + J_{\text{EVaR}}(\bar{\theta}_t')\right)\Delta_t \otimes \Delta_t \otimes \Delta_t$$

Now,

$$b_t(\theta_t) = \mathbb{E}\left[\widehat{\nabla J}_{\text{EVaR}}(\theta_t) - \nabla J_{\text{EVaR}}(\theta_t) \mid \mathcal{F}_t\right]$$

$$= \mathbb{E}\left[\frac{J_{\text{EVaR}}(\theta_t + c_t\Delta_t) - J_{\text{EVaR}}(\theta_t - c_t\Delta_t)}{2c_t\Delta_t} - \nabla J_{\text{EVaR}}(\theta_t)\Big|\mathcal{F}_t\right].$$

Let $b_{t_l}$ denote the $l^{th}$ term of the bias vector $b_t$. Then

$$b_{t_l} = \mathbb{E}\left[\frac{\Delta_{t_l}}{\Delta_{t_l}}\nabla J_{\text{EVaR}}(\theta_t) + \frac{c_t^2}{12\Delta_{t_l}}\nabla^3\left(J_{\text{EVaR}}(\bar{\theta}_t) + J_{\text{EVaR}}(\bar{\theta}_t^{\,\prime})\right)\Delta_t \otimes \Delta_t \otimes \Delta_t - \nabla_\ell J_{\text{EVaR}}(\theta_t)\Big|\mathcal{F}_t\right] \tag{67}$$

where $\bar{\theta}_t^{\prime}$ lies on the line segment between $\theta_t$ and $\theta_t - c_t\Delta_t$. Now note that,

$$\mathbb{E}\left[\Delta_t^{-1}\Delta_t^\top \nabla J_{\text{EVaR}}(\theta_t) \mid \mathcal{F}_t\right] = (\nabla J_{\text{EVaR}}(\theta_t))_1 \mathbb{E}\left[\Delta_t^{-1}\Delta_{t_1} \mid \mathcal{F}_t\right] + \cdots + (\nabla J_{\text{EVaR}}(\theta_t))_p \mathbb{E}\left[\Delta_t^{-1}\Delta_{t_p} \mid \mathcal{F}_t\right]$$

$$\text{(Since, } \nabla J_{\text{EVaR}}(\theta_t) \text{ is measurable } w.r.t. \ \mathcal{F}_t\text{)}$$

$$= (\nabla J_{\text{EVaR}}(\theta_t))_1 \mathbb{E}\left[\begin{pmatrix} 1 \\ \Delta_{t_2}^{-1}\Delta_{t_1} \\ \vdots \\ \Delta_{t_p}^{-1}\Delta_{t_1} \end{pmatrix} \mid \mathcal{F}_t\right] + \cdots + (\nabla J_{\text{EVaR}}(\theta_t))_p \mathbb{E}\left[\begin{pmatrix} \Delta_{t_1}^{-1}\Delta_{t_p} \\ \Delta_{t_2}^{-1}\Delta_{t_p} \\ \vdots \\ 1 \end{pmatrix} \mid \mathcal{F}_t\right]$$

$$= (\nabla J_{\text{EVaR}}(\theta_t))_1 \begin{pmatrix} 1 \\ \mathbb{E}\Delta_{t_2}^{-1}\mathbb{E}\Delta_{t_1} \\ \vdots \\ \mathbb{E}\Delta_{t_p}^{-1}\mathbb{E}\Delta_{t_1} \end{pmatrix} + \cdots + (\nabla J_{\text{EVaR}}(\theta_t))_p \begin{pmatrix} \mathbb{E}\Delta_{t_1}^{-1}\mathbb{E}\Delta_{t_p} \\ \mathbb{E}\Delta_{t_2}^{-1}\mathbb{E}\Delta_{t_p} \\ \vdots \\ 1 \end{pmatrix}$$

$$\text{(Since, } \mathbb{E}\Delta_{t_i} = 0, \forall i \in [1\ldots p] \text{ and } \Delta_{t_i} \text{ is independent of } \Delta_{t_j} \forall i \neq j\text{)}$$

$$= (\nabla J_{\text{EVaR}}(\theta_t))_1 \begin{pmatrix} 1 \\ 0 \\ \vdots \\ 0 \end{pmatrix} + \cdots + (\nabla J_{\text{EVaR}}(\theta_t))_p \begin{pmatrix} 0 \\ 0 \\ \vdots \\ 1 \end{pmatrix} = \nabla J_{\text{EVaR}}(\theta_t) \tag{68}$$

Therefore, from Eq.(67) and Eq.(68), we get,

$$b_{t_l} = \frac{1}{12}\mathbb{E}\left[\frac{1}{\Delta_{t_l}}\left(\nabla^3 J_{\text{EVaR}}(\bar{\theta}_t) + \nabla^3 J_{\text{EVaR}}(\bar{\theta}_t^{\,\prime})\right)\bar{\Delta}_t \otimes \bar{\Delta}_t \otimes \bar{\Delta}_t \mid \mathcal{F}_t\right] \tag{69}$$

We can bound the term on the right-hand side of Eq.(69) in magnitude as follows:

$$b_t\left(\theta_t\right) = \frac{1}{12}\mathbb{E}\left[\frac{1}{\Delta_{t_l}}\left(\nabla^3 J_{\text{EVaR}}(\bar{\theta}_t) + \nabla^3 J_{\text{EVaR}}(\bar{\theta}_t^{\prime})\right)\bar{\Delta}_t \otimes \bar{\Delta}_t \otimes \bar{\Delta}_t \mid \mathcal{F}_t\right]$$

$$\leqslant \frac{\epsilon c_t^2}{6}\sum_{i_1}\sum_{i_2}\sum_{i_3}\mathbb{E}\left[\frac{\Delta_{t_{i_1}}\Delta_{t_{i_2}}\Delta_{t_{i_3}}}{\Delta_{k_l}}\right] \leqslant \frac{p^3\epsilon c_t^2}{6} = \mathcal{O}(c_t^2). \tag{70}$$

The first inequality follows as $\nabla^3 J_{\text{EVaR}}(\bar{\theta}) \leqslant \epsilon, \forall \theta$ and the latter inequality follows since $\frac{\Delta_{t_{i_1}}\Delta_{t_{i_2}}\Delta_{t_{i_3}}}{\Delta_{t_l}} \leqslant 1$. $\quad\square$

**Proof of Theorem 4**

*Proof.* Consider the recursion from Step 12 of the algorithm:

$$
\theta_{t+1} = \theta_t + a_t \Bigg( \underbrace{\mathbb{E}\left[\widehat{\nabla J}_{\text{EVaR}}(\theta_t) - \nabla J_{\text{EVaR}}(\theta_t) \mid \mathcal{F}_t\right]}_{b_t} - \mathbb{E}\left[\widehat{\nabla J}_{\text{EVaR}}(\theta_t) - \nabla J_{\text{EVaR}}(\theta_t) \mid \mathcal{F}_t\right] +
$$

$$
\widehat{\nabla J}_{\text{EVaR}}(\theta_t) + \frac{z(\theta_t^+) - z(\theta_t^-)}{2c_t \Delta_t} \Bigg)
$$

$$
= \theta_t + a_t \left( b_t + \varphi_t + \nabla J_{\text{EVaR}}(\theta_t) + \widehat{\nabla J}_{\text{EVaR}}(\theta_t) - \mathbb{E}\left[\widehat{\nabla J}_{\text{EVaR}}(\theta_t) \mid \mathcal{F}_t\right] \right)
$$

$$
= \theta_t + a_t \left( b_t + e_t + \varphi_t + \nabla J_{\text{EVaR}}(\theta_t) \right), \text{ where } e_t = \widehat{\nabla J}_{\text{EVaR}}(\theta_t) - \mathbb{E}\left[\widehat{\nabla J}_{\text{EVaR}}(\theta_t) \mid \mathcal{F}_t\right]
$$

$$
\text{and } \varphi_t = \frac{z(\theta_t^+) - z(\theta_t^-)}{2c_t \Delta_t}. \tag{71}
$$

Then

$$
\mathbb{E}\left[|\varphi_t| \,|\mathcal{F}_t\right] = \mathbb{E}\left[\left|\frac{z(\theta_t^+) - z(\theta_t^-)}{2c_t \Delta_t}\right| \Big| \mathcal{F}_t\right] \leqslant \frac{1}{2c_t} \mathbb{E}\left[\Delta_t^{-1}(z(\theta_t^+) - z(\theta_t^-)) \big| \mathcal{F}_t\right]
$$

$$
\leqslant \frac{1}{2c_t} \mathbb{E}\left[|\Delta_t^{-1}|(|z(\theta_t^+)| + |z(\theta_t^-)|) \big| \mathcal{F}_t\right]
$$

$$
= \frac{1}{2c_t} \mathbb{E}\left[|\Delta_t^{-1}||z(\theta_t^+)| \big| \mathcal{F}_t\right] + \mathbb{E}\left[|\Delta_t^{-1}||z(\theta_t^-)| \big| \mathcal{F}_t\right]
$$

$$
\leqslant \frac{L_G}{2c_t} \mathbb{E}\left[|\Delta_t^{-1}||x_t^+ - x^*(\theta_t^+)| \big| \mathcal{F}_t\right] + \frac{L_G}{2c_t} \mathbb{E}\left[|\Delta_t^{-1}||x_t^- - x^*(\theta_t^-)| \big| \mathcal{F}_t\right] \quad \text{(Since } G \text{ is Lipschitz}
$$

$$
\text{with constant } L_G \text{ which follows from Theorem 1)}
$$

$$
\leqslant \frac{1}{2c_t} \left( \underbrace{\mathbb{E}\left[\Delta_t^{-2}\right]^{\frac{1}{2}}}_{=1} \mathbb{E}\left[|x_t^+ - x^*(\theta_t^+)|^2 \big| \mathcal{F}_t\right]^{\frac{1}{2}} + \underbrace{\mathbb{E}\left[\Delta_t^{-2}\right]^{\frac{1}{2}}}_{=1} \mathbb{E}\left[|x_t^- - x^*(\theta_t^-)|^2 \big| \mathcal{F}_t\right]^{\frac{1}{2}} \right)
$$

$$
\text{(by Cauchy-Schwartz Inequality)}
$$

$$
\leqslant \frac{\sqrt{K}}{c_t N_t^{\frac{b}{2}}} \quad \text{(by Theorem 5)}
$$

Now, by the Monotone Convergence Theorem, we have

$$
\mathbb{E}\left[\sum_{t \geqslant 1} a_t |\varphi_t|\right] = \sum_{t \geqslant 1} a_t \mathbb{E}\left[|\varphi_t|\right] \leqslant \sum_{t \geqslant 1} \frac{a_t \sqrt{K}}{c_t N_t^{\frac{b}{2}}} < \infty.
$$

Therefore

$$
\mathbb{P}\left(\sum_{t \geqslant 1} a_t |\varphi_t| < \infty\right) = 1 \quad \Rightarrow \sum_{t \geqslant 1} a_t \varphi_t < \infty \quad a.s. \quad \Rightarrow \sum_{t \geqslant k} a_t \varphi_t \xrightarrow[k \to 0]{} 0. \tag{72}
$$

Define

$$
\xi_{t+1} = \sum_{i=0}^{t} a_t e_t, t \geqslant 0. \tag{73}
$$

Then

$$
\mathbb{E}\left[\xi_{t+1} | \mathcal{F}_t\right] = \mathbb{E}\left[\sum_{i=0}^{t-1} a_t e_t \Big| \mathcal{F}_t\right] = \sum_{i=0}^{t} a_t \mathbb{E}\left[e_t | \mathcal{F}_t\right] + a_t \left( \mathbb{E}\left[\widehat{\nabla J}_{\text{EVaR}}(\theta_t) | \mathcal{F}_t\right] - \mathbb{E}\left[\widehat{\nabla J}_{\text{EVaR}}(\theta_t) \mid \mathcal{F}_t\right] \right) = \xi_t. \tag{74}
$$

This implies that $\{\xi_t\}$ is a martingale with respect to filtration $\{\mathcal{F}_t\}$. Also, since $J_{\texttt{EVaR}}$ is continuously differentiable, we have $\xi_t$ is square-integrable, $\forall t$, $i.e$, $\mathbb{E}\left[\|\xi_t\|^2\right] < \infty$, $\forall t$. Again, by the continuous differentiability of $J_{\texttt{EVaR}}$ and Assumption 4, we obtain

$$\sum_t \mathbb{E}\left[\|\xi_{t+1} - \xi_t\|^2 | \mathcal{F}_t\right] = \sum_t a_t^2 \mathbb{E}\left[\|e_t\|^2\right] < \infty \text{ on the set } \{\sup_t \|\theta_t\| < \infty\}.$$

Therefore, by the Martingale convergence theorem, we get

$$\lim_{t \to \infty} \xi_t \text{ exists on the event } \{\sup_t \|\theta_t\| < \infty\}. \tag{75}$$

Hence, by Theorem 2, Chapter 2 of Borkar (2008), the asymptotic behavior of the sample paths belonging to the event $\{\sup_t \|\theta_t\| < \infty\}$ is equivalent to the long-term behavior of the dynamical system induced by the ODE

$$\frac{d\theta(t)}{dt} = \nabla J_{\texttt{EVaR}}(\theta(t)), t \geq 0. \tag{76}$$

This further implies that the iterates $\theta_t$ corresponding to the sample paths belonging to the event $\{\sup_t \|\theta_t\| < \infty\}$ converge to any of the compact transitive invariant sets connected internally in chains of (76). Invariant sets are subsets of the state space that remain unchanged under the flow of the dynamical system. The dynamical system (76) driven by the gradient of the $J_{\texttt{EVaR}}$ is a gradient flow where the only possible invariant sets are the subsets of $H = \{\theta | \nabla \texttt{EVaR}_\alpha(\theta) = 0\}$ (Lemma 1, Section 10.2 of Borkar (2008)). Further, by invoking the LaSalle invariance principle and the Lyapunov theorem, one can obtain that the asymptotically stable points inside $H$ are given by $\{\theta \in H | \nabla^2 J_{\texttt{EVaR}}(\theta) \preccurlyeq 0\}$. $\qquad\square$

## B    Comparison of `EVaR` estimation methods

One can also estimate `EVaR` of discounted cumulative rewards for a sample trajectory using a disciplined convex programming characterization as stated by Cajas (2021) as follows

$$J_{\texttt{EVaR}}(\theta) = \min_{\beta,t,u} \ t - \beta \ln(\alpha N), \tag{77}$$

$$\text{subject to } \beta \geq \sum_{j=1}^N u_j \text{ and } (J_{\texttt{EVaR}}(\theta)_j - t, \beta, u_j) \in K_{\exp}, \forall j \in [1, N]$$

where $\{J_{\texttt{EVaR}}(\theta)_j\}_{j=1}^N$ are $N$ realizations of $J_{\texttt{EVaR}}(\theta)$, $\beta$, $t$ and $u$ are variables and $K_{\exp}$ is an exponential cone (Chares, 2009) which is defined as follows

$$K_{\exp} = \left\{(a, b, c) \mid b > 0, c \geq b \exp\left(\frac{a}{b}\right)\right\} \cup \{(a, b, c) \mid a \leq 0, b = 0, c \geq 0\}. \tag{78}$$

The above optimization can be solved efficiently to any desired level of accuracy via interior-point methods due to the existence of computationally tractable barrier functions, which enable the efficient exploration of the solution space (Chandrasekaran & Shah, 2017). In Algorithm 2, we give an `EVaR` optimization algorithm where the `EVaR` estimate computation is done using the disciplined convex cone method by solving the optimization problem. This requires access to $N$ trajectories for the computation and which decides the desired accuracy of the estimate. This process is not an online process with respect to the estimation, and the movement of the samples has little influence over the `EVaR` estimate. When this process is compared with the Stochastic Approximation (SA) version of `EVaR` estimation, we see in Figure 25 is more resilient to changing environment dynamics, and a higher error band justifies the online nature and adaptability to the samples.

**Remark.** *The rewards for the environment are negated to comply with the structure of the solver in Algorithm 2.*

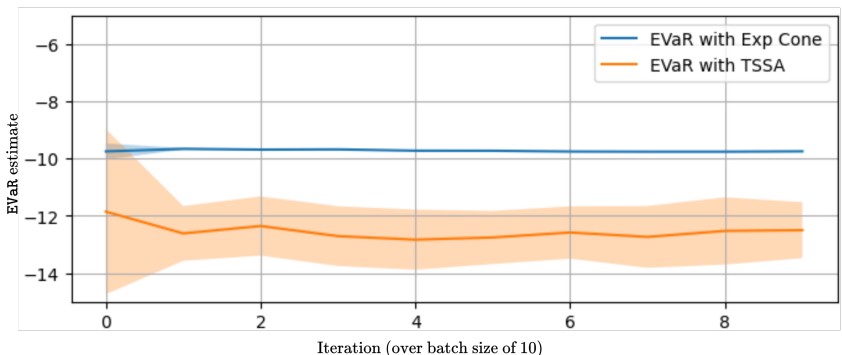

Figure 25: `EVaR` estimation using two-time scale schotastic approximation, where the blue represents the estimate using the convex optimization and the orange band represents the estimate using the SA method. Run for 10 batches with CartPole as the environment with an episode length of 100.

---

**Algorithm 2** `EVaR` Optimization using Disciplied Convex Cone

---

**Require:** risk level $\alpha \in (0,1)$, initial $\theta_0 \in \mathbb{R}^p$, step-sizes $\{a_t, c_t, \delta_t, \xi_t\}$, inner lengths $N_t$

1: Initialize policy network $(\pi_\theta)$ parameters $\theta \in \Theta$
2: **for** $t = 0, \ldots, T-1$ **do**
3:      Draw $\Delta_t \in \{\pm 1\}^p$ IID.
4:      $\theta_t^+ \leftarrow \theta_t + c_t \Delta_t, \quad \theta_t^- \leftarrow \theta_t - c_t \Delta_t$
5:      **for** $k = 1, \ldots, N_t$ **do**                          ▷ `EVaR estimation for "+"`
6:          Sample trajectory $\tau_{t,k}^+ \sim \pi_{\theta_t^+}$, compute $R_{t,k}^+ = \sum_{u=0}^{H} \gamma^u r_u$
7:      **end for**
8:      Solve using the Interior point method

$$G_{t,j}^+ \leftarrow \texttt{Solve}\big( \min_{\beta,t,u} \ t - \beta \ln(\alpha N), \ \texttt{subject to } \beta \geqslant \sum_{j=1}^{N} u_j$$

$$\texttt{and } \left( R_{t,k}^+ - t, \beta, u_j \right) \in K_{\exp}, \forall j = 1, \ldots, N \big)$$

9:      **for** $k = 1, \ldots, N_t$ **do**                          ▷ `EVaR estimation for "-"`
10:        Sample trajectory $\tau_{t,k}^- \sim \pi_{\theta_t^-}$, compute $R_{t,k}^- = \sum_{u=0}^{H} \gamma^u r_u$
11:      **end for**
12:      Solve using the Interior point method

$$G_{t,j}^- \leftarrow \texttt{Solve}\big( \min_{\beta,t,u} \ t - \beta \ln(\alpha N), \ \texttt{subject to } \beta \geqslant \sum_{j=1}^{N} u_j$$

$$\texttt{and } \left( R_{t,k}^- - t, \beta, u_j \right) \in K_{\exp}, \forall j = 1, \ldots, N \big)$$

13:      $\hat{g}_t \leftarrow \dfrac{G_t^+ - G_t^-}{2\,c_t} \, \Delta_t^{-1}$
14:      $\theta_{t+1} \leftarrow \theta_t + a_t \, \hat{g}_t$
15: **end for**
16: **return** $\theta_T$

---

## C   Experiments Detail

### C.1   Finite Difference Gradient Estimation

The hyperparameters used in our finite difference-based gradient estimation have a significant impact on the algorithm's performance. The timeout parameter controls the time allotted for each function evaluation, ensuring efficient computation. Iterations govern the number of optimization steps, balancing convergence speed and accuracy. Learning rate decay and power adjust the step size dynamically, allowing the learning process to slow down as the model converges. Perturbation size affects the extent of exploration in the gradient estimation, while its decay and power ensure that perturbations shrink over time, refining the gradient's precision. Momentum helps maintain stable updates by incorporating past gradients, and the Adam parameters (Beta and Epsilon) enhance robustness, particularly in the face of noisy gradient estimates, providing smoother and more reliable updates. Together, these hyperparameters create a dynamic and adaptable optimization process, crucial for navigating uncertain and noisy environments.

| Hyperparameter | Value |
|---|---|
| Timeout | $1 \times 10^{-4}$ |
| Iterations | 10,000 |
| Learning Rate Decay | $1 \times 10^{-3}$ |
| Learning Rate Power | 0.5 |
| Perturbation Size ($px$) | 2.0 |
| Perturbation Decay | $1 \times 10^{-2}$ |
| Perturbation Power | 0.161 |
| Momentum | 0.9 |
| Beta (Adam Parameter) | 0.999 |
| Epsilon (Adam Parameter) | $1 \times 10^{-7}$ |

Table 5: Hyperparameters for Finite Difference Gradient Estimation

### C.2   MuJoCo and Gridworld

We evaluate our proposed algorithm on various continuous control tasks from the OpenAI Gym suite Brockman et al. (2016), where we augment the environment with random normal noise to introduce uncertainty. For the discrete setting, we use a custom GridWorld environment with randomly placed obstacles covering approximately 30% of the grid. Importantly, we ensure that a path always exists from the start to the goal state, even with obstacles. The hyperparameters used across all experiments are summarized below :

| Hyperparameter | Value |
|---|---|
| Learning rate | $a_k = \frac{a}{(k+1+A)^{0.602}}, c_k = \frac{c}{(k+1)^{0.101}}$ |
| Constant $A$ | $10^{-7}$ |
| Constant $c$ | 0.999 |
| Random noise parameter $\delta$ | $1 - \alpha$ |
| Action sampling | $a_t \sim \pi_\theta(\cdot|s) + \delta \mathcal{N}(0, 1)$ |
| Step size $\delta$ | 0.12 |
| Step size $\xi$ | $3 \times 10^{-10}$ |

Table 6: Hyperparameters used in experiments

**Remark.** *The confidence value $\alpha$ that is used for* `EVaR` *is flipped, i.e $1 - \alpha$ is used for the same experiment for* `CVaR` *and* `VaR` *estimates to align the measures in the direction of the upward risk, which is used for most reward distributions.*

### C.3 Implementation Details for baseline comparison

The tabular implementations of EVAR–SA and the three CVaR-based baselines, present their mathematical update rules, and summarize all hyperparameters. By re-implementing each algorithm in an identical finite-horizon MDP framework, we ensure that performance differences stem solely from the risk criteria and their estimators.

#### C.3.1 Environments and Reward Structure

- **Cliff Walk (4×12).** States form a $4 \times 12$ grid. Start at $(3, 0)$, goal at $(3, 11)$. Stepping into any cliff cell $(3, i)$ for $1 \leqslant i \leqslant 10$ yields $r = -100$ and resets to start; all other moves incur $r = -1$; reaching the goal yields $r = 0$ and termination.

- **Windy GridWorld (7×10).** States form a $7 \times 10$ grid. Start at $(3, 0)$, goal at $(3, 7)$. At each step, the agent chooses one of four cardinal moves, then experiences an upward wind of strength $w_j \in \{0, 0, 0, 1, 1, 1, 2, 2, 1, 0\}$ in column $j$, paying $r = -1$ per step and terminating with $r = 0$ upon reaching the goal.

#### C.3.2 Algorithmic Update Rules and Hyperparameters

All four algorithms are implemented within an identical tabular Q-learning framework. Specifically, at each time step, they perform the update

$$Q_{t+1}(s, a) = Q_t(s, a) + \alpha_t[y_t - Q_t(s, a)],$$

where $y_t$ is the algorithm-specific TD target and $a_t$ is the step size ( as described in Table 7). Action selection follows an $\varepsilon$–greedy policy with $\varepsilon = 0.1$, and future returns are discounted by $\gamma = 0.99$. Each method runs for 500 episodes per random seed (with a maximum of 200 steps per episode), and results are averaged over eight independent seeds. All Q-tables are initialized to zero, and reproducibility is ensured by calling `np.random.seed(seed)` at the start of each seed.

Table 7: Algorithmic update rules and hyperparameters. All methods use tabular Q-learning with base $\alpha = 0.1$, $\gamma = 0.99$, and $\varepsilon = 0.1$.

| Algo. | Risk Objective | TD Target $y_t$ | Params |
|---|---|---|---|
| **EVAR–SA** | $EVaR_\alpha(R) = \inf_{\beta>0} \frac{1}{\beta} \ln\left(\frac{\mathbb{E}[e^{\beta R}]}{\alpha}\right)$, $G(R) = \frac{1}{\beta} \ln\left(\frac{1}{\alpha} \sum_i e^{\beta R_i}\right)$ | $r + \gamma \max_{a'} Q(s', a') + \alpha_t \frac{G(R+c)-G(R)}{c}$, $\alpha_t = \frac{0.1}{1+0.01\,N(s)}$ | $c = 0.05$, $N = 20$ |
| **CVaR–PG** | $CVaR_\alpha(R) = \frac{1}{k} \sum_{i=1}^{k} R_{(i)}$, $k = \lfloor \alpha N \rfloor$ | $(1 - \lambda)[r + \gamma \max_{a'} Q(s', a')] + \lambda\,CVaR_\alpha(R)$ | $\lambda = 0.3$, $N = 15$ |
| **SDPG–CVaR** | $\min_Q \sup_u \mathbb{E}[R - u(s)] + \frac{1}{\alpha} \mathbb{E}[(u(s) - R)^+]$ | $u_{t+1} = u_t + \alpha[R_t - u_t]$, $y_t = r + \gamma \max_{a'} Q(s', a') - u_t$ | dual step = 0.1 |
| **D4PG–CVaR** | $CVaR_\alpha(R) = \frac{1}{k} \sum_{i=1}^{k} R_{(i)}$, $k = \lfloor \alpha N \rfloor$ | $0.6[r + \gamma \max_{a'} Q(s', a')] + 0.4\,CVaR_\alpha(R)$ | window $N = 12$ |

In EVAR-SA, gradients of the entropic value-at-risk are estimated using a two-point finite-difference stochastic approximation on recent returns, allowing for unbiased updates without backpropagation through analytic risk gradients. The method also adapts its learning rate per state, ensuring convergence stability across heterogeneous state visitation frequencies. In contrast, `CVaR`-PG augments standard temporal-difference (TD) targets with the empirical CVaR of the worst $\alpha$-fraction of returns, SDPG-CVaR solves a saddle-point dual formulation with per-state dual variables, and D4PG-CVaR employs sliding-window return samples for quantile averaging within a distributional Q-learning framework. The Cliff Walk risk landscape contrasts a high-penalty "cliff path" with a uniformly safe alternative; EVAR-SA's entropic objective naturally avoids the peak risk region. In Windy GridWorld, per-column wind strength yields uneven state-transition uncertainty, which EVAR-SA accommodates via its adaptive step size in the SA update.

**Note:** *All algorithms evaluated in this work confirm that the observed empirical advantages of EVAR–SA arise from its risk criterion and estimator alone.*

## C.4 Glycemic Control

The environment is modelled as an MDP in such that the state space consists of multiple noisy glucose measurements at various time points in the past, the carbohydrate intakes, and other relevant information about the patient. The action is the amount of insulin to be delivered to the patient, also a scalar that indicates the insulin to be administered. We observe from our experiments that our algorithm can keep the patient at an admissible level of risk. There are instances where the glucose breaches into the hypo or hyper region, the policy course corrects to maintain the stable condition.

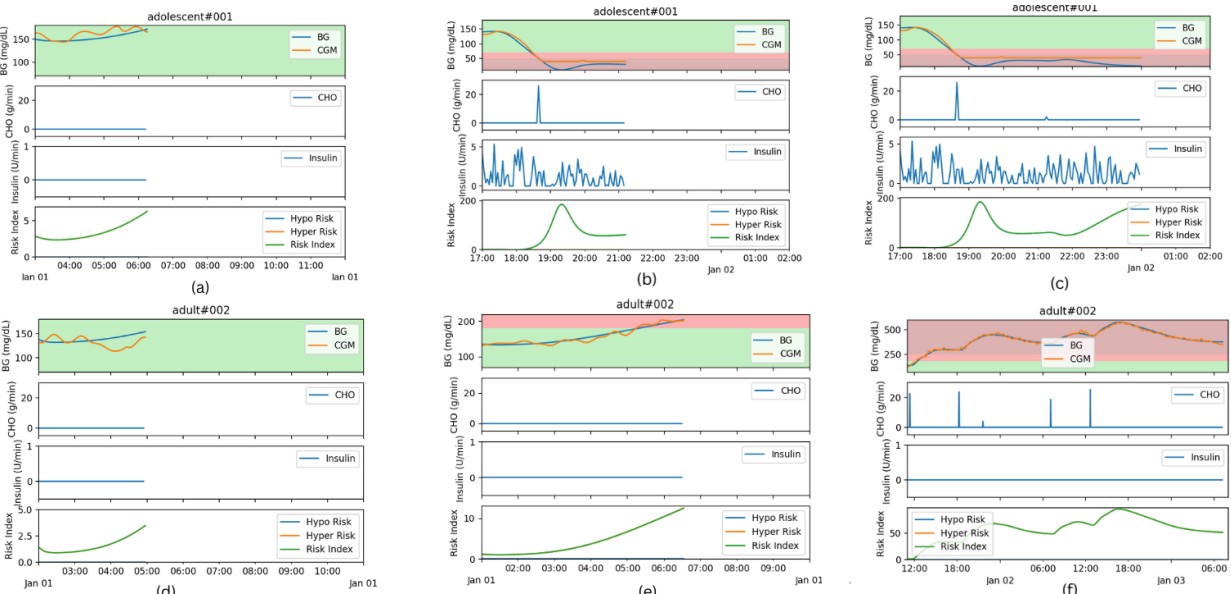

Figure 26: In plots (a),(b) and (c) represent the patient profile of *adoloscent001* and the rest of the plots represent the patient *adult002*. Both patients show up to be stable and alive under the influence of the administration of insulin via the controller, where the signals are optimized by the `EVaR` optimizer.

We show that our algorithm is equipped to handle high-risk scenarios like administering insulin to Type-1 diabetes patients of type *adult#002, adolescent#001* registered on the Gym environment using the simulator developed by Xie (2018) for RL control tasks. The simulator emulates a PID controller that provides the insulin to maintain the Blood Glucose levels, which is the action for the agent, and we try to optimize by increasing the longevity of the patient using the finite difference `EVaR` optimization. The patient profiles are given below, where CF: Carbohydrate Factor, CR: Carb Ratio (often referred to as Carb-to-Insulin Ratio), and TDI: Total Daily Insulin.

| Name | CR | CF | Age | TDI |
|---|---|---|---|---|
| adolescent#001 | 12 | 15.0360 | 18 | 36.7339 |
| adult#002 | 8 | 9.2128 | 65 | 57.8688 |

Table 8: Patient profiles

## C.5 Portfolio Optimization

**Problem Definition :** We focus on the top 10 US DJIA stocks—`[KO, AAPL, MSFT, JPM, WMT, UNH, V, PG, JNJ, HD]`—using market data from January 1, 2014, to January 1, 2024, which provides a decade of market information encompassing various economic cycles. Our custom `PortfolioEnv` class, designed to adhere to the OpenAI Gym interface (Brockman et al., 2016) and compatible with Stable Baselines 3 (Hill et al., 2018), represents a 50-day window of historical returns as a matrix $\mathbf{O} \in \mathbb{R}^{50 \times 10}$. To minimize the `EVaR` value of the negative returns, which in our case are the portfolio values represented as $\mathcal{R}$ under

the constraints that the weights of the portfolio always sum up to 1 after each allocation step. We find the optimal `EVaR` portfolio with N-assets represented by the optimization problem:

$$\max_{\beta > 0} \; \beta^{-1} \log\left(\alpha^{-1} \mathbb{E}[e^{\beta \mathcal{R}}]\right) \tag{79}$$

$$\text{s.t.} \; \sum_{i=1}^{N} w_i = 1 \tag{80}$$

$$\beta > 0, w \geqslant 0 \tag{81}$$

here the weights of the $N$ assets in the portfolio is represented by the weight vector $\boldsymbol{w} = [w_1, \ldots, w_n]^\top$ such that $\sum_{i=1}^{N} w_i = 1$, where $0 \leqslant w_i \leqslant 1$. An asset weight of 0 indicates zero holdings of a particular asset in a portfolio, whereas a weight of 1 means that the entire portfolio is concentrated in the said asset.

**Action Space :** For each action $A \in \mathcal{A}$, the action $A$ represents portfolio weights for portfolio allocation on $N$ assets. The weight constraint makes each action represented as $A = [a_1, \ldots, a_N]^\top$ so that $\sum_{i=1}^{N} a_i = 1$ where $0 \leqslant a_i \leqslant 1$. In alternative implementations of this framework, $a_i < 0$ would permit short-selling an asset, while $a_i > 1$ would allow for leveraged positions. However, in our scenario, we limit the actions to non-leveraged, long-only positions. To enforce these constraints, we apply the softmax function to the agent's continuous actions.

**State Space :** The state space $\mathcal{S}$ is represented by a matrix $S_t \in \mathbb{R}^{w \times n}$, where $w$ is a predefined window size and $n$ is the number of assets. Each element $s_{i,j}$ of the matrix represents the return of the asset $j$ at the time step $t - w + i$. Formally:

$$S_t = \{r_{i,j} | i \in [t - w + 1, t], j \in [1, n]\} \tag{82}$$

where $r_{i,j}$ is the return of asset $j$ at time $i$. The observation space is bounded, with $S_t \in [-1, 1]^{w \times n}$.

At each time step $t$, the state $S_t$ is represented by a matrix $S_t \in \mathbb{R}^{w \times n}$, where:

- $w$ is the window size (number of historical time steps considered)

- $n$ is the number of assets in the portfolio

Formally, we define the state matrix as follows:

$$S_t = \begin{bmatrix} r_{t-w+1,1} & r_{t-w+1,2} & \cdots & r_{t-w+1,n} \\ r_{t-w+2,1} & r_{t-w+2,2} & \cdots & r_{t-w+2,n} \\ \vdots & \vdots & \ddots & \vdots \\ r_{t,1} & r_{t,2} & \cdots & r_{t,n} \end{bmatrix} \tag{83}$$

where $r_{i,j}$ represents the return of asset $j$ at time step $i$. Each element $s_{i,j}$ of the matrix $S_t$ corresponds to the return of a specific asset at a specific time:

$$s_{i,j} = r_{t-w+i,j} = \frac{P_{t-w+i,j} - P_{t-w+i-1,j}}{P_{t-w+i-1,j}} \tag{84}$$

where $P_{t,j}$ is the price of asset $j$ at time $t$. To ensure numerical stability and consistent scale across different assets, we bound the elements of the state matrix $S_t \in [-1, 1]^{w \times n}$. The rows of the matrix $S_t$ represent different time steps, with the most recent returns in the bottom row and the oldest returns in the top row. The rows of the matrix $S_t$ represent different time steps, with the most recent returns in the bottom row and the oldest returns in the top row. This structure allows the agent to identify temporal patterns or trends in asset returns, potentially.

**Reward Function :** The reward function is designed to balance portfolio return with risk. We use the Entropic Value at Risk (`EVaR`) as a risk measure. The reward $R_t$ at time $t$ is defined as:

$$R_t = -\texttt{EVaR}_\alpha(\mathcal{R}_t) \tag{85}$$

where the negative portfolio returns are given by $\mathcal{R}_t = (V_0 - V_t)/V_0$ is the portfolio return up to time $t$, and $\alpha$ is the confidence level for $\texttt{EVaR}$ calculation. The negative sign before $\texttt{EVaR}$ ensures that minimizing risk corresponds to maximizing reward.

An episode terminates when either:

- The end of the available market data is reached ($t = T$), or

- The portfolio value drops to zero ($V_t \leqslant 0$).

**Transistion Dynamics :**   The environment functions as a comprehensive interface to the market, utilizing a technique known as market replay to traverse historical data. Additionally, it operates as both a broker and an exchange; at each timestep, it processes the agent's actions to rebalance the portfolio according to the latest prices and specified allocations. As the trading day progresses and new price data is acquired, the environment provides these updates to the agent as observations, accompanied by the Entropic Value at Risk ($\texttt{EVaR}$) reward. For the scope of this study, we permit instantaneous rebalancing of the portfolio. Given an action $a_t$, the environment updates the portfolio value $V_t$ according to:

$$V_{t+1} = V_t \cdot \left(1 + \sum_{i=1}^{n} a_{t,i} \cdot r_{t+1,i} - c \cdot \sum_{i=1}^{n} |a_{t,i} - a_{t-1,i}|\right) \tag{86}$$

where $r_{t+1,i}$ is the return of asset $i$ at time $t+1$, and $c$ is the transaction cost rate, which is kept constant in our case. After rebalancing, the environment creates the next state $S_{t+1}$ and proceeds to the next timestep $t+1$. It calculates the new portfolio value based on $V_{t+1}$ and computes the reward $R_t$, which it returns to the agent.

### C.5.1   Algorithm

We use Soft Actor Critic Haarnoja et al. (2018) for policy optimization and incorporate the stochastic recursive updates for the convergence of $\texttt{EVaR}$ sample estimate by augmenting the critic function. We employ the SAC algorithm with its default MLP policy as implemented in Stable Baselines 3. The actor (policy) and critic (value) networks consist of two hidden layers with 256 units each, using ReLU activations. The model is trained for 1000 timesteps, which we found sufficient for convergence in our environment.

Table 9: Hyperparameters used for training

| Hyperparameter | Value |
|---|---|
| Learning rate | 0.0003 |
| Buffer size | 1,000,000 |
| Batch size | 256 |
| Gamma | 0.99 |
| Train frequency | TrainFreq(frequency=1, unit=STEP) |
| Gradient steps | 1 |
| Entropy coefficient (ent coef) | auto |
| Target entropy | -10.0 |
| Tau | 0.005 |
| Policy kwargs | {use_sde: False} |

We see that our $\texttt{EVaR}$ estimation is more robust as it adapts to the samples and the error band is consistent, which shows a contained variance and provides a more conservative estimate of the sample averages, which is desirable in a minimal risk portfolio setting. In Figs .(28a,27a) we visualise our portfolio and how it fairs

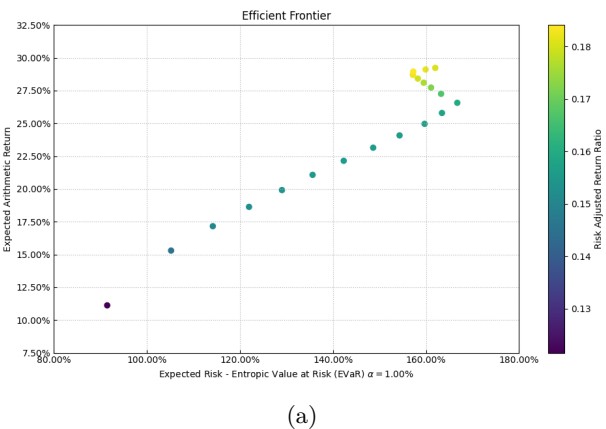

(a)

Figure 27: **(a)** Efficient Frontier of `EVaR` at $\alpha = 0.01$.

against other risk measures like `VaR` and `CVaR`, where we employ RiskFolioCajas (2024) to generate the plots. The importance of these plots is discussed below.

| Date | AAPL | HD | JNJ | JPM | KO | MSFT | PG | UNH | V | WMT |
|---|---|---|---|---|---|---|---|---|---|---|
| 2014-01-03 | -2.20% | -0.16% | 0.90% | 0.77% | -0.49% | -0.67% | -0.11% | 0.71% | 0.07% | -0.33% |
| 2014-01-06 | 0.55% | -0.96% | 0.52% | 0.58% | -0.47% | -2.11% | 0.24% | -1.15% | -0.60% | -0.56% |
| 2014-01-07 | -0.72% | 0.49% | 2.12% | -1.15% | 0.30% | 0.78% | 0.97% | 3.06% | 0.76% | 0.31% |
| ⋮ | ⋮ | ⋮ | ⋮ | ⋮ | ⋮ | ⋮ | ⋮ | ⋮ | ⋮ | ⋮ |
| 2024-01-26 | -0.90% | 1.23% | -0.04% | -0.38% | 0.36% | -0.23% | 0.33% | 1.99% | -1.71% | 0.88% |
| 2024-01-29 | -0.36% | 0.11% | -0.09% | 0.26% | 0.61% | 1.43% | 0.01% | 0.27% | 2.13% | 0.47% |

Table 10: Snapshot of portfolio assets and their adjusted returns over a decade.

### C.5.2 Our Portfolio and Results

**Returns histogram :** provides a visual representation of the distribution of portfolio returns, allowing for a comprehensive analysis of the performance of our `EVaR` agent compared to other risk measures such as `VaR` and `CVaR`. This comparison is crucial as it reveals how our `EVaR`-based portfolio exhibits a more conservative allocation, which is characterized by a lower frequency of extreme negative returns. By evaluating the shape and spread of the distribution, we can gauge the robustness of our portfolio against adverse market conditions, highlighting the potential benefits of utilizing `EVaR` as a risk measure.

The comparison of various risk measures is essential for understanding the effectiveness of our portfolio strategy in managing risk. By analyzing the performance of the `EVaR` portfolio against traditional measures like `VaR` and `CVaR`, we can illustrate the advantages of adopting an entropic approach to risk assessment. Figure (28b) illustrates that these comparisons indicate that our `EVaR` strategy not only limits potential losses but also adapts more dynamically to changing market conditions. This adaptability is reflected in the reduced variance of estimated returns, which is desirable for investors seeking minimal risk.

**Efficient Frontier :** is a key concept in modern portfolio theory, representing the set of optimal portfolios that offer the highest expected return for a given level of risk or the lowest risk for a given level of expected return. By analyzing the Efficient Frontier in the context of our portfolio optimization with `EVaR` measure, we can visually assess the trade-offs between risk and return for various portfolio allocations. Figure (27a) illustrates the Efficient Frontier for our `EVaR`-based portfolio, demonstrating that it achieves a favorable balance between risk and return. This indicates that our optimization strategy effectively identifies portfolios that maximize returns while minimizing downside risk.

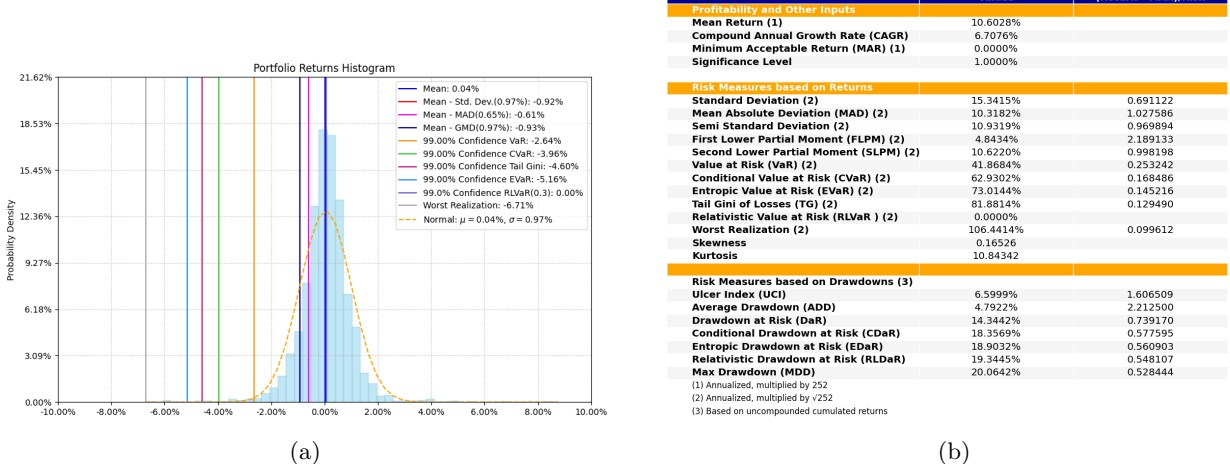

Figure 28: **(a)** Returns histogram with comparison of `VaR`, `CVaR` and `EVaR` which shows `EVaR` portfolio being the most conservative allocation. **(b)** Comparison of various risk measures using a risk-adjusted portfolio.

# D    Reproducibility Details

The experiments were conducted on NVIDIA DGX A100, having an AMD EPYC 7742 64-core processor operating at 1.5 GHz ∼ 3.39 GHz, with GDDR5 32 GB RAM, NVIDIA A100-SXM4-40 GB GPU at 1.41 GHz, and memory clocked at 1.21 GHz.The operating CUDA version for PyTorch 1.13.1 is 11.6 for Python version 3.10.13. The supplementary material provided includes all the experiments with their obtained values, which are reported here in a visual format.

