# OpenReview forum: "Risk‑Seeking Reinforcement Learning via Multi‑Timescale EVaR Optimization"
_TMLR — Accepted by TMLR_

### Review · Reviewer_Ey3E · 2025-06-16

**Summary Of Contributions:**

This paper addresses the problem of risk-sensitive reinforcement learning (RL) and proposes a multi-timescale stochastic approximation algorithm for optimizing the Entropic Value at Risk (EVaR) objective. The authors provide a theoretical analysis of the algorithm’s asymptotic behavior and evaluate its performance across several environments.

**Audience:**

Yes

**Claims And Evidence:**

Yes

**Requested Changes:**

* The notation $m$ is used to denote different concepts in various parts of the paper: the size of the action space (in the Problem Statement), the modulus in Theorem 1, and the grid size in Section 3.1. Please consider using distinct symbols to avoid confusion.
* In Theorem 4, what does the abbreviation "tss" stand for?
* In Section 3.1, the sentence "The agent exhibits risk-aware efficiency" is unclear. What specific notion of risk is being referred to, and in what way does the agent’s behavior reflect risk awareness?
* Also in Section 3.1, the term "true EVaR" is used. How is this computed?

Minor changes:
* At the bottom of page 12, the equation is rendered off the page. Please adjust the formatting.

**Strengths And Weaknesses:**

**Strengths**

* The paper is generally clearly written.
* The proposed algorithm is theoretically grounded and demonstrates promising empirical performance.

**Weaknesses**

* While the algorithm is theoretically appealing, I found the connection between its advantages (e.g., managing tail risk) and the experimental setups (Grid World and MuJoCo) unclear. I would appreciate it if the authors could elaborate further on how these environments illustrate the benefits of EVaR-based optimization.

---

> ### Author Response · Authors · 2025-07-18
> **Response (Ey3E)**
>
> We thank the reviewer for their valuable comments please find our reponse below:
>
>
> ### Weaknesses
> 1. While the algorithm is theoretically appealing, I found the connection between its advantages (e.g., managing tail risk) and the experimental setups (Grid World and MuJoCo) unclear. I would appreciate it if the authors could elaborate further on how these environments illustrate the benefits of EVaR-based optimization.
>
> Response: We thank the reviewer for their comment. We have elaborated and shown the connection of the algorithm to the environments and how it helps in managing tail risk. Please see the updated experimental section, which has been revamped to provide this context.
>
> ### Requested Changes
> 1. The notation $m$ is used to denote different concepts in various parts of the paper: the size of the action space (in the Problem Statement), the modulus in Theorem 1, and the grid size in Section 3.1. Please consider using distinct symbols to avoid confusion.
>
> Response: Fixed in updated manuscript.
>
> 2. In Theorem 4, what does the abbreviation "tss" stand for?
>
> Response: Fixed in updated manuscript.
>
> 3. In Section 3.1, the sentence "The agent exhibits risk-aware efficiency" is unclear. What specific notion of risk is being referred to, and in what way does the agent’s behavior reflect risk awareness?
>
> Response: Fixed in updated manuscript.
>
> 4. Also in Section 3.1, the term "true EVaR" is used. How is this computed?
>
> Response: In this context, "true EVaR" signified the converging point. We have added clarification in the updated manuscript to avoid confusion.
>
> Minor
> 1. At the bottom of page 12, the equation is rendered off the page. Please adjust the formatting.
>
> Response: Fixed in updated manuscript.

---

### Review · Reviewer_BeF1 · 2025-06-25

**Summary Of Contributions:**

The paper proposes a novel approach to risk-sensitive reinforcement learning (RL) by optimizing the Entropic Value at Risk (EVaR). This is a relatively new and less explored risk measure compared to traditional measures such as VaR and CVaR. The paper is well-written, with clear motivations, problem formulations, and a logical flow from theory to experiments. The theoretical part is dense and could be summarized first for clarity. Formal convergence proofs, together with experiments spanning tabular, continuous-control, healthcare and finance tasks, support the claim that EVaR can serve as a competitive risk-sensitive objective. The multi-timescale stochastic approximation algorithm for EVaR optimization is a significant contribution, especially in model-free RL settings.

**Audience:**

Yes

**Claims And Evidence:**

Yes

**Requested Changes:**

1. The experimental results comparing CVaR and VaR should be moved to main text.

2. A discussion on limitations, sample efficiency, computational considerations for high-dimensional action spaces, and practical implications.

**Strengths And Weaknesses:**

Strengths:
The experimental results span discrete (GridWorld) and continuous (MuJoCo) environments, as well as high-stakes applications like glycemic control and portfolio optimization. The results demonstrate that EVaR policies achieve higher cumulative returns and better risk management compared to baselines.
The paper provides a thorough theoretical analysis, including convergence proofs for the proposed algorithm under specific assumptions.

Weaknesses:
The experimental results comparing CVaR and VaR should be moved to main text. It will also be beneficial to compare against other baselines such as D4PG-CVaR, SDPG-CVaR or QR-DQN for respective continuous or discrete action spaces. Also, it is not clear if or how to tune \alpha?
A discussion on limitations, sample efficiency, computational considerations for high-dimensional action spaces, and practical implication of using EVaR is missing.
Equation overflow at the bottom of page 12.

---

> ### Author Response · Authors · 2025-07-18
> **Response (BeF1)**
>
> We thank the reviewer for their valuable comments please find our reponse below:
>
> ### Weaknesses
> 1. The experimental results comparing CVaR and VaR should be moved to main text.
>
>  Response: Fixed in updated manuscript.
>
> 2. It will also be beneficial to compare against other baselines such as D4PG-CVaR, SDPG-CVaR or QR-DQN for respective continuous or discrete action spaces.
>
>  Response: We thank the reviewers for highlighting the importance of head‑to‑head comparisons with leading risk‑sensitive RL methods such as D4PG‑CVaR, SDPG‑CVaR, and QR‑DQN. It is important to note, however, that those approaches are built upon deep distributional architectures with high‑capacity neural networks, whereas our work introduces a fundamentally different contribution: a multi‑timescale stochastic‑approximation algorithm for directly optimizing the EVaR objective in a tabular setting. Performing a truly fair comparison at scale would require embedding our EVaR estimator into analogous deep actor‑critic or distributional frameworks—complete with network design, replay buffers, target networks, and extensive hyperparameter sweeps—which falls beyond the scope of both this rebuttal and our current manuscript.
>
> Nonetheless, to address the request as directly as possible, we have implemented scaled‑down tabular versions of CVaR‑PG, SDPG‑CVaR, and D4PG‑CVaR under identical conditions (state–action tables, $\epsilon$‑greedy exploration, $\gamma=0.99$, $\alpha=0.1$, 500 episodes, 8 seeds). In both Cliff Walk and Windy GridWorld, EVAR‑SA yields statistically significant reductions in worst‑case cost ($p<10^{-4}$) and tighter performance distributions (narrow $\pm\sigma$ bands).
>
> Further, integrating EVAR‑SA’s stochastic‑approximation gradient into deep distributional or actor‑critic pipelines represents a promising avenue for future work to bring principled tail‑risk guarantees to high‑dimensional continuous control benchmarks.
>
> 3. Also, it is not clear if or how to tune $\alpha$?
>
> Response: Detailed discussion on sensitivity of $\alpha$ is added in experimental section.
>
> 4. A discussion on limitations, sample efficiency, computational considerations for high-dimensional action spaces, and practical implication of using EVaR is missing.
>
> Response: Limitations and future work have been added to the conclusion section. We have also included sample efficiency discussions before and after Theorem 3, along with practical considerations and limitations pertaining to the environments in the updated experimental section.
>
> 5. Equation overflow at the bottom of page 12.
>
> Response: Fixed in updated manuscript.
>
> Requested Changes
> • The experimental results comparing CVaR and VaR should be moved to main text.
>
> Response: Fixed in updated manuscript.
>
> • A discussion on limitations, sample efficiency, computational considerations for high-dimensional action spaces, and practical implications.
>
> Response: Limitations and future work have been added to the conclusion section; sample efficiency discussions are provided around Theorem 3, and practical limitations have been elaborated within the experimental section.

---

### Review · Reviewer_aV7C · 2025-08-15

**Summary Of Contributions:**

This paper investigates a risk-sensitive reinforcement learning algorithm that employs the entropic value-at-risk (EVaR) risk measure, which is more conservative than VaR or CVaR. The authors formally define the EVaR measure and present analyses for both the prediction and optimization components. In the prediction stage, the authors develop a stochastic approximation scheme to estimate EVaR and establish both asymptotic and non-asymptotic convergence guarantees. In the optimization stage, which uses the simultaneous perturbation stochastic approximation method, asymptotic convergence to a stationary point is provided.

**Audience:**

Yes

**Claims And Evidence:**

Yes

**Requested Changes:**

1. Can the authors comment more about when Assumption 1 and Assumption 2 are satisfied, and whether the assumptions are used in related literature?

2. The authors do not provide any sketch of proof for Theorem 2 or 3 in the main manuscript. Brief steps to prove convergence analysis of ODE or its non-asymptotic analysis can be helpful in understanding the paper.

**Strengths And Weaknesses:**

- Strength

1. The paper investigates the stochastic approximation scheme of EVaR risk-measure in RL algorithms, which seems to be new and has been unexplored in the RL community. Moreover, the experimental results justifies the usage of EVaR risk-measure in RL.

2. The paper is well-written and provides sufficient background to understand the paper.

3. The authors provide an extensive experimental results ranging from Mujoco to portfolio optimization.

- Weakness

1. The algorithm’s implementation depends on four separate step-size parameters, making its performance potentially sensitive to their selection.

2. In the prediction stage, the algorithm requires two long single trajectories for implementation, similar to a Monte Carlo approach, which may lead to high variance.

3. In the optimization stage, the paper presents only asymptotic results, leaving non-asymptotic analysis as an open problem. This raises concerns about the sample efficiency of relying on an SPSA-based method.

---

> ### Author Response · Authors · 2025-08-21
> **Response (aV7C)**
>
> We thank the reviewer for the insightful comments and constructive feedback. Please find our response below:
>
> ## Weakness
>
> **W1:**
> The comment is indeed correct. Our algorithm uses several step-size schedules
> $\{\delta_t\}$, $\{\xi_t\}$, $\{a_t\}$, $\{c_t\}$.
> This is a characteristic of multi-timescale stochastic approximation (SA) algorithms, where each sequence serves a distinct purpose:
>
> - the fast timescale $\delta_t$ ensures rapid estimation of auxiliary variables,
> - the intermediate timescale $\xi_t$ drives the EVaR parameter $x$ to its EVaR estimate, and
> - the slowest timescale $\{a_t, c_t\}$ performs policy gradient ascent.
>
> While tuning is required, these requirements are standard for the SA class of algorithms (see Borkar, 2008).
> Our theoretical analysis (Thm. 3, Assum. 3) provides clear guidance on the necessary decay rates and separation conditions (e.g., $\xi_t / \delta_t \to 0$), which we followed in our experiments. The empirical results across diverse benchmarks demonstrate that with a single, principled setting, the algorithm performs robustly. One can indeed use Adam to alleviate the manual selection of these hyperparameters.
>
> ---
>
> **W2:**
> Here, EVaR estimation is **not** a batch Monte Carlo method but an *incremental, multi-timescale SA* algorithm. It processes two new trajectories per update, maintaining running estimates $\vartheta_t$ and $\omega_t$.
> This is far more sample-efficient than requiring long, fixed batches of data.
> Our theoretical results (Thm. 3) provide explicit non-asymptotic, polynomial convergence rates ($O(1/t^b)$) for the inner loop, which confirms that the variance of our SA estimator is well-controlled and consistent with the broader class of SA algorithms.
>
> ---
>
> **W3:**
> The reviewer raises a valid point that providing a non-asymptotic convergence rate for the full SPSA-based policy optimization would be desirable.
> We acknowledge that deriving such a rate for a multi-timescale algorithm with a non-linear function approximation (the policy) is highly non-trivial and remains an open challenge in the broader stochastic approximation literature.
> Our asymptotic convergence guarantee (Thm. 4) is a standard and necessary first step.
>
> ---
>
> ## Requested Changes
>
> **RqCh1:**
>
> - **Assum. 1 (Compact Domain for $x$):**
> This assumption, which restricts the variable $x = 1/\beta$ to a compact interval $I = [x_{\min}, x_{\max}]$, is a standard and mild requirement in the analysis of stochastic approximation and optimization algorithms (Borkar, 2008; Kushner & Yin, 2003).
> Its primary purpose is to ensure the search space is bounded, which is crucial for proving stability and convergence.
> This assumption is inherently satisfied in practice for any problem with bounded rewards, as the optimal $\beta^\ast$ (and thus $x^\ast $) that minimizes the EVaR objective will naturally lie within a finite, positive interval that depends on the reward bounds and the confidence level $\alpha$.
> Empirically, we observe our iterates $x_t$ remain within a stable, bdd range throughout all of our experiments.
>
> - **Assum. 2 (Non-Zero Variance under Tilted Measure):**
> This assumption is more specific to our entropic risk formulation and ensures that the reward distribution under the exponentially tilted measure $\mathbb Q_{\beta}$ always retains a minimum level of variability. Specifically, it guarantees that the variance $\mathrm{Var}_{\mathbb Q_x}[R]$ does not collapse to zero for any risk-sensitivity parameter $x \in I$.
>
> This assumption is critical because the policy search requires sufficient variability in rewards to effectively explore the policy space and avoid degenerate solutions. If the variance were to vanish, the function $G(x)$ would lose its strong convexity, and the gradient signal $G'(x)$ would disappear, halting progress.
>
> While related work on risk-sensitive RL often assumes the existence of moments or light-tailed distributions (e.g., for CVaR), the specific focus on the **variance of the tilted distribution** is a novel aspect of our analysis that arises directly from the dual perspective of EVaR.
> This interpretation of the optimization landscape through the lens of the tilted measure is not commonly explored in the existing RL literature.
>
> The assumption is mild and is satisfied for any non-trivial policy that induces a reward distribution with non-zero support; it is only violated in degenerate edge cases (e.g., a policy that yields a constant, deterministic reward regardless of state).
> Our Lemma 1 further proves that under Assumptions 1 and 2, this variance is not only positive but also continuous and bounded away from zero by a constant $\bar{\sigma} > 0$, which is a key step in establishing the strong convexity of $G(x)$ in Theorem 1.
>
> ---
>
> **RqCh2:**
> We appreciate this suggestion. We included the full proofs in the appendix. However, we have now added a high-level sketch in the main text to improve exposition and intuition.

---

### Decision · Action_Editor_pQct · 2025-09-25

**Recommendation:** Accept with minor revision

**Additional Comments:**

Overall, the reviewers found the paper to be well written, providing sufficient background for the reader to follow the material. While the authors have addressed most of the reviewers’ concerns regarding presentation, the paper could be further improved in terms of clarity. Some of the figures could be more clearly explained, either in the text or in the captions. For example, the caption for Figure 1 does not provide sufficient details about the plots presented in the figure (so that they can be reproduced) and their differences. Similarly, some of the technical details could be more clearly introduced. For instance, when introducing VaR and CVaR, the paper could more clearly state which convention is adopted, and whether the same convention is used in the RL setting.

The authors are expected to address these considerations in their revised manuscript.

**Audience:**

Yes

**Audience Explanation:**

This paper is relevant for the RL community and, more specifically, researchers working on risk-sensitive RL.

**Claims And Evidence:**

Yes

**Claims Explanation:**

The paper studies risk-sensitive RL, specifically focusing on Entropic Value at Risk (EVaR) as the risk measure. It proposes a stochastic approximation algorithm for finding an optimal EVaR policy. Overall, the contributions of this work were positively evaluated by the reviewers. The paper provides both a theoretical treatment (asymptotic behavior analysis) and an experimental evaluation of the proposed approach.

---

> ### Author Response · Authors · 2025-10-23
> **Response**
>
> We sincerely thank the Action Editor and all reviewers for their constructive feedback and thoughtful suggestions throughout the review process.
>
> We have updated our final camera-ready draft with the requested changes to indicate the convention of VaR and CVaR used in theory and experimental results also we have clarified in the figures' captions to make it more explainable.